# LLARK 🐦 : A MULTIMODAL MODEL FOR MUSIC

## ABSTRACT

Music has a unique and complex structure which is challenging for both expert humans and existing AI systems to understand, and presents unique challenges relative to other forms of audio. We present LLARK, an instruction-tuned multimodal model for *music* understanding. We detail our process for dataset creation, which involves augmenting the annotations of diverse open-source music datasets and converting them to a unified instruction-tuning format. We propose a multimodal architecture for LLARK, integrating a pretrained generative model for music with a pretrained language model. In evaluations on three types of tasks (music understanding, captioning, and reasoning), we show that our model matches or outperforms existing baselines in music understanding, and that humans show a high degree of agreement with the model's responses in captioning and reasoning tasks. LLARK is trained entirely from open-source music data and models, and we make our training code available along with the release of this paper. Additional results and audio examples are at https://bit.ly/3ZyzbGG .

**Note to reviewers:** all revised/added text will appear in this color.

## 1 INTRODUCTION

The creation, sharing, discovery, and understanding of music are important activities for billions of people around the globe. Music is also distinct from other modalities, and even other types of audio, addressed by existing AI systems. For example, core attributes of music, such as key, tempo, and instrumentation are not present in non-musical audio. Many tasks studied for non-musical audio (e.g. captioning, transcription) require unique forms of understanding when applied to music. However, to date, no model has made progress in music understanding comparable to recent multimodal advances in vision and speech.

Our work addresses these limitations with a model that takes (audio, text) pairs as inputs, and produces text outputs. This form of specifying tasks as text is often referred to as "instruction-following," and fine-tuning pretrained large language models (LLMs) to this end as "instruction-tuning" (Wei et al., 2021; Wang et al., 2022; Taori et al., 2023). Recent works across many modalities have demonstrated that this general multimodal approach (*Language + Multimodal → Language*) can provide a foundation for flexible and even zero-shot multimodal modeling, such as InstructBLIP (Dai et al., 2023), LLaVA (Liu et al., 2023a), LLaMA-Adapterv2 (Gao et al., 2023) and Mini-GPT4 (Zhu et al., 2023).

Multimodal LLMs for audio have been an area of active research (e.g. Guzhov et al. (2022); Elizalde et al. (2023); Deshmukh et al. (2023); Girdhar et al. (2023)), with few exceptions (Doh et al., 2023; Liu et al., 2023b; Manco et al., 2021) focusing specifically on music. However, the challenges of obtaining large, high-quality, richly-annotated music datasets has limited the multitask effectiveness of these works, and most are trained for individual tasks (question answering, captioning).

This paper presents LLARK, a model to address the unique challenges of music understanding. We train LLARK from a set of open-source music datasets using an end-to-end instruction-tuning approach with musical augmentations. Our contributions include:

**Instruction-Tuning Recipe for Multimodal Music Modeling:** We develop an end-to-end procedure for transforming diverse, noisy, unaligned music data into a unified instruction-following

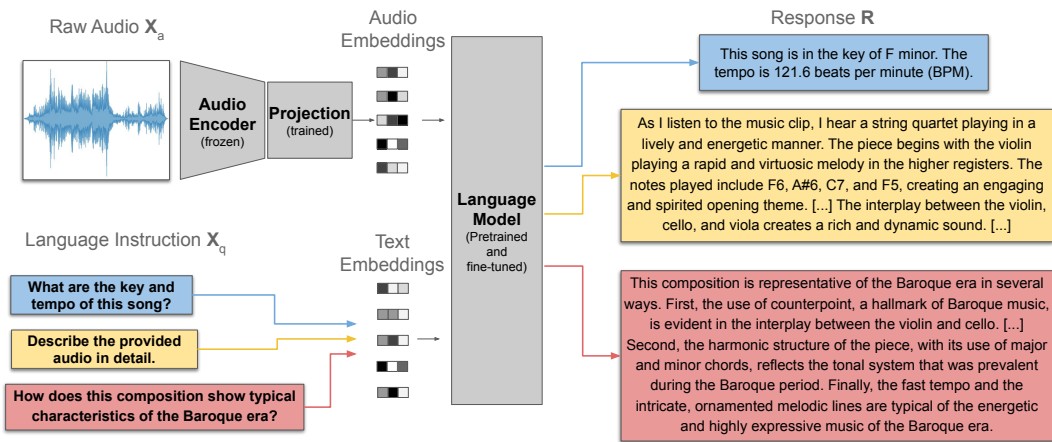

Figure 1: Overview of LLARK. Given audio input and text instructions, LLARK can answer a variety of queries, including music understanding, music captioning, and reasoning queries. Real sample inputs shown, alongside LLARK's outputs for examples from each of the task families addressed in this work (indicated as three colored input/output pairs).

format in three task categories (music understanding, captioning, reasoning), augmenting the data with musical annotations.

**Model Architecture:** We propose an architecture, shown in Figure 1, which leverages (1) a pretrained generative audio encoder, (2) a pretrained language model, and (3) a simple multimodal projection module that maps encoded audio into the LLM embedding space. While the individual components predate this work, LLark is the first work to demonstrate how these can be combined.

**Empirical Evaluation:** We conduct a rigorous evaluation across several music tasks, ranging from classification and regression to captioning and reasoning. We evaluate LLARK alongside state-of-the-art (SOTA) models on benchmark datasets, and with human studies. Via ablation studies, we evaluate the model components and investigate scaling behavior with respect to training data. We show that LLARK achieves improved task performance and greater breadth than previous works.

## 2 RELATED WORK

Our work is related to (*i*) multimodal modeling, (*ii*) Music Information Retrieval (MIR), and (*iii*) foundation modeling for music and audio. (*i*): Several multimodal modeling studies (Liu et al., 2023a; Zhu et al., 2023; Gao et al., 2023; Alayrac et al., 2022) have demonstrated the use of pretrained LLMs and pretrained modality-specific encoders as a paradigm for multimodal modeling. (*ii*) the broader field of Music Information Retrieval (MIR) addresses a diverse set of musical tasks, including estimating properties of music (e.g. key, tempo, tags, instruments, music captioning), such as in Faraldo et al. (2016); Won et al. (2021); Manco et al. (2021) using both machine learning and other approaches. Finally, (*iii*) our work is related to recent efforts to build multimodal foundation models for audio (Guzhov et al., 2022; Wu et al., 2022; Deshmukh et al., 2023; Han et al., 2023; Radford et al., 2023), particularly to studies extending this paradigm to music Liu et al. (2023b).

Our work is distinct from these recent efforts in particular due to (1) use of augmentation to extract musical characteristics from audio; (2) use of a *generative* audio encoder for music, building on the insights from previous work (Castellon et al., 2021); (3) larger and higher-quality training dataset; and (4) thorough empirical evaluations, which demonstrate (a) the increased breadth of LLARK's capabilities and (b) improved performance on the tasks addressed by these prior works.

We provide a more comprehensive overview of related work in Supplementary Section D.

Table 1: Training datasets used in our instruction-generation pipeline. Task families key: 💬: captioning; 𝄞 : music understanding; 🧠: reasoning.

| Dataset | Tracks | Task Families | | |
|---------|-------:|:-------------:|:-:|:-:|
| MusicCaps (Agostinelli et al., 2023) | 2,663 | 💬 | | |
| YouTube8M-MusicTextClips (McKee et al., 2023) | 4,169 | 💬 | | |
| MusicNet (Thickstun et al., 2017) | 323 | 💬 | 𝄞 | 🧠 |
| FMA (Defferrard et al., 2017) | 84,353 | | 𝄞 | 🧠 |
| MTG-Jamendo (Bogdanov et al., 2019) | 55,609 | | 𝄞 | 🧠 |
| MagnaTagATune (Law et al., 2009) | 16,761 | | 𝄞 | 🧠 |

## 3  TASK AND NOTATION

We address the task of generating a "response" sequence of natural language tokens $R = [r_1, \ldots, r_n]$, given a raw audio waveform $X_a = [x_{a,1}, \ldots, x_{a,t}]$ and sequence of input "query" tokens $X_q = [x_{q,1}, \ldots, x_{q,m}]$. Following existing works in language modeling, we model this as a task of auto-regressively estimating $\mathbb{P}(r_i | X_a, X_q, r_{1:i-1})$. This estimate is parameterized by three functions: $\mathcal{A}$, an audio encoder, which computes a representation $\mathcal{A}(X_a)$; $\mathcal{P}$, a projection module which operates on $\mathcal{A}(X_a)$; and $\mathcal{M}$, a language model, which operates jointly on representations of language tokens $X_q$ and audio representations $\mathcal{P} \circ \mathcal{A}(X_a)$. Together, this produces the following formal model:

$$\mathbb{P}(r_i | X_a, X_q, R_{1:i-1}) = \mathcal{M}\big(X_q, \mathcal{P} \circ \mathcal{A}(X_a), R_{1:i-1}\big)$$

This model is illustrated in Figure 1. Let $\Theta = [\theta_{\mathcal{M}}, \theta_{\mathcal{P}}, \theta_{\mathcal{A}}]$ represent the parameters of $\mathcal{M}, \mathcal{P}, \mathcal{A}$ respectively. Our goal is to identify parameters which minimize some loss $\mathcal{L}(\mathcal{M}, \mathcal{P}, \mathcal{A})$ on a dataset $D$ consisting of $(X_a, X_q, R)$ triplets.

We note that many music tasks (classification, regression, sequence-to-sequence) can be encapsulated within this general framework, as long as the desired behavior can be specified with a natural language query (e.g., "What is the tempo of this song in beats per minute (BPM)?") and the output can be represented as a sequence of discrete tokens.

## 4  INSTRUCTION-TUNING DATASET

This Section describes our process for transforming large, diverse, and noisy annotated music datasets into the $(X_a, X_q, R)$ triplets described in Section 3.

Recent works, particularly in the instruction-following domain, have shown that, using relatively small, diverse, and high-quality datasets, pretrained LLMs can be fine-tuned to high quality for tasks such as chat (Taori et al., 2023; Zhou et al., 2023) and vision-language modeling (Gao et al., 2023; Liu et al., 2023a; Zhu et al., 2023). This is a particularly useful insight for the music domain: open-source music datasets are relatively limited in size, and the available datasets often have very different *annotations* due to differences in data collection and intended downstream use. For example, the FMA dataset (Defferrard et al., 2017) contains sparse, user-generated free-form text (among other metadata); in contrast, MagnaTagaTune (Law et al., 2009) contains 160 crowd-sourced binary tags for each track related to musical and stylistic attributes ("hard rock", "bongos", "synth", "weird", etc.).

The instruction-tuning paradigm presents a natural approach to leverage the diversity of these datasets while also converting them into a unified format suitable for training a single model. Indeed, a number of recent works have shown that multimodal models can generalize even when trained on semi-automatically generated text (Wu et al., 2023; Doh et al., 2023; Nguyen et al., 2023). While this lack of feature alignment across datasets has presented a challenge for traditional supervised learning methods that require fixed feature schemas, we hypothesize that this diversity may in fact be an asset for an instruction-tuned model.

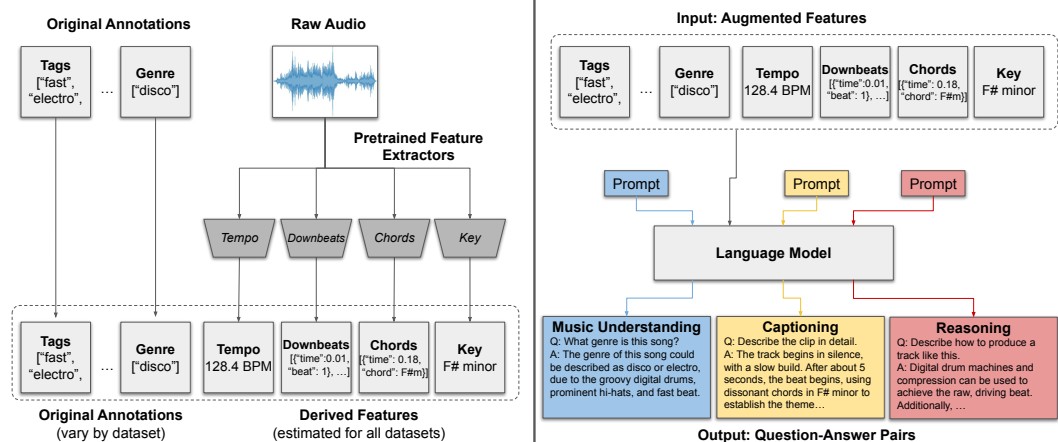

Figure 2: The core LLARK data pipeline. Left: The metadata augmentation procedure. Right: Query-Response generation from augmented data via LLM for the three task families considered in this work (Music Understanding, Captioning, Reasoning).

## 4.1 DATA SOURCES

To construct our instruction-tuning datasets, we use a set of only publicly-available, open source, permissively-licensed music datasets. The datasets we use for training are summarized in Table 1. For each dataset, we collect the audio and any accompanying annotations available for that dataset. The audio from these sources consist of a variety of styles, ranging from classical to electronic music, rock, and experimental, and comprise approximately 164,000 distinct tracks from which we ultimately construct approximately 1.2M instruction pairs over three task families.

Since our audio encoder is limited to 25-second clips of audio, we crop the audio, selecting a random 25-second clip from each track (one clip per track is used).[1]

## 4.2 INSTRUCTION DATA GENERATION

To generate instruction-tuning data from the raw (audio, annotations) pairs, we perform a two-step procedure. A sketch of each step of the procedure is provided in Figure 2.

**1. Metadata Augmentation:** Many music datasets lack important musical information that is useful for music understanding, and can be estimated directly from the audio. In this step, we extract a set of features from the raw audio files using pretrained models.

We extract four features: tempo (in beats per minute, or BPM), global key and mode (e.g. 'F# minor'), timestamped chords, and beat grid (timestamped markers of where downbeats occur, along with numeric indicators of the beat "number", e.g. 1, 2, 3, 4 for a song in 4/4 time). For all features, we use open-source estimators via Böck et al. (2016).

We hypothesize that extracting and providing this information alongside the available annotations can improve the music understanding capabilities of a downstream model and can act as a guardrail against hallucination. Indeed, these features should not only allow the model to learn to directly identify the features in the annotations, but also to reason about how these characteristics relate to higher-level properties of the music, such as genre, harmonic and compositional structure, and emotional content.

**2. Instruction-Tuning Generation via Language Model** Using the original, dataset-provided metadata for each track alongside the augmented metadata (tempo, key, beat grid, and chords), we prompt a large language model to generate question-answer pairs.

We provide the metadata for a given clip as raw JSON, alongside a system prompt. We use distinct prompts for each of the three task families (described in Section 4.3 below), but the overall procedure

---

[1]The sole exception to our one-clip-per-track rule is captioning on MusicNet; see H.8.

is the same. Each prompt describes some of the metadata in the JSON (not all fields are described, as some datasets contain more than 150 annotations), alongside the desired types of question-answer pairs to be produced by the language model.

We use variants of ChatGPT (`GPT3.5-turbo`, `GPT3.5-turbo-16k`, `GPT4`) to generate the training examples. Details on the models and prompts used to generate the data for each dataset-task pair are listed in Sections H.1.1 and H.1.2, respectively. In addition to the existing captioning datasets (MusicCaps, YouTube8M-MusicTextClips), we generate captions for MusicNet, the only dataset in our study where note-level metadata is available.

As the result of this step, we obtain one or more Query-Response pairs for each input example. These Query-Response pairs are then subject to a data filtering step, where we remove pairs which contain certain keywords indicating that instructions were not followed ; see Section H.1.3 for filtering details. Our pipeline ultimately yields approximately 1.2M training samples from the original $164,000$ tracks, as multiple query-response pairs are generated for each track and task family.

## 4.3 Task Families

Our work focuses on three conceptual "families" of tasks, which are used both to prompt the language model for instruction pairs, and in our evaluations (described in Section 6. These task families reflect three forms of understanding associated with music data:

**Music Understanding:** We define as "music understanding" tasks which require identifying a single global property of a piece of music. This includes: tempo, key, genre, instrumentation, etc. These are the lowest-level tasks addressed by our model. These tasks mostly relate to prior work in the Music Information Retrieval (MIR) community.

**Captioning:** Music captioning, similar to image captioning, involves summarizing the content of a piece of audio in language. This task has been of increasing interest to the multimodal and music communities,[2] and has many possible applications including accessibility and music summarization.

**Higher-Level Reasoning:** We define as "higher-level reasoning" (or simply "reasoning") tasks which require either (a) combining knowledge of *multiple* aspects of a track or (b) reasoning about how aspects of this track combine to *external* knowledge about the world. This can include reasoning about how instruments and playing techniques demonstrate the Baroque composition style, or identifying what aspects of a track make it appropriate for certain settings (e.g. dinner party, studying, or a dance club).

Each task comprises a separate system prompt used at instruction data creation time, and a distinct set of evaluations (in Section 6) at test time. Supplementary Table 7 gives the count of instruction pairs generated for each dataset and task family.

## 5 Model Architecture and Training

LLARK is a 12B parameter model consisting of three modules, introduced in Section 3.

We parameterize the language model $\mathcal{M}$ via Llama 2 (Touvron et al., 2023). Specifically, we use the `Llama2-7b-chat` variant which is a 7B-parameter language model fine-tuned for chat applications via Reinforcement Learning from Human Feedback (RLHF).

We parameterize the audio encoder $\mathcal{A}$ via Jukebox-5B (Dhariwal et al., 2020). In contrast to the encoders used for many other multimodal applications, where contrastively-trained models (e.g., CLIP for images/text; CLAP for audio) are often used, Jukebox is a *generative* model. Previous work has shown that Jukebox's representations can be effective features for task-specific linear classifiers (Castellon et al., 2021). We hypothesize that a generative model may create representations of audio which are useful beyond merely classification, and which are sufficiently general to be used by a *single* model to effectively represent many attributes of music simultaneously (our ablation study validates this decision; see Sections 6.5, F). Following Castellon et al. (2021), we use the output of the 36th layer of the Jukebox encoder. Jukebox encodes audio in 4800-dimensional vectors at a frequency of 345Hz, which means that the embedding of a 25s audio clip contains over $4.14 *$

---

[2]See e.g. https://dcase.community/challenge2022/task-automatic-audio-captioning

Table 2: Music Understanding task results on zero-shot datasets. Metrics for the tasks are: MIREX Score for Key estimation, Acc2 for Tempo estimation, Acc@1 for Genre classification, and F1 for Instrument ID. In all cases, a larger score is better. ♮: Essentia task-specific algorithm. ♭: Majority class predictor. For instrument ID, this is the five most frequent instruments (drums, bass, vocals, piano, guitar).

| Task | Dataset | Baseline | IB-LLM | LTU-AS | LLark |
|---|---|---|---|---|---|
| **Key Estimation** | GiantSteps-Key | 0.32 ♮ | 0.048 | 0.00 | **0.70** |
| **Tempo Estimation** | GiantSteps-Tempo | 0.77 ♮ | 0.05 | 0.00 | **0.86** |
| **Genre Classification** | GTZAN | 0.1 ♭ | **0.71** | 0.30 | 0.56 |
| | MedleyDB | 0.125 ♭ | **0.57** | 0.378 | 0.56 |
| **Instrument ID** | MedleyDB | 0.25 ♭ | 0.25 | 0.24 | **0.31** |

$10^7$ floating-point values. Castellon et al. (2021) averages over the time dimension. In contrast, we mean-pool the Jukebox embeddings within 100ms frames, downsampling the embeddings to a frequency of 10Hz and a size of $1.2 \times 10^6$ for a 25s audio clip while retaining temporal information. We note that this is roughly $6\times$ the embedding size of the CLIP ViT-L14 models used in many multimodal vision models.

Our projection module $\mathcal{P}$ is parameterized by a single linear projection layer. This is in following recent multimodal works (e.g. LLaVA Liu et al. (2023a)) which have shown projection layers to be effective for combining strong encoders with strong language models for multimodal modeling in the image-text domain. Using a single layer for $\mathcal{P}$ is also compute-efficient, adding fewer than 0.1% additional parameters relative to the base models.

Our model is trained on (audio, text) inputs in the instruction-tuning format described in Section 4. We use the same preprocessing as in LLAVA (Liu et al., 2023a) to convert instruction pairs into training examples. The model is trained with stochastic gradient descent using the AdamW optimizer and the standard cross-entropy training objective over the response tokens $R$. We freeze the encoder weights and fine-tune both $\mathcal{M}$ and $\mathcal{P}$. Additional training details for reproducibility are provided in Section K.

# 6 EVALUATION

We evaluate our model on all task families described above (music understanding, music captioning, reasoning), to assess the flexibility of our general framework.

## 6.1 BASELINES

For all tasks, we compare our model to other open-source multimodal models capable of generating text from (text, audio) inputs. Specifically, we compare to:

**ImageBind-LLM (Han et al., 2023) (IB-LLM)**: This multimodal model is an improved version of LLaMA-Adapter (Gao et al., 2023) trained on multimodal (text, audio, video, image) embeddings from ImageBind Girdhar et al. (2023) which are combined with a LLaMA language model via interleaved cross-attention layers in the language model.

**Listen, Think and Understand (LTU-AS) (Gong et al., 2023b)** : LTU-AS is an improvement to (Gong et al., 2023c) using Whisper Radford et al. (2023) and TLTR (Gong et al., 2023a) audio encoders and LLaMA-7B language model, integrated via a set of low-rank adapters. LTU-AS is trained on an audio question-answering dataset generated by prompting GPT3.5-Turbo on both musical and non-musical audio.

For Music Understanding and Captioning tasks, we compare to additional task-specific baselines; see Sections 6.2 and 6.3 for details. Additionally, for selected Music Understanding tasks (key, tempo detection), we compare to conventional baselines. More details on all baselines are provided in Supplementary Section J.

## 6.2 Music Understanding (Classification and Regression) Tasks

Our Music Understanding evaluations focus on recognizing the following global properties of music: the overall key and mode of the song (i.e. 'A major' or 'F# minor'); the tempo of the song in BPM; the genre associated with a song; and the set of all instruments present in the song.

For each Music Understanding task, we evaluate on a zero-shot dataset (a dataset which was not used to train the model; note that this is more strict than simply using the "test" split of a training dataset as it requires generalization to a potentially different data distribution and task). Our results are shown in Table 2. We use conventional metrics from the MIR literature for evaluating performance on each tasks; details on these metrics are in Section E.1 and additional results for more datasets are in Section G. While our focus is in comparing to audio-text models capable of completing all tasks, we provide results from task-specific baseline models in Section E.

While LLARK does not reach the level of the strongest task-specific specialised for all tasks, our results show that it achieves strong performance across the zero-shot datasets in Table 2. Indeed, LLARK is the top performer among music-text models for all tasks, besides genre classification, where it achieves the second-highest performance. We hypothesize that the strong genre performance of ImageBind-LLM is due to exposure to (a) popular music and (b) genre tags during the training of its multimodal backbone, ImageBind (ImageBind was trained on a set of web videos and associated text. It is likely that these contained both popular music and genre tags, e.g. as hashtags, including even the popular tracks present in GTZAN, but the ImageBind training set is not publicly available to confirm or disconfirm this hypothesis.). We also show in Supplementary Figure 10 that LLARK's genre prediction errors tend to be related genres higher in the same branch of the genre hierarchy (i.e., predicting "rock" for songs labeled as "metal").

## 6.3 Music Captioning Tasks

Evaluating LLMs for open-ended tasks, such as captioning and reasoning, is an open research problem. Furthermore, we cannot access the raw logits of all baseline models (and these models do not all share the same tokenization scheme), so likelihood-based metrics, such as perplexity, are not possible to compute or compare across all models. Therefore we use human evaluation in this setting, which has been called the "gold standard" of chatbot evaluation Touvron et al. (2023). We also provide additional quantitative evaluation results for these tasks in the supplement (Section G).

We evaluate our models' music captioning capabilities on three datasets: (1) MusicCaps Agostinelli et al. (2023), a recently-introduced music captioning dataset consisting of audio extracted from a wide variety of YouTube videos; (2) MusicNet Thickstun et al. (2017), a dataset consisting of freely-licensed classical recordings; and (3) FMA Defferrard et al. (2017), a diverse set of royalty-free music covering an eclectic mix of genres and styles. For the test split of each dataset, we ask humans to compare captions from our model to those from the baseline models. Details on this procedure are given in Section L.1. The ordering of captions in the interface is always randomized.

In addition to the baseline models described in Section 6.1, we also compare to two additional captioning-specific models: (1) Whisper Audio Captioning (WAC) Kadlčík et al. (2023), a fine-tuned variant of Whisper-Large (Radford et al., 2023) trained for audio captioning, and (2) LP-MusicCaps (LP-MC) (Doh et al., 2023), a Transformer-based multimodal model with a convolutional encoder that operates on audio spectrograms.

Our results, shown in Figure 3, show that humans consistently prefer LLARK's captions. We note that LLARK's performance is particularly strong on the datasets containing solely musical recordings (MusicNet and FMA). The smaller performance gap on MusicCaps could be attributed to the fact it contains many non-musical samples (sound effects, television and radio recordings, etc.), as well as relatively shorter recordings, where superficial captions are less detrimental.

We also evaluate the musical detail of our model's captions using GPT-4. These results in Supplementary Table 5 demonstrate that our model's outputs contain more musical details than baseline models, likely due to our metadata augmentation strategy. In contrast, the baseline models often contain irrelevant or non-musical details, such as imagined descriptions of the appearance of the musicians making the music.

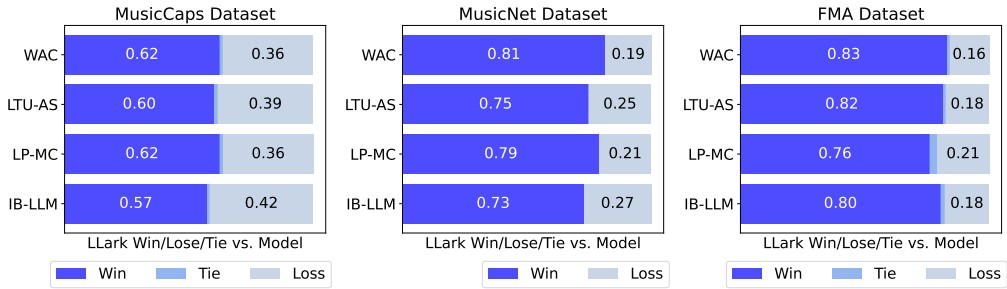

Figure 3: Win rates of LLARK vs. existing captioning models on test data.

We provide additional metrics, including linguistic measures of caption correspondence to the ground truth, token counts, and token diversity metrics, in Section G.2.1.

## 6.4 REASONING TASKS

Evaluating the quality of a models' responses to complex, open-ended questions is an open and unresolved research challenge. Reasoning about music often requires skills and knowledge that only expert musicians possess, including the ability to discern musical details (tempo, key, chords) and knowledge of music composition and production. As a result, we found basic comparisons similar to those in Section 6.3 to be unreliable for evaluating models' reasoning capabilities in initial exploratory evaluations. In this section, we construct two different experiments to assess the quality of our models' responses on reasoning tasks.

First, we conduct a human evaluation based on *audio-to-text matching*. We found that this setup helped mitigate the susceptibility of non-expert raters to model hallucinations and generic responses not grounded in the specific audio. We present raters with a (question, audio) pair from the test split of our data. We also present raters with three randomly-ordered answers to this question, all from the same model. One is the true model response for the given audio; the remaining two are randomly-sampled responses for the same model and prompt but different audio. We ask raters to determine which response best answers the question, for the provided audio. (More details on this evaluation are given in Section L.2.) The results of the human study are given in Figure 4.

Second, we prompt GPT-4 to compare the musical detail of models' outputs on a random subset of 1,000 samples from the test dataset for four datasets. The results for this are shown in Table 3 with the procedure detailed in Section E.3.2.

These results show that LLARK's outputs surpass existing multimodal models in terms of their correspondence to audio and queries. Additionally, they show that LLARK's provide considerably more *musical* detail, validating our data augmentation strategy. While LLARK outperforms existing SOTA models in our study, we observe that the performance is perhaps lower than expected given its strong performance on other task families; we hypothesize that this is due to limitations in the musical expertise of the (non-expert) raters in our study.

## 6.5 ABLATION AND SCALING STUDY

We conduct an ablation study to investigate two main factors: (1) the impact of the language model and audio encoder, and (2) scaling behavior with respect to training dataset size.

The full results presented in Supplementary Section F. Briefly, we find (1) that both the Jukebox audio encoder and the Llama 2 language model contribute to critical performance gains on benchmark tasks; and (2) there are diminishing marginal returns to increased training set size, which aligns with recent work suggesting that small, diverse, high-quality instruction-tuning datasets are sufficient when fine-tuning high-quality encoders pretrained on large pretraining datasets (Zhou et al., 2023; Taori et al., 2023).

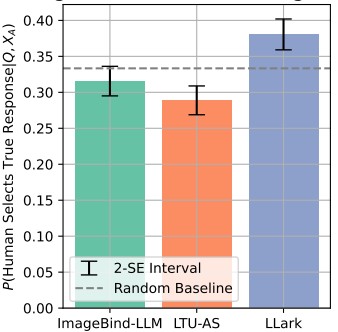

Figure 4: Audio-Text matching rates of human evaluators across 10 reasoning tasks.

Table 3: Win rates of LLARK in GPT-4 musical detail comparison on reasoning tasks.

| Dataset | IB-LLM | LTU-AS |
|---------|--------|--------|
| **MusicNet** | 57.2 % | 90.5 % |
| **FMA** | 72.2 % | 88.8 % |
| **MTG-Jamendo** | 68.1 % | 90.7 % |
| **MagnaTagATune** | 69.5 % | 90.1 % |

## 7 LIMITATIONS

LLARK is limited to the 25-second context window of the Jukebox audio encoder, but in principle, it is possible to extend the context window by e.g. concatenating encodings of consecutive audio segments; we leave such analyses to future work.

Our human evaluations are conducted by non musical-expert annotators. As a result, it is possible that these annotators may lack relevant musical knowledge for certain evaluation tasks, or may be biased toward specific forms of model output.

Finally, LLARK was trained only on the limited available open-source music data. It is likely that training on additional (but copyright-protected) music data would significantly improve the model. However, there are important ethical and legal considerations surrounding the use of such data which are beyond the scope of the current work to address.

## 8 CONCLUSIONS AND FUTURE WORK

This work introduced LLARK, a multimodal model for music using a novel data augmentation strategy, multimodal instruction-tuning dataset, and a generative audio encoder. Our evaluations demonstrate LLARK's music understanding, captioning, and reasoning capabilities at a level of quality unseen so far from a single model.

Our study points to several directions for future work. First, our ablation studies point toward gains from improving both the audio encoder and language model, which are substantially larger than the gains from scaling training data. Future work improving these parts would offer improved multimodal capabilities. Second, our study emphasizes the importance of adding rich musical annotations to multimodal training data. We encourage future audio modeling efforts to incorporate musical annotations even beyond those used in this work. Incorporating future improvements in the feature annotation models used would also increase the underlying quality of LLark's training data, which would likely lead to improved performance on these tasks (key, tempo, etc.). Finally, we note the lack of, and need for, high-quality benchmarks for many musical tasks, including those addressed in this work. Many tasks require high-quality data that is expensive and time-consuming to collect (genre, chord labeling and harmonic analysis, captioning, reasoning). We encourage the field to continue development of such benchmarks and to utilize them to measure future progress, as high-quality evaluation is critical to achieving robust and reliable gains in ML/AI research (Liao et al., 2021).

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

## A    ETHICS STATEMENT

There are important ethical considerations associated with training multimodal music models. These include: bias toward Western music in music datasets and the features used to represent them (i.e. chords in 12-tone scale, instruments in MIDI or common tagging datasets), and potential gender or other biases inherited from the pretrained language model and training dataset annotations (for example, MusicCaps and Magnatagatune annotations sometimes specify the inferred gender of a vocalist, but these may be unreliable, incorrect, or otherwise biased). Additionally, there is no guarantee that the information produced by the model is factually accurate, as these types of models are known to hallucinate in some cases, which should be carefully considered when building applications for such models.

We strongly encourage potential users of LLARK's data, model, and training methods to consider the impacts of each of these factors on the downstream learned model (e.g., the impact of foundation model pretraining data, LLARK multimodal training data, and other factors) on the resulting model. Furthermore, we encourage the risks associated with using a multimodal language model to be made transparent to users in any downstream application of such a model. These include flagging the risk of persuasive but factually incorrect, biased, or harmful outputs.

We provide a Model Card Mitchell et al. (2019) for LLARK in Section M. We encourage readers to consult the Model Card, as it also highlights considerations relevant to ethical training, use, and deployment of LLARK.

## B    REPRODUCIBILITY STATEMENT

We provide several artifacts to reproduce the analysis in this work. These include: scripts to reproduce the model training; details on the datasets used (Section 4 and H); prompts and additional details for instruction data generation (Section 4 and the provided code); hyperparameter and hardware details for model training (Section K). Our code also includes Python scripts and instructions for extracting the metadata used to augment our training examples, and for extracting Jukebox embeddings from audio (modified from the open-source code of Castellon et al. (2021)[3]. We provide exact software dependencies for our code, alongside Dockerfiles to reproduce our training and data preprocessing environments. We will publicly release this code on publication of the paper.

In order to comply with the licenses specified by the artists who contributed to the training data, we are unable to provide the exact training data, instruction data, or trained model weights. Specifically, while our training datasets are open-source and Creative Commons-licensed, each audio file is typically governed by its own license, specified by the artist or rightsholder. Many audio files in the datasets used in our study contain "no derivatives" licenses, which prohibit the sharing of any artifact derived from the audio. Thus would include estimated or extracted metadata and annotations; instruction-tuning Q/A pairs, or model weights derived from these audio files. This, we are not able to share these artifacts in order to honor the license put in place by the original artists who created the music used in this study. However, we provide the technical resources for other researchers to reproduce our methods.

## C    ACKNOWLEDGEMENTS

We would like to acknowledge the many artists, producers, annotators, and researchers who contributed to the creation of the datasets used in this work. It would not be possible without them, and we are grateful for their unique contributions to these valuable publicly accessible resources.

Additionally, we would like to acknowledge the efforts of the developers and contributors of the open-source software tools used in our research. This includes the core teams of the Python packages `torch`, `transformers`, and `webdataset`. We are particularly grateful to the authors of `madmom`[4] and LLaVA[5], which provided a critical technical foundation for this work.

---

[3] https://github.com/p-lambda/jukemir/
[4] https://github.com/CPJKU/madmom
[5] https://github.com/haotian-liu/LLaVA

## D    RELATED WORK

**Music Information Retrieval:** The discipline of "music information retrieval" refers to a broad research area, covering many tasks beyond purely information retrieval. The tasks addressed in this domain reflect the diverse variety of characteristics embodied by music, and the diverse set of stakeholders involved in music creation and consumption (listeners, artists, producers, platforms). This includes: key (Faraldo et al., 2016) and tempo estimation (Schreiber et al., 2020), music transcription (Benetos et al., 2018; Gardner et al., 2021), chord recognition (Pauwels et al., 2019), captioning (Manco et al., 2021), source separation (Cano et al., 2018), music tagging (including genre classification) (Won et al., 2021; George et al., 2001), and musical version identification (Yesiler et al., 2021), among many other tasks. Most prior work in this area focuses on developing task-specific classification or regression models. In contrast, our work is focused on training a generalist model for all tasks which can be framed as *Audio + Text → Text* tasks, which we discuss formally in Section 3.

**Multimodal Learning:**    Multimodal learning has increasingly been explored across all combinations of the text, audio and image/video modalities, with the majority of works focused on the image + text modalities. Within the audio domain, the majority of multimodal approaches are focused on speech or environmental sound (Arandjelovic & Zisserman, 2017), do not contain any music-specific training and often treat music as its own class (i.e. a general "music" class in common datasets such as AudioSet (Gemmeke et al., 2017)) with no fine-grained understanding of unique musical properties such as key, genre, or instrumentation. Multimodal modeling has been explored extensively in the music domain in general, but usually with very specific tasks in mind Simonetta et al. (2019). There have also been explorations of contrastive models for audio, which have included some music-focused training (Elizalde et al., 2023; Wu et al., 2023; Guzhov et al., 2022; Huang et al., 2022; Wu et al., 2022; Ma et al., 2021), but contrastive models are limited to applications that can be framed as a function of distances between predefined set of (audio, text) pairs in the model's embedding space and cannot be used for open-vocabulary tasks or generate free-form text.

**Foundation Models for Audio and Music:** There has been limited work on foundation models for audio, and in particular for music audio. Whisper (Radford et al., 2023) supports a predefined set of speech-related tasks, including transcription and translation, but is confined to only a specific set of speech tasks and does not address music or other forms of audio. Jukebox (Dhariwal et al., 2020) is a music generation model whose embeddings have been shown to be useful for fine-tuning task-specific linear classifiers for lower-level music understanding tasks such as music tagging, emotion classification, and genre classification Castellon et al. (2021). While this has shown promise for specific downstream tasks, Jukebox embeddings have not been more deeply explored as a basis for a foundation model for music understanding (an exception to this is Liu et al. (2023b), which investigated Jukebox embeddings in an exploratory study but did not use them as the basis for their final model). However, we hypothesize that, due to Jukebox's ability to accurately model both global and time-varying properties of music (i.e. produce detailed songs with a consistent tempo, genre, instrumentation, key, etc.) using a single representation, as well as its generative training setting, the representations in its encodings can be the basis for a more general music language model. For this reason, we focus on Jukebox's musical tokens as the basis for our work.

Text-to-audio models have demonstrated promising capabilities to generate music from text, but audio-to-text models that can tackle both close-ended and open-ended tasks are far less common. A recent exception is Deshmukh et al. (2023), which addresses general audio tasks and only a small set of music tasks. Finally, there is a growing literature on music captioning, where models input audio and produce textual descriptions, such as Manco et al. (2021), LP-MusicCapsDoh et al. (2023), WAC Kadlčík et al. (2023)), and MU-LLaMA Liu et al. (2023b). However these models are built to describe musical clips at the level of detail provided based on the training set, and the models are not able to be further "prompted" to perform different types of music understanding tasks.

More broadly, various representation learning methods have also been used to generate task-independent representations of audio (and, in some cases, text) that have been shown to be useful for a variety of downstream tasks. These include contrastive methods such as CLAP Elizalde et al. (2023); Wu et al. (2023), and more general representation learning methods such as MERT Li et al. (2023) and the work of McCallum et al. (2022). However, these methods do not directly

output predictions for target tasks, and thus often rely on either a form of zero-shot adaptation for closed-vocabulary problems, or on linear probing (which consists of fine-tuning a linear output layer directly on the target task of interest). Thus, the utility of general representation learning methods on zero-shot and open-vocabulary problems is limited.

**Instruction Tuning:** Fine-tuning language models on a collection of datasets described via natural-language instructions was originally introduced for language-only tasks in Wei et al. (2021). This paradigm has emerged as a successful approach for a wide variety of modeling tasks, including chatbots Taori et al. (2023) and vision-language models Liu et al. (2023a); Dai et al. (2023); Zhu et al. (2023); Gao et al. (2023). The only application to audio of which we are aware is a recent extension to Gao et al. (2023)[6] which, to our knowledge, has not been formally described or evaluated.

# E  TASK DETAILS

## E.1  MUSIC UNDERSTANDING (CLASSIFICATION/REGRESSION) TASKS

This section provides details on the task background, definition, and metrics for our Music Understanding tasks. For details on the datasets used in these tasks, see Section H.

### E.1.1  KEY ESTIMATION

**Description:** The key represents the dominant harmonic mode of a song. The key of a piece is the group of pitches, or scale, that forms the basis of a musical composition. Understanding the key is useful for many reasons, which include playing a song, harmonizing, and finding other compatible songs (e.g. DJs typically mix songs in the same or compatible keys).

**Metric**: We evaluate key using the MIREX Score[7], on the Giant Steps Key Dataset Knees et al. (2015). This is a measure widely used in the Music Information Retrieval (MIR) field for key estimation. The MIREX Score assigns a value between 0 and 1 representing representing how closely related an estimated key is to a reference key. The relationships between reference and estimated keys, and their associated scores, are given in Table 4.

**Prompt:** For this task, we prompt all models with the phrase: "What is the key of this song?".

Table 4: Scoring function for MIREX Score. This is a standard metric used for evaluating key detection algorithms.

| Relationship | Score |
|---|---|
| Same key and mode | 1.0 |
| Estimated key is a perfect fifth above reference key | 0.5 |
| Relative major/minor (same key signature) | 0.3 |
| Parallel major/minor (same key) | 0.2 |
| Other | 0.0 |

We use the implementation of MIREX scoring in the `mir_eval` library (Raffel et al., 2014).

**Task-Specific SOTA Baseline:** The existing state of the art for key estimation on Giant Steps is the model of Korzeniowski & Widmer (2017), which achieves accuracy of 74.3%. We note that this model was trained directly on audio from the same source (Beatport) and genre distribution as the Giant Steps Key dataset.

**Feature Extractor Performance:** The feature extraction model used for key estimation in our metadata augmentation pipeline Böck et al. (2016) achieved a MIREX score of 0.729 in key estimation on this dataset.

---

[6]See https://github.com/OpenGVLab/LLaMA-Adapter/tree/main/imagebind_LLM
[7]https://www.music-ir.org/mirex/wiki/2021:Audio_Key_Detection

### E.1.2 TEMPO ESTIMATION

**Description:** The tempo, or frequency of beats in a track in beats per minute, is a widely used musical feature in the field of music information retrieval.

**Metric**: Measuring the global tempo of a piece of music is a potentially under-determined task. For many tracks with a fixed tempo of $x$, so-called "octave errors" of $1/2x$ and $2x$ are also plausible tempi. The Acc2 score is originally described alongside the Giant Steps Tempo Dataset in Knees et al. (2015), and considers an estimate to be correct if it is within $\pm 4\%$ of either a third, half, double or triple of the true tempo, thus allowing octave errors of factors of 2 or 3.

**Prompt:** This task, we prompt all models with the phrase: "What is the tempo of this song?"

**Task-Specific SOTA Baseline:** The existing state of the art for tempo estimation on Giant Steps is the model of Schreiber & Müller (2019), as benchmarked in de Souza et al. (2021) which reports an Acc2 score of 0.925.

**Feature Extractor Performance:** The feature extraction model used for tempo estimation in our metadata augmentation pipeline Böck et al. (2016) achieved an Acc2 of 0.947 in on this dataset. We hypothesize that the gap between LLark's performance and that of the feature extraction model is due to the challenges in learning to output numeric labels, illustrated in Figure 7.

### E.1.3 GENRE CLASSIFICATION

**Description:** The genre of a song is a categorization that identifies the song as belonging to a shared tradition or set of conventions[8]. Similar to other properties of music, genre is a subjective label and reflects cultural norms and associations related to a given piece of music Sturm (2013). Most pieces of music are associated with multiple (often many) genres. Despite this, genre classification is a widely-used categorization for music, and so we attempt to address this task as a measurement of our models' ability to understand the cultural associates of a given song.

**Metric:** We use a simple accuracy metric, ACC1, to evaluate genre classification performance. For each model's output, we compute the embedding of the full text. Then, we compare this embedding to the text label of all candidate classes. If the true label (according to the dataset annotation) is the nearest to the model's outputs in embedding space (in terms of Euclidean distance), the prediction is considered correct, otherwise it is incorrect.

**Prompt:** For this task, we prompt all models with the phrase "What genre is this song?"

**Task-Specific SOTA Baseline:** The existing state of the art for genre estimation on GTZAN is McCallum et al. (2022), which achieves accuracy of 0.835 after linear probing on GTZAN.

### E.1.4 INSTRUMENT IDENTIFICATION

**Description:** Instrument identification is a multi-label classification task that consists of predicting the full set of labels present in a given audio clip. Instrument identification is widely useful for many music applications, but it requires precise labels of an audio file in order to know whether an instrument is playing at any given time in the audio (since instruments typically do not play continuously in any given song).

**Metric:** We evaluate instrument identification performance by computing the F1 score on only instruments present in the MIDI protocol[9], treating drums as a single instrument. We ignore instruments not present in the MIDI protocol, such as yangqin and guzheng; we map guitar-like instruments ('lap steel guitar', 'mandolin') to a single 'guitar' instrument and treat drums and vocals as separate instruments.

**Prompt:** For this task, we prompt all models with "List the instruments you hear in this clip, including vocals and drums."

---

[8]https://en.wikipedia.org/wiki/Music_genre
[9]https://en.wikipedia.org/wiki/General_MIDI

### E.2 MUSIC CAPTIONING

Music captioning is the automated description of musical audio using natural language. The task of audio captioning has been of broader interest to the audio research community, see e.g. the 2021 and 2023 DCASE workshop challenge[10].

Measuring the quality of captioning is a subjective and challenging open research task, both in the vision and audio communities. Within the domain of music, different metrics are used, including human evaluation Doh et al. (2023), metrics of token length, diversity and non-duplication of training captions Doh et al. (2023), and other linguistic metrics (BLEU, ROUGE, METEOR) that measure structural and semantic similarity between a predicted and ground-truth captions Deshmukh et al. (2023); Liu et al. (2023b).

We focus primarily on human evaluation of musical captions, as this is a task we believe even non-experts are capable of performing for basic summaries of musical audio, while this is currently hard for machines to assess automatically. As a result, we compare win rates of our model in head-to-head measurements of human preference, in line with works in the music captioning domain (Doh et al., 2023) and broader efforts on LLM chatbot evaluation Touvron et al. (2023).

We believe that the linguistic metrics which are sometimes used to measure captioning performance are not well-suited to musical audio. In particular, this is due to the much larger space of potential musical descriptors used to describe the "contents" of a musical excerpt; while the "main elements" of an image might be considered widely recognizable in an image caption (where these linguistic metrics were originally adopted for captioning), we believe that using them for music introduces an unnecessarily strict dependence on a "ground truth" or reference caption which itself is only a subjective description of the content of the original audio. As a result, we believe that human evaluation (the "gold standard" of chatbot evaluation (Touvron et al., 2023)), comparing a caption to the original audio, is the most appropriate metric for evaluating our model. For comparison, we also provide the linguistic and token-based metrics in Section G.

### E.3 REASONING

For reasoning tasks, we use datasets with the same preprocessing as our other datasets (MusicNet, FMA, MTG-Jamendo, MagnaTagATune).

#### E.3.1 AUDIO-TEXT MATCHING

The prompts we used for the audio-text matching study are:

- **Recreating the audio:** How could a music producer recreate the sounds in this track?
- **Defining characteristics:** What are some characteristics that potentially differentiate the song from other similar songs?
- **Suitable listening environments:** In what kind of environments or situations would someone likely listen to this track?
- **Style and genre:** Describe the styles or genres of this song and explain how the song illustrates each style or genre mentioned.
- **Music professor description:** How would a music professor describe the structure, sound, and instrumentation of this track?
- **Main instrument(s):** What are the main instruments present in this track and how do they contribute to the sound?
- **Main elements:** What are the main elements that give this piece its distinctive style and sound?
- **Modification:** I need to remove one instrument in this track but want to keep the results as close as possible to the original. Which instrument should I pick and why?
- **Emotions:** What moods, emotions or sentiments might the song be trying to convey, and how does it do so?

---

[10]https://dcase.community/challenge2022/task-automatic-audio-captioning

- **Associated products:** What kind of consumer product might be associated with this song, and why?

We manually curate these prompts to be suitable for human evaluation. The audio used for this study is a random subset of 64 tracks from the MTG-Jamendo test dataset, which we selected due to its diversity across genres and its mix of popular and less-popular music. For each of the 64 test tracks, we use the identical set of prompts above for each model (LLARK, IB-LLM, LTU-AS), resulting in a set of $64 \times 10$ outputs which is the cross-product of the test audio and prompts.

### E.3.2 MUSICAL DETAIL

```
You will be provided two different pieces of text ("captions"). Both captions describe the same piece of
music. Your goal is to determine which caption contains the most musical detail.

Musical detail can include any information about the musical characteristics of the audio. This includes:
- instruments present (or absent) in the audio
- notes, patterns, or themes being played by different instruments
- the style, genre, or other general descriptors of the type of music being played
- harmonic characteristics of the song such as the key, mode (major/minor), and chords being played
- techniques used by the performers playing the instruments
- audio effects applied to the instruments (delay, distortion, etc.)
- techniques used in the songwriting or composition of the music
- information about the time signature
- the tempo of the song, e.g. in beats per minute (BPM)
- descriptions of the emotional characteristics of the song

The following would NOT be considered musical details:
- where the song might be played (e.g. in a church, in a dance club, in a video game)
- descriptions of what the performers are doing while they are making the music (what they are wearing, how
they are dancing, etc.)
- subjective judgments about whether the music is good or bad

Since you do not have direct access to the audio being described, assume both captions are correctly
describing the audio and that the information contained in them is true.

IMPORTANT: your goal is to assess only which caption has the most MUSICAL details. Ignore details which are
not about the music.

The provided captions will be labeled "A" and "B". In your response, return only either "A" or "B".
```

Figure 5: Prompt used for GPT-4 musical detail analyses (Tables 3, 5).

For the musical detail study, we use a random subset of 512 samples from the test sets of the four instruction-following datasets shown in Table 3: MusicNet, FMA, MTG-Jamendo, MagnaTagATune. Note that we do *not* use the manually-selected prompts described for the audio-text matching study (although those prompts are similar to prompts that occur in our instruction-following data).

Recent work has demonstrated that strong language models such as GPT-4 can match both controlled and crowd-sourced human preferences well, and can be effective judges in basic language understanding tasks (Zheng et al., 2023). We prompt GPT-4 to determine which of two randomly-selected responses to a query (LLARK vs. a randomly-selected model), for the same audio input, contains more musical detail. We believe that GPT-4 is a suitable judge of this, since this task only assesses the *presence* of musical detail; our experiments in Section 6.2 assess the *correctness* of our model's musical understanding (and show that its performance for basic musical properties, such as key, tempo, genre, and instrument, is strong).

The exact prompt used for the musical detail studies is shown in Figure 5. (This prompt is also used for the musical detail captioning results in Table 5).

## F ABLATION STUDY

We conduct a series of ablations to evaluate the respective components of our model.

### F.1 AUDIO ENCODER ABLATION

First, we ablate the audio encoder module $\mathcal{A}$. We replace the audio encoder with CLAP Wu et al. (2023), a contrastively-trained language-audio model. We follow the same procedure for training this model as for LLARK (same training data and hyperparameters), only changing the audio encodings. We use the LAION CLAP model, and in particular use the recommended CLAP checkpoint for music and the pretrained models available in the CLAP repository[11].

The results of our audio encoder ablation are shown in Figure 6 (top row). Our study shows that replacing the Jukebox encoder with CLAP significantly degrades the model performance on all music understanding tasks. This is consistent with the degradations that have been observed in other related works exploring contrastively-trained music and audio encoders i.e. Liu et al. (2023b); Castellon et al. (2021).

We hypothesize that there are several specific factors that could contribute to the decreased performance with CLAP. First, CLAP's training data differs from that of Jukebox. CLAP's pretraining data consists of 630k audio-text pairs (of which a substantial but unspecified fraction is non-musical sound effects) Elizalde et al. (2023), while Jukebox is trained on 1.2M songs (only music) Dhariwal et al. (2020). Second, the keyword-to-caption augmentation used in CLAP also likely leads to representations that do not capture temporal information, making it difficult for a downstream model to estimate time-varying features such as tempo or groove. Third, CLAP's representations are fundamentally not time-varying: in CLAP, a single 768-dimensional embedding is used to represent audio. This in contrast to our encoder, which uses a $250 \times 4800$-dimensional vector for a 25-second audio clip. It is possible that applying temporal averaging to the Jukebox encodings (instead of the windowed averaging used in our work to compress the embeddings), as in Castellon et al. (2021), would also reduce the performance of a model trained with a Jukebox encoder. Additionally, we note that this comparison may not be compute-matched, as a forward pass on an input with Jukebox encodings includes up to 250 tokens of initial embedding size 4800 before the projection, while with a CLAP model the encoding consists of a single token of size 768.

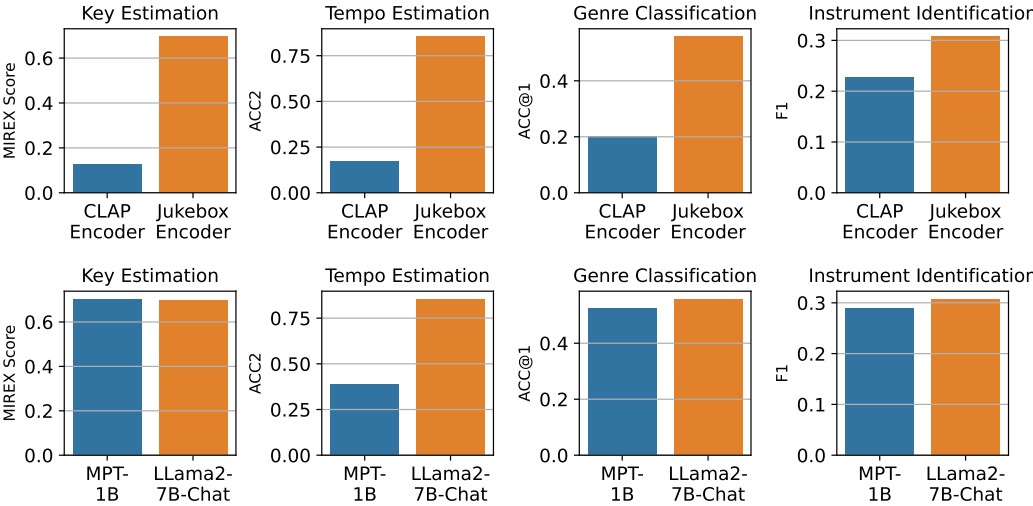

Figure 6: Ablation studies for the audio encoder $\mathcal{A}$ (top) and language model $\mathcal{M}$ (below). See Figure 7 for details on language model ablation in Tempo Estimation task.

### F.2 LANGUAGE MODEL ABLATION

Second, we ablate the language model $\mathcal{M}$, replacing Llama2 language model with MPT-1b-RedPajama-200b-dolly. MPT-1b-RedPajama-200b-dolly is a 1.3 billion parameter decoder-only Transformer pre-trained on the RedPajama dataset and subsequently fine-tuned on the Databricks

---

[11]https://github.com/LAION-AI/CLAP

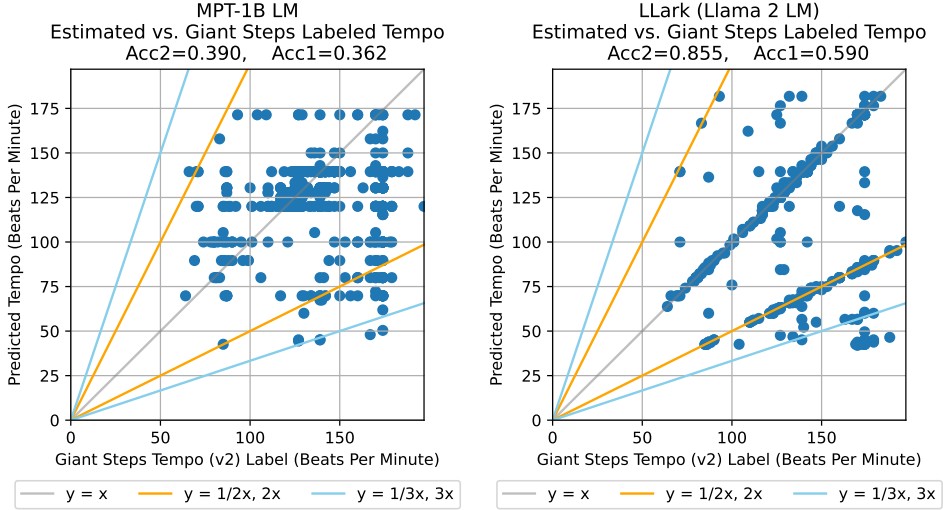

Figure 7: Ablation study results for the tempo prediction task, with lines showing the true tempo and the "octave errors" permitted by the ACC2 metric. Left: true and predicted tempos for model with MPT-1b-RedPajama-200b-dolly. Right: the same values for LLARK. While the MPT-based model achieves performance close to LLARK on the other music understanding tasks (key, genre, instrument ID; Figure 6), we hypothesize that the Llama tokenizer's improved handling of numeric digits allows for improved regression outputs on the tempo prediction task.

Dolly instruction dataset. The model was pre-trained for 200B tokens by sampling from the subsets of the RedPajama dataset in the same proportions as were used by the Llama series of models.

The results of our language model ablation are shown in Figure 6 (bottom row). These results demonstrate more modest gains than the audio encoder ablation study. However, we note two particular findings of interest. First, MPT-1B performance degrades particularly in the task of tempo estimation, the only regression task in our study. We provide some additional results on this task in Figure 7, which shows that the MPT model makes far less precise tempo predictions, often predicting the same numeric values for tracks with widely varying tempos. We hypothesize that this is due to differences in the tokenization scheme between MPT-1B and Llama 2, the latter of which takes special steps to ensure numeric digits (1, 2, 3, etc.) are tokenized individually. Figure 7 reflects the impact of this design decision. Second, we note that Figure 6 only shows performance on music understanding tasks. Subjectively, the performance of Llama 2-based models on captioning and general instruction-following tasks was significantly improved beyond MPT-1B.

## F.3 TRAINING DATA SCALING

It is widely understood that foundation models require large datasets to achieve good generalization performance. However, there is also evidence that the size of *pretraining* datasets is particularly important, and that it may be possible to fine-tune pretrained models (via instruction-tuning or reinforcement learning from human preferences) on smaller datasets Zhou et al. (2023); Taori et al. (2023). We investigate the scaling properties of our model with respect to the training dataset size by training identical, compute-matched models on 1%, 10%, and 50% subsets of the training data. These models are then evaluated on our Music Understanding tasks. The results of this study are shown in Figure 8. It suggests that, while having "large enough" training data is important, the marginal returns to data in our case may be limited; indeed, there is some evidence of model saturation or even small performance drops as dataset size decreases. We note that Figure 8 does not investigate performance on the other tasks we evaluate (captioning, reasoning); subjectively, we find that performance of a model trained on the full training set is improved relative to a model trained on 50% less data.

These results may also reflect the fact that this experiment scales data from the same mix of training distributions, covering the same (mixture) distribution with increasing sample size. It may not

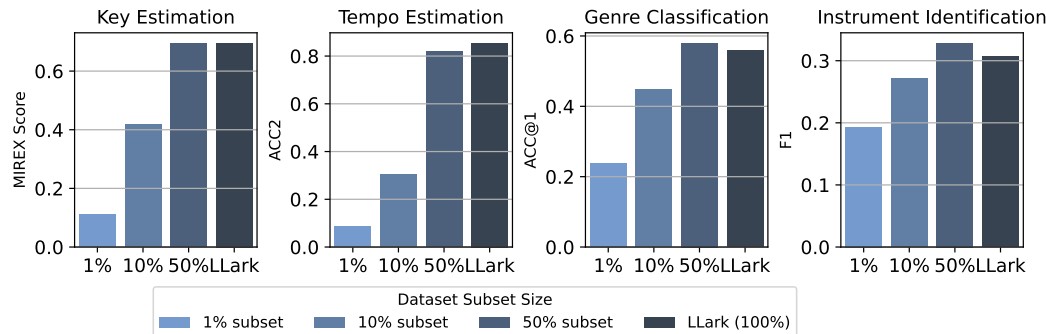

Figure 8: Dataset scaling study on music understanding tasks. We train a model identical to LLARK using 1%, 10%, and 50% of the data respectively.

|  | LAv2 | WAC | LTU | LP-MC |
|---|---|---|---|---|
| **MusicCaps** | 100.0 % | 100.0 % | 100.0 % | 99.6 % |
| **MusicNet** | 100.0 % | 100.0 % | 100.0 % | 100.0 % |
| **FMA** | 100.0 % | 100.0 % | 99.7 % | 95.7 % |

Table 5: Win rates of LLARK vs. other models in GPT-4 evaluations of musical detail on captioning tasks. (See Figure 5 for prompt.)

reflect the potential benefits from new, unobserved datasets. We note, qualitatively, that we explored adding further open-source datasets to our training mixture (Slakh (Manilow et al., 2019), FSL10k (Ramires et al., 2020)), but found that these degraded performance and ultimately excluded them from training.

## G  ADDITIONAL RESULTS

This section provides additional experimental results not included in the main text.

### G.1  MUSIC UNDERSTANDING

We provide additional results to contextualize our model's performance on music understanding tasks.

Figure 9 shows LLARK's predictions vs. ground truth on the Key Estimation task. Figure 9 shows that LLARK generally achieves strong key estimation results. We also note that not all errors are considered equal in this matrix; see Table 4 and Section E.1.1 for details on how the MIREX score is calculated.

Figure 10 shows LLARK's predictions vs. ground truth on the GTZAN Genre Estimation task. While LLARK achieves ACC1 of only 0.56 (relative to approximately 0.71 for the best-performing model on this task), Figure 10 shows that LLARK makes mistakes that appear subjectively reasonable. For example, LLARK tends to mistake "metal" songs for "rock" and categorizes "disco" and "country" songs as "pop" (we note that in both cases, the genres are actually at different levels of the genre hierarchy, and LLARK's predictions are actually a level above the GTZAN-labeled genre in the same branch of the genre tree at https://en.wikipedia.org/wiki/List_of_music_genres_and_styles).

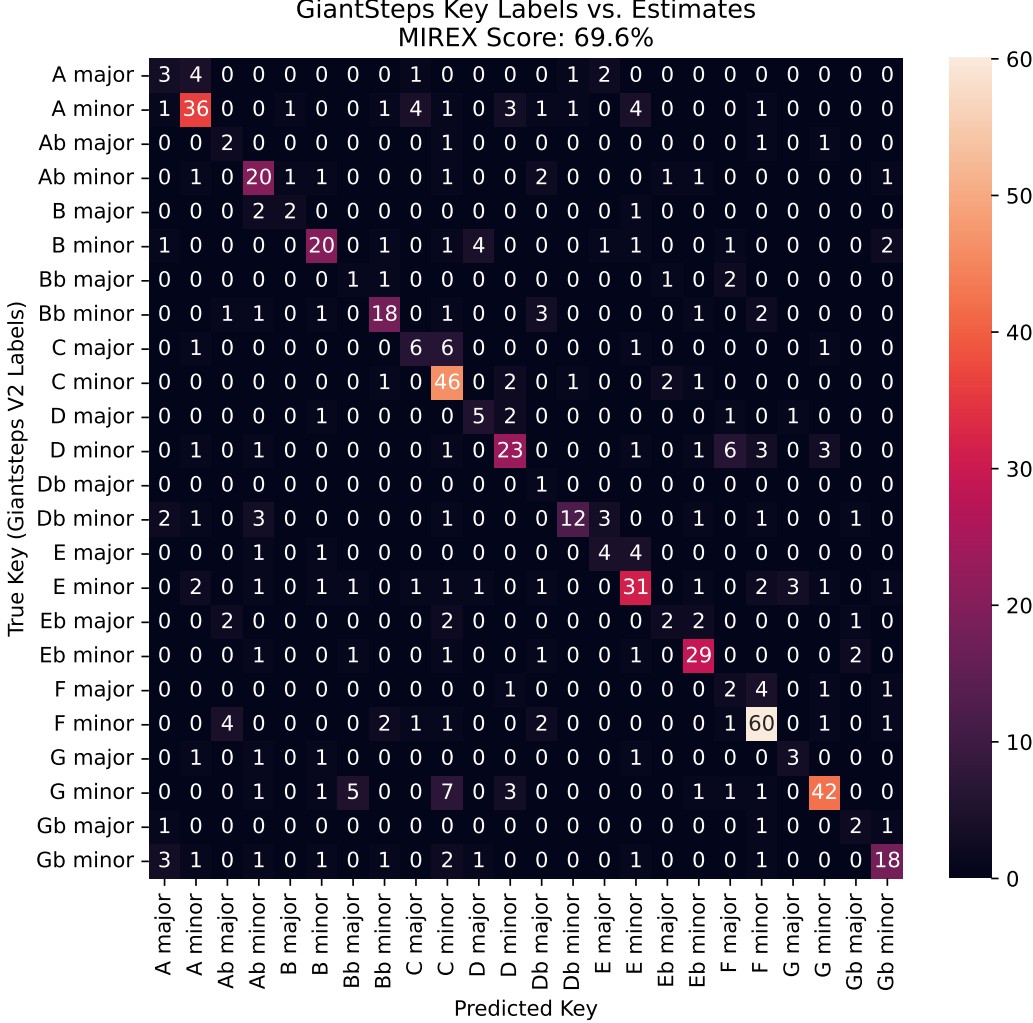

Figure 9: Confusion matrix for LLARK on Key Estimation task. Table 4 and Section E.1.1 provide details on computation of MIREX scores from key estimates.

## G.2 CAPTIONING

### G.2.1 QUANTITATIVE CAPTIONING METRICS

This section provides additional results regarding captioning performance.

Captioning is an inherently subjective task, and the evaluation of captioning models is also an open research question, with varying approaches in the literature. Many audio captioning works have adopted metrics from the *image* captioning community, which themselves were borrowed from the machine translation literature (such as the BLEU score (Papineni et al., 2002)).

These metrics measure the similarity between a proposed caption (or translation) and a reference caption. The differ in how they measure this similarity. They share an emphasis on measuring *lexical similarity*, specifically the similarity between $n$-grams present in the candidate and ground truth captions (either individually, or as a set). However, they are inherently limited for an art form like music, where describing the data has many valid answers, both on the style and on the content itself, and where there is not a ground truth to be as similar as possible.

We provide a set of these linguistic captioning metrics in Table 6, along with some additional experimental results which we believe demonstrate why these metrics may be misleading for music

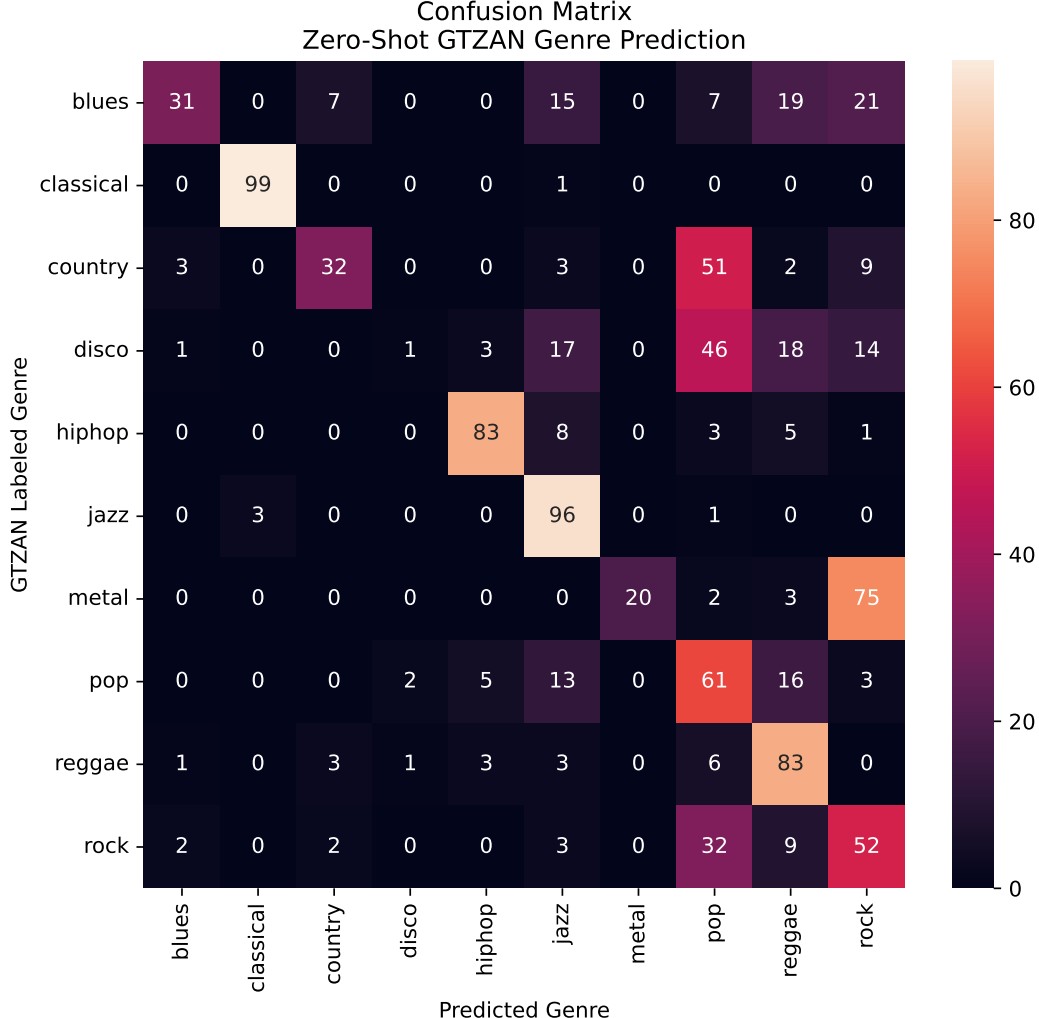

Figure 10: Confusion matrix for LLARK on GTZAN Genre Estimation task.

captioning. Table 6 shows a set of common linguistic captioning metrics for both datasets in our captioning study which include ground truth captions (FMA does not contain ground truth captions; our MusicNet captions are generated by `GPT-3.5-turbo` using the provided metadata and the precise note-leve MIDI data for each track in MusicNet). In addition to the captions for all models in our original study (human evaluation of these results is discussed in Section 6.3), we also provide a second set of results for LLARK using the prompt from our instruction-following study (Section G.4): "Give a short summary of the provided audio"; Table 6 thus contains two entries for the same LLARK model, but using different prompts to elicit captions.

Table 6 demonstrates several interesting results. To interpret these results, we remind the reader that the MusicCaps captions tend to be short, informal, no more than a few sentences, and formulaic (they typically describe (1) the main aspects of a clip, (2) the audio quality, and (3) where such a song might be heard). In contrast, our MusicNet captions tend to be long (2-3 paragraphs), more formal, and focused explicitly on musical qualities (which instruments play, how they interact, compositional aspects of the music, etc.).

First, from Table 6 we see that LLARK's performance according to these metrics varies considerably based on the prompt used. On MusicCaps, LLARK with our standard captioning prompt ("Describe the contents of the provided audio in detail.") is the lowest-performing model; when changing the prompt, LLARK is the second-highest across all metrics on the same dataset. In contrast, LLARK

Table 6: Captioning metrics. *: nonzero, but too small to display in table ($< 10^{-50}$). ♯: uses the prompt "Give a short summary of the provided audio"; see Section G.2.1 for discussion.

| Dataset | Model | BLEU | BLEU-4 | METEOR | ROUGE | CIDER |
|---|---|---|---|---|---|---|
| MusicCaps | LLARK | 0.02 | 0.01 | 0.07 | 0.07 | 0.00 |
| | LLARK ♯ | 0.28 | 0.14 | 0.21 | 0.25 | 0.08 |
| | LTU-AS | 0.21 | 0.09 | 0.19 | 0.25 | 0.01 |
| | IB-LLM | 0.26 | 0.11 | 0.17 | 0.19 | 0.02 |
| | WAC | 0.10 | 0.04 | 0.15 | 0.30 | 0.00 |
| | LP-MC | 0.34 | 0.18 | 0.22 | 0.23 | 0.09 |
| MusicNet | LLARK | 0.62 | 0.45 | 0.38 | 0.33 | 0.05 |
| | LLARK ♯ | 0.09 | 0.06 | 0.21 | 0.45 | 0.00* |
| | LTU-AS | 0.06 | 0.04 | 0.16 | 0.49 | 0.00 |
| | IB-LLM | 0.21 | 0.13 | 0.29 | 0.40 | 0.00* |
| | WAC | 0.03 | 0.02 | 0.09 | 0.59 | 0.00 |
| | LP-MC | 0.20 | 0.10 | 0.21 | 0.27 | 0.00* |

achieves significantly higher scores than any other model on MusicNet (except ROUGE score) with the standard prompt, but tends to perform poorly with the "short" prompt. This reflects both the advantages of our model's instruction-following capabilities, but also the limitations of the linguistic metrics, which largely reward similarity in $n$-gram distribution, but not semantic similarity or going "above and beyond" the reference captions in musical detail, as LLARK tends to do relative to the short MusicCaps captions.

Second, Table 6 shows how existing captioning-only models, such as WAC and LP-MusicCaps, can perform well when the target dataset is close to their training distribution (LP-MusicCaps, as its name suggests, was trained on both MusicCaps and a set of artificially-generated captions designed to match the caption style of MusicCaps), but can perform poorly when the reference captions are linguistically different. Since neither of these models is capable of general instruction-following, this limitation may restrict their ability to generate different forms of captions where needed.

Finally, we believe that Table 6 shows that, while these linguistic metrics may be a useful signal of strict lexical closeness between candidate and reference captions, they can be unreliable and potentially misleading for music captioning. Since many different captions might describe a given music art piece, we believe that these metrics are of limited utility in the music domain, and should be accompanied by other forms of evaluation. (Consider, for example, the number of reviews that might be given for a single song, compared to a caption for a single photo – where these metrics are more widely used.)

In Figure 11, we provide two quantitative measures related to captioning: The number of unique tokens across *all* captions in a dataset, and the average token length of the captions. We tokenize the text on whitespace and punctuation via `nltk.wordpunct_tokenize()` after converting to lowercase; thus, each token is roughly equivalent to a word.

First, Figure 11 demonstrates that LLARK yields captions with consistently higher token counts, relative to the other captioning models. We consider this a positive attribute, as LLARK is capable of providing more detail than the other multimodal models (we show in Section G.4 and Figure 13 that LLARK is also capable of producing shorter captions when desired, but our intention in this study was to demonstrate the maximal level of detail obtainable from each model). This is consistent with the GPT-4 judgments regarding musical detail in Tables 5 and 3, which confirm that the additional tokens in our model's outputs also produce a higher level of *musical* detail.

Second, Figure 11 provides some evidence to support the results of the captioning study. For example, we can see that LLaMA-Adapter tends to produce large numbers of unique tokens in its responses; despite this apparent diversity LLaMA-Adapter performs poorly relative to LLARK in our human evaluations. We hypothesize that this is due to the tendency of LLaMA-Adapter to hallucinate. Its captions often include descriptions of nonexistent visual aspects of the audio (for example, musicians seated in a row, performers dancing) which are irrelevant to understanding the musical or auditory contents of the provided clip. This result also demonstrates the usefulness of

modality-specific training data: LLaMA-Adapter uses an ImageBind backbone which is trained primarily on visual (image + video) data, and which potentially biases the outputs of the model towards these modalities.

Third, Figure 11 provides insight into the relatively poor performance of generic captioning models, such as Listen-Think-Understand (LTU) and Whisper Audio Captioning (WAC). We hypothesize that, because these models are trained on many types of audio data (i.e. sound effects and sound scenes, speech) and the musical subset of their training data is not richly annotated, they tend to produce fewer unique tokens and shorter captions, for example describing a piece simply as "classical music" or "a clip of an orchestra playing".

### G.3 REASONING

Figure 12 provides similar metrics as Figure 11, but for the reasoning test datasets instead of captioning.

**Analysis of ImageBind-LLM results:** Figure 4 shows that raters performed audio-text matching for ImageBind-LLM at a rate slightly below a random baseline. Figure 12 provides some insight into how this can occur. In particular, Figure 12 shows that ImageBind-LLM provides lengthy responses to reasoning questions, generating the largest number of tokens, and the highest number of average tokens, for every dataset evaluated. Qualitatively, we observe that these responses tend to consist of long descriptions with hallucinated details (such as fictional artists and song titles, and detailed visual scenes) which do not correspond to the provided audio. We hypothesize that this reflects the image-alignment strategy used to train the ImageBind backbone Girdhar et al. (2023) which thus leads to an overemphasis on visual elements. As a result, these detailed responses can lead raters to select persuasive responses other than the correct, matching response.

**Analysis of LTU-AS results:** Figure 4 shows that raters performed audio-text matching for LTU-AS at a rate slightly below a random baseline. In this case, as Figure 12 shows, we hypothesize that the main factor was vagueness and lack of detail in the responses. As 12 (left) shows, LTU-AS responses contained the smallest number of unique tokens. This reflects a pattern we observed, where LTU-AS tended to produce similar responses for every piece of audio for a given question, irrespective of the nature of the audio. Additionally, as 12 (right) shows, LTU-AS also tended to produce the shortest responses, more than $2 \times$ shorter than any other model. This reflects the brevity of its responses. As a result, raters had a challenging time disambiguating the model's responses, which tended to be very similar. Further, raters tended to *prefer* outputs which were more detailed, regardless of the length; these factors together produce below-baseline selection rates.

Collectively, the performance of ImageBind-LLM and LTU-AS highlight how the lower bound for audio-text matching is not random chance, but is in fact closer to zero. Consider the extreme case where every option is always presented, but reviewers prefer a single very detailed response, regardless of the provided audio – in this case, the matching rate would be 1 / (number of response choices), which approaches zero as the number of responses grows.

In contrast to ImageBind-LLM and LSU-AS, LLARK provides an intermediate level of details and tokens, while also matching the music content, as Figure 4 shows. This could reflect our model's emphasis on *musical* attributes due to our musical data augmentation: because the other models are exposed to less musical detail during their multimodal training, they may be less sensitive to changes in the audio, and therefore more inclined toward predicting text sequences with high unconditional probability (that is, unconditional of the audio) but potentially poor correspondence with a given piece of audio, while LLARK has stronger musical conditioning.

We also wish to emphasize that, while higher matching rates are certainly achievable for this task, the best matching rates with even expert human responses may not reach 100%, due to factors such as inherent similarity between input audios or responses which make it impossible to perfectly match each audio to the correct response.

We provide additional examples in the demo page associated with this paper which highlight the descriptive, but often either incorrect (describing an imagined song or visual scene not associated with the audio) or generic (verbose, but sufficiently general as to apply to any audio and not specific to the given audio) behavior of the ImageBind-LLM baseline. We hypothesize that this behavior

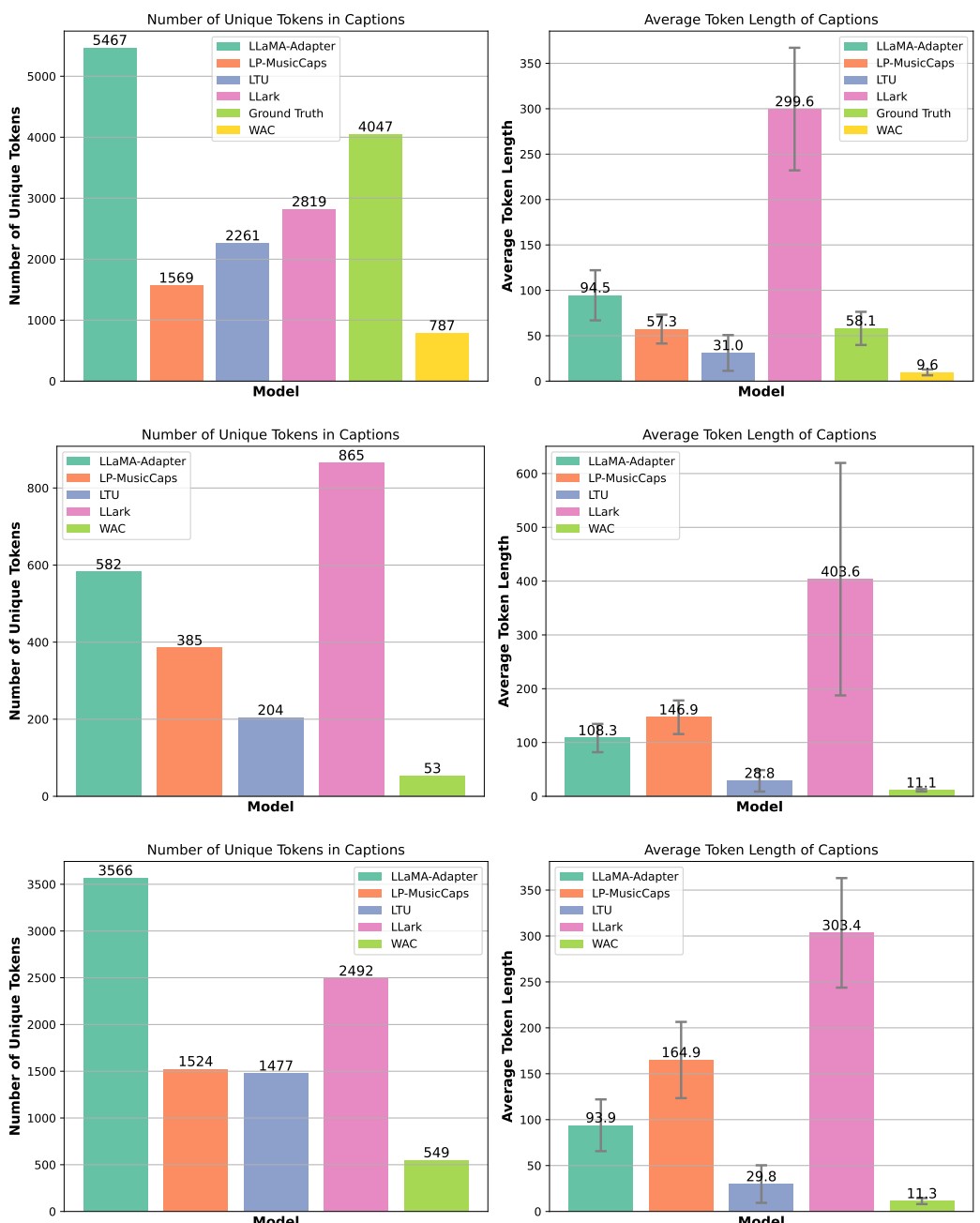

Figure 11: Quantitative metrics for generated captions on each captioning dataset. ("LLaMA-Adapter" refers to the ImageBind-LLM model, which is a variant of LLaMA-Adapter.) Top: MusicCaps dataset. Center: MusicNet dataset. Bottom: FMA dataset.

is linked to the multimodal pretraining of the ImageBind-LLM model (which includes images and videos alongside their corresponding audio).

### G.4 INSTRUCTION FOLLOWING

We design a small experiment to probe LLARK's instruction-following capabilities. For each of the 3 captioning datasets described in Section 6.3 and Figure 3, we do the following: First, we select a random subset of 64 tracks from the test set. Second, for each track, we probe

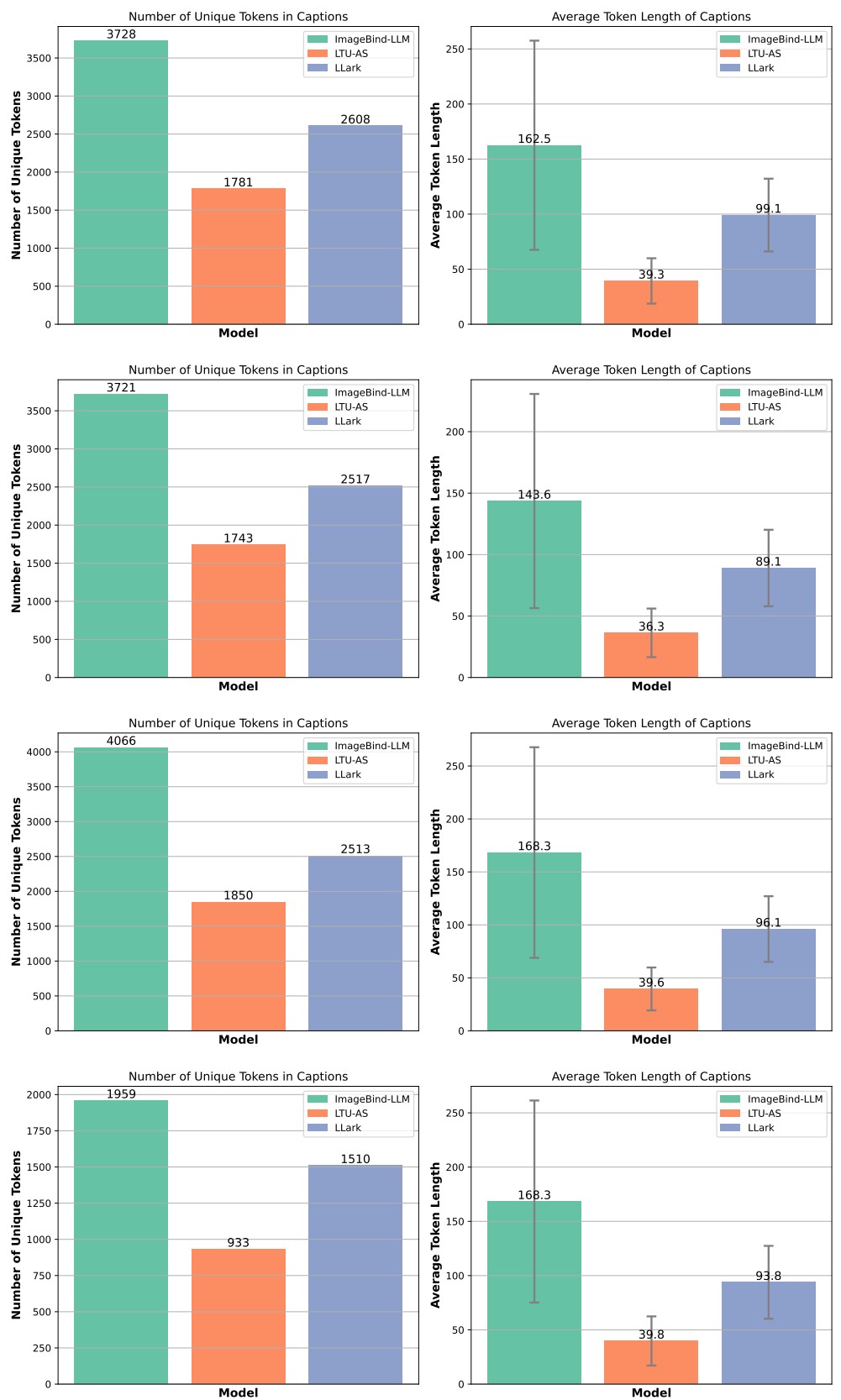

Figure 12: Quantitative metrics for generated responses on each reasoning dataset (test split). ("LLaMA-Adapter" refers to the ImageBind-LLM model, which is a variant of LLaMA-Adapter.) In order from top to bottom: FMA, Magnatagatune, MTG-Jamendo, MusicNet.

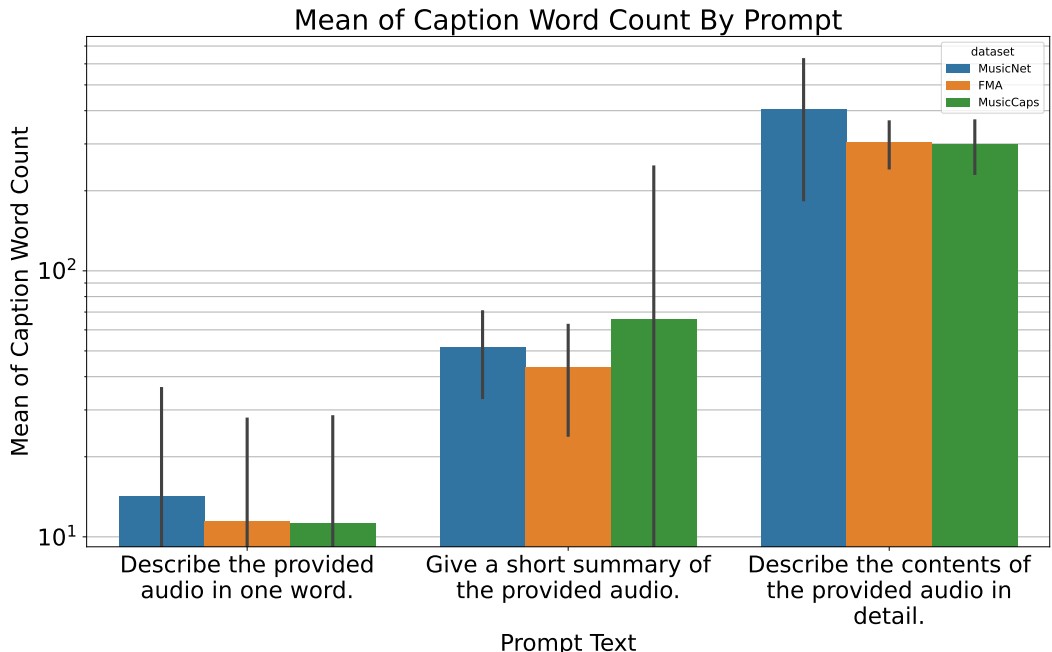

Figure 13: Word counts of LLARK responses across captioning test datasets for varying prompts. As the prompt specifies a greater level of detail, the word count of model outputs increases. Similarly, as prompts specify shorter responses, word counts decrease. LLARK's response consists of exactly one word for $54.9\%$ of the collective outputs across the three datasets. 2-SD error bars shown.

the model with three different prompts designed to elicit different levels of detail (the prompts are shown in Figure 13). Finally, we compute the word count of the model's response (using `nltk.workpunct_tokenize`).

The results are shown in Figure 13. They show that, across all three datasets, the model clearly adapts its responses to instructions. Indeed, for the prompt "Describe the provided audio in one word", LLARK's response consists of exactly one word for $54.9\%$ of the collective outputs across the three datasets.

## H  DATASET DETAILS

This section describes details of our data preprocessing, including any information related to train-test splitting, data filtering, etc.

We provide additional descriptive metrics in Tables 7 and 8.

### H.1  PREPROCESSING

We apply a similar preprocessing step to all datasets in our study. First, we convert all audio to 16-bit 44.1kHz wav files (we convert the audio to other formats where required by other models, e.g. for some baselines that require 16kHz audio). We crop audio into 25-second chunks according to the following procedure: if a track is less than 60 seconds in duration, we retain the first 25 seconds of the clip, or the entire clip, whichever is shorter. If a track is longer than 60 seconds, we crop the interval $[30, 55)$ with probability $p = 0.8$, and the interval $[0, 25)$ with probability $(1 - p)$. This helps ensure that the model observes audio from more active sections of tracks, but still sometimes hears the opening sections of songs.

We retain all annotations accompanying each dataset, and augment these annotations with those extracted according to our augmentation pipeline described in Section 4. The union of the original

Table 7: Per-dataset statistics of instruction pairs.

| Split | Dataset | Captioning | MIR | Reasoning |
|-------|---------|-----------|-----|-----------|
| **Test** | FMA | N/A | 33,185 | 29,053 |
| | MTG-Jamendo | N/A | 7,499 | 3,299 |
| | MagnaTagATune | N/A | 33,342 | 39,171 |
| | MusicCaps | 2,858 | N/A | N/A |
| | MusicNet | 45 | 558 | 139 |
| **Train** | FMA | N/A | 237,599 | 61,373 |
| | MTG-Jamendo | N/A | 407,070 | 173,604 |
| | MagnaTagATune | N/A | 119,352 | 123,727 |
| | MusicCaps | 2,663 | N/A | N/A |
| | MusicNet | 3,799 | 44,457 | 15,533 |
| | YT8M-MusicTextClips | 4,169 | N/A | N/A |

Table 8: Aggregate statistics of instruction pairs across tasks.

| Split | Captioning | MIR | Reasoning | Total |
|-------|-----------|-----|-----------|-------|
| Train | 10,631 0.9 % | 808,478 (67.7 %) | 374,237 (31.4 %) | 1,193,346 |
| Test | 2,903 (1.9 %) | 74,584 (50.0 %) | 71,662 (48.0 %) | 149,149 |

dataset features and the augmented features are provided to the language models at instruction-generation time.

### H.1.1 INSTRUCTION DATA LANGUAGE MODELS

We use variants of ChatGPT to extract the instruction-tuning data for all experiments. However, the exact language model used varies by dataset. We select the OpenAI model as follows: We use GPT-4 for all reasoning tasks. We found that GPT-4 was much more adept at following the complex instructions in the Reasoning task family. For datasets with more than 25k samples, we limit Reasoning data to a random subsample of 25k tracks.

For Music Understanding and captioning tasks, we use GPT3.5-turbo, except when the metadata is too large to fit into the model's context window; in those cases (MagnaTagaTune, MusicNet), we use GPT-3.5-turbo-16k. Note that we only generate captions for the MusicNet dataset; captions for the MusicCaps and YT8M-MusicTextClips dataset are used as provided. We generate captions for MusicNet, and not for other datasets in our sample, because only MusicNet contains note-level metadata (in the form of MIDI data), which allows the caption-generation model to observe the musical events of an audio in detail; we found that captions generated from global, non-time-varying features such as tags or generic instrument labels led to lower-quality captions and degraded downstream performance in initial experiments.

### H.1.2 INSTRUCTION DATA GENERATION PROMPTS

For each task (Music Understanding, Captioning, Reasoning), we use a different base prompt to describe the desired outputs for that task. While other works have used an approach of prompting the language model to output diverse Q-A pairs Liu et al. (2023b), we found that separately prompting the model for more specific forms of query-response pairs led to higher quality data.

The exact prompts used for each task and dataset are provided in the code released in conjunction with this paper. However, we show three example prompts from the same dataset in Figures 14, 15, and 16 to demonstrate their structure.

### H.1.3 INSTRUCTION DATA FILTERING

After generating instruction data, we found that the language model still sometimes did not follow the prompt. For example, it was common for the model to ask about metadata fields which we

```
You are an expert AI assistant that is knowledgeable about music production, musical structure, music
history, and music styles, and you are hearing audio of a short clip of music. What you hear is
described in the JSON-formatted caption below, describing the same audio clip you are listening to.
Answer all questions as if you are hearing the audio clip. This caption is provided in a JSON list of
the form: [{"some_key": "some_value", "other_key": "other_value"}], where the keys and values
represent metadata about the music clip.

The JSON may contain the following fields:

'album.information': optional user-provided information about the album.
'album.tags': optional user-provided tags associated with the track album.
'artist.tags': optional user-provided tags associated with the track artist.
'track.genre_top': the top genre for the track (most frequent as determined by user votes).
'track.genres_all': all genre labels for the track.
'track.information': optional user-provided information about the track.
'track.language_code': the language of the track.
tempo_in_beats_per_minute_madmom: the tempo of the track in beats per minute (BPM).
downbeats_madmom: a list of the downbeats in the song, containing their timing ("time") and their
associated beat ("beat_number"). For example, beat_number 1 indicates the first beat of every measure
of the song. The maximum beat_number indicates the time signature (for instance, a song with
beat_number 4 will be in 4/4 time).
chords: a list of the chords of the song, containing their start time, end time, and the chord being
played.
key: the key of the song.

Design a conversation between you and a person asking about this music. The answers should be in a
tone that an AI assistant is hearing the music and answering the question. Ask diverse questions and
give corresponding answers.
Ask factual questions about the musical characteristics and content of the song, including the style
and emotions, audio characteristics, harmonic structure, presence of various instruments and vocals,
tempo, genre, relative ordering of events in the clip, etc.

Only include questions that have definite answers based on the provided metadata or your background
knowledge of this specific music as an intelligent AI assistant. Write as many question as you can
using the provided inputs. Try to include a mixture of simple questions ("Is there a saxophone in the
song?" "Are there vocals in the clip?" "What is the approximate tempo of the clip in beats per minute
(BPM)?")) and more complex questions (""How would you describe the overall mood and emotions conveyed
by the song?"). Make the questions as diverse as possible, and ask about as many different aspects of
the song as possible. Do not mention the name of the artist in the response.

Again, do not ask about uncertain details. Provide detailed answers when answering complex questions.
For example, give detailed examples or reasoning steps to make the content more convincing and
well-organized. Explain any musical concepts that would be unfamiliar to a non-musician. You can
include multiple paragraphs if necessary. Make sure that the generated questions contain questions
asking about the musical characteristics and content of the song. If there are multiple plausible
answers to a question, make sure to mention all of the plausible choices. Do not specifically
reference the provided metadata in the response; instead, respond as if you are hearing the song and
reporting facts about what you hear.

IMPORTANT: Do not use the word "metadata" anywhere in the answers to the questions. DO NOT disclose
that metadata about the song is provided to you. Always answer as if you are an expert who is
listening to the audio.

Return a single JSON list object containing the question-answer pairs. Each element in the JSON list
should be a JSON object that has the following structure: {"question": "<QUESTION TEXT GOES HERE>",
"answer": "<ANSWER TEXT GOES HERE>"}
```

Figure 14: Example prompt for instruction-data generation. This prompt is for Music Understanding instruction data generation on the FMA dataset.

```
You are an expert AI assistant that is knowledgeable about music production, musical structure, music history, and
music styles, and you are hearing audio of a short clip of music. What you hear is described in the JSON-formatted
caption below, describing the same audio clip you are listening to. Answer all questions as if you are hearing the
audio clip. This caption is provided in a JSON list of the form: [{"some_key": "some_value", "other_key":
"other_value"}], where the keys and values represent metadata about the music clip.

The JSON may contain the following fields:
    genre: a list of genres associated with the song.
    instrument: a list of instruments known to be in the song. Other instruments not listed here may also be
present. If the song contains vocals, they will not be mentioned here.
    mood/theme: a list of moods or themes associated with the song.
    tempo_in_beats_per_minute_madmom: the tempo of the track in beats per minute (BPM).
    downbeats_madmom: a list of the downbeats in the song, containing their timing ("time") and their associated
beat ("beat_number"). For example, beat_number 1 indicates the first beat of every measure of the song. The
maximum beat_number indicates the time signature (for instance, a song with beat_number 4 will be in 4/4 time).
    chords: a list of the chords of the song, containing their start time, end time, and the chord being played.
    key: the key of the song.

Design a conversation between you and a person asking about this music. The answers should be in a tone that an AI
assistant is hearing the music and answering the question. Ask diverse questions and give corresponding answers.

Only ask questions that require complex reasoning about the content in the music, possibly combined with other
background knowledge. Here are some examples of complex questions that you could ask:
- Ask about background knowledge about the music.
- Ask for songs or artists with a similar style.
- Ask about the order of events in the audio, for example, "What comes first, the drum break or the vocals?" Do
the piano and the guitar play at the same time? (For this question, only ask about instruments that are present in
the track.)
- Ask about how to learn to play this type of music.
- Ask how a music producer would create the sounds heard in this track.
- Ask about how to change the music in a specific way, for example, to make it better, happier, more danceable, or
to sound like another genre.
- Ask how a music professor would describe the track.
- Ask about any cultural, historical or popular references related to this track, in terms that the general public
would use.
- Ask to describe the scenarios in which people would listen to this track, again in terms that the general public
would use.
- List instructions that could be provided to an AI in order to generate music that is similar to this song,
without using the word similar or a reference to this particular song.

Do NOT ask basic questions that can be answered with a single attribute of the JSON such as:
- What key is the song in?
- What is the genre of this song?
etc.

Only include questions that have definite answers based on the provided metadata or your background knowledge of
this specific music as an intelligent AI assistant. Write as many question as you can using the provided inputs.
Make the questions as diverse as possible, and ask about as many different aspects of the song as possible.

Again, do not ask about uncertain details. Provide detailed answers to all questions. For example, give detailed
examples or reasoning steps to make the content more convincing and well-organized. Explain any musical concepts
that would be unfamiliar to a non-musician. You can include multiple paragraphs if necessary. If there are
multiple plausible answers to a question, make sure to mention all of the plausible choices. Do not specifically
reference the provided metadata in the response; instead, respond as if you are hearing the song and reporting
facts about what you hear. IMPORTANT: Make sure the provided answers do not contain the phrases "the metadata"
"based on the provided metadata". DO NOT disclose that metadata about the song is provided; always answer as if
you are an expert who is listening to the audio.

Make sure that the questions are complex, and that the detailed answers reflect your expertise as an expert AI
assistant that is knowledgeable about music production, musical structure, music history, and music styles
listening to the clip.

Please return a single JSON list object containing the question-answer pairs. Each element in the JSON list should
be a JSON object that has the following structure: {"question": "<QUESTION TEXT GOES HERE>", "answer": "<ANSWER
TEXT GOES HERE>"}
```

Figure 15: Example prompt for instruction-data generation. This prompt is for Reasoning instruction data generation on the MTG-Jamendo dataset.

```
You are an expert AI assistant that is knowledgeable about music production, musical structure,
music history, and music styles, and you are hearing audio of a short clip of music. What you hear
is described in the JSON-formatted outputs below, describing the same audio clip you are listening
to. Answer all questions as if you are hearing the audio clip. This description is provided in a
JSON dictionary, where the keys and values represent events in the music clip.

The JSON dictionary contains the following keys: "composer", "composition", "movement",
"ensemble", "notes".

The main component of the JSON is the "notes" field, which is a nested JSON dictionary. The keys
in "notes" represent individual instruments, and the values is a JSON list representing all of the
notes played by that instrument in the music clip. Each element in the value JSON list represents
one note played in the music, and includes the following keys:
- start: the start time of the note, in seconds
- end: the end time of the note, in seconds
- pitch: the pitch and octave of the note

In addition to these fields, the JSON also contains the following special annotations:
    - tempo_in_beats_per_minute_madmom: the tempo of the track in beats per minute (BPM).
    - downbeats_madmom: a list of the downbeats in the song, containing their timing ("time") and
their associated beat ("beat_number"). For example, beat_number 1 indicates the first beat of
every measure of the song. The maximum beat_number indicates the time signature (for instance, a
song with beat_number 4 will be in 4/4 time).
    - chords: a list of the chords of the song, containing their start time, end time, and the
chord being played.
    - key: the key of the song.

Provide a detailed musical description of the clip, from the perspective of a musical expert
describing the clip as they hear it being played. Make sure to describe the ordering of the
different instruments (which plays first, which plays at the end), themes or rhythms, arpeggios,
chords, repeating patterns, etc.

The answers should be in a tone that an AI assistant is hearing the music and describing it to a
listener.

Only provide details that are based on the provided metadata or your background knowledge of music
as an intelligent AI assistant. Assume that there are no notes or instruments in the clip besides
those in the "notes" data. Explain any musical concepts that would be unfamiliar to a
non-musician. You can include multiple paragraphs if necessary. Do not specifically reference the
provided metadata in the response; instead, respond as if you are hearing the song and reporting a
rich description of what you hear. The descriptions should keep in mind that this may only be an
excerpt or part of a song, and not the complete song.

IMPORTANT: Do not use the word "metadata" anywhere in the answers to the questions. DO NOT
disclose that metadata about the song is provided to you. Do not specifically reference the
instruments by number (do not say "Violin 1" or "Violin 2"; instead just say "a violin"). Focus
more on a high-level description of the audio, and do not simply list the notes being played;
specific notes (i.e. G5 or F#0) should only be mentioned if they are particularly important to the
description of the song. Always answer as if you are an expert who is listening to the audio. Do
not mention or ask about the track title, artist, or album.
```

Figure 16: Example prompt for instruction-data generation. This prompt is for Captioning instruction data generation on the MusicNet dataset. Note that this is the only dataset where we generate captions, due to the unique MIDI data available in this dataset

Table 9: Keywords and phrases used to filter questions and answers after instruction data generation. Any query-response pairs where the query or response contained a disallowed phrase from the respective list was excluded.

| Query Keywords | Response Keywords |
|---|---|
| "what is the composer", "who is the composer", "tell me about the composer", "name of the composer", "who is the artist", "tell me about the artist", "what tags are associated with the artist", "what are the tags associated with the artist", "is there any information available about the album", "about the album", "name of the artist", "what is the name", "what is the movement", "what is the specific movement", "what is the title", "which movement is", "what is the length of this clip", "duration", "pack", | "metadata", "is not provided", "based on the provided metadata", "based on the provided beat", "based on the provided chord", "based on the provided information", "based on the provided annotations", "no specific mood", "there is no mention of", "there is no specific mention of any", "As an AI assistant, I am unable to", "As an AI assistant, I do not", "it is difficult to determine", "it is not possible to determine", "no information is available about the album", "cannot determine", "violin 1", "violin 2", "violin 3", "viola 1", "viola 2", "viola 3", "pack" |

provided but instructed it not to ask about (e.g. artist, song title), to ask questions where the "answer" was some form of "this answer cannot be determined", or to give answers of the form "from the provided metadata, we can determine...". As a result, we found that filtering the QA pairs was important to improve the data quality, both in order to avoid low-quality training samples being included in the data, and to ensure desirable behavior from LLARK.

We manually collect a set of substrings for both questions, and answers, which represent question/answer formats that violate our instructions. We then remove any Q/A pairs which contain the disallowed substrings in either the question or answer, respectively. Examples of disallowed phrases in the question include "who is the artist" and "what is the length of this clip"; examples of disallowed phrases in the answer include "based on the provided metadata", "it is not possible to determine", and "as an AI assistant, I am unable to".

The list of phrases we remove from questions and answers are shown in Table 9.

The list of phrases we remove from answers is given in Table 9.

Depending on the language model, this filtration process excludes roughly between 1% and 10% of the generated instruction data.

## H.2 FMA

The Free Music Archive (FMA) Defferrard et al. (2017) (`https://github.com/mdeff/fma`) is a dataset comprising $106,574$ Creative Commons-licensed tracks from $16,341$ artists spanning a taxonomy of $161$ genres. FMA includes high-quality audio together with track- and user-level metadata, tags, and free-form text provided by users of an online interface. We use the default set of metadata provided by the FMA Python API, but do not use the extracted audio features (neither the `librosa` nor the Echonest features).

We use the default train/test split for FMA.

## H.3 GIANT STEPS (KEY, TEMPO)

The Giant Steps Key and Tempo datasets, originally proposed in Knees et al. (2015), are two widely-used benchmark datasets for key and tempo estimation. They contain sets of over 600 tracks each, mostly of the electronic genre.

For tempo, we use the 'v2' labels, which are labels that are corrected by human annotators using the process described in Knees et al. (2015). We note that there are three tracks in Giant Steps Tempo that have labeled tempi of 0 BPM; we exclude these tracks.

## H.4 GTZAN

The GTZAN dataset George et al. (2001) contains 1000 tracks of 30 seconds each, uniformly distributed across 10 genres: blues, classical, country, disco, hip-hop, jazz, metal, pop, reggae, and rock. While some of these genres are not entirely distinct from each other and the task itself highly subjective, it is nevertheless a widely-used benchmark in the music information retrieval community, and so we adopt it here.

## H.5 MEDLEYDB

We use the MedleyDB 1.0 dataset[12] Bittner et al. (2014) with fine-grained (time-varying) instrument activity labels. MedleyDB contains 74 tracks covering a variety of instruments and genres (Singer/Songwriter, Classical, Rock, World/Folk, Fusion, Jazz, Pop, Musical Theatre, Rap).

## H.6 MAGNATAGATUNE

The MagnaTagATune dataset[13][14] (Law et al., 2009) is a dataset consisting of audio clips from the Magnatune label[15], annotated by users playing the TagATune game (Law et al., 2009). It consists of a set of approximately $25,000$ 29s-long music clips alongisde a set of 188 binary tags rated by platers of the TagATune game.

## H.7 MTG-JAMENDO

The MTG-Jamendo dataset (Bogdanov et al., 2019) is a dataset built using music available on the Jamendo platform (https://www.jamendo.com/) under Creative Commons licenses and tags provided by content uploaders. The dataset includes annotations for genre, instrument, and mood/theme, which comprise a set of around 195 tags collectively. We use the default autotagging feature set provided by the MTG-Jamendo Python API.[16] We use the full-quality audio, and do not use the mel spectrograms provided with the dataset.

There is no official train-test split for the MTG-Jamendo dataset. We use a random subset of $1,000$ tracks as the test set. The IDs of the tracks in the train and test sets are provided in the code.

## H.8 MUSICNET

We use the official train-test split for the MusicNet dataset.

MusicNet provides a uniquely rich set of annotations, as it is the only dataset in our study which includes complete MIDI transcriptions (precise note-by-note descriptions of the exact pitches and timings of each instrument in the track). As a result, we also generate captions from the MusicNet dataset. This allows us to enrich our pool of captioning data, which is only around 1% of our total training data, and to do so with annotations not available from other captioning dataset in our study.

In order to maximize the number of captioning examples we are able to obtain from MusicNet, we make one exception to our one-audio-crop-per-track rule for MusicNet captioning data only: we take *all* crops from the MusicNet captioning data, which yields a total of 3,799 captioned audio segments from the songs in the MusicNet train split.

We use the improved MIDI data from MusicNet-EM (Maman & Bermano, 2022)[17] in place of the original MusicNet MIDI data.

---

[12] https://medleydb.weebly.com
[13] https://mirg.city.ac.uk/codeapps/the-magnatagatune-dataset
[14] https://musicmachinery.com/2009/04/01/magnatagatune-a-new-research-data-set-for-mir/
[15] http://magnatune.com
[16] https://github.com/MTG/mtg-jamendo-dataset/
[17] https://github.com/benadar293/benadar293.github.io

## H.9    MUSICCAPS

MusicCaps (Agostinelli et al., 2023) is a dataset consisting of 5.5k music-text pairs, with rich text descriptions provided by humans. MusicCaps is extracted from from AudioSet. The overall musicality of the dataset is mixed, and MusicCaps contains a relatively high proportion of musical audio that might not be considered studio-quality: field recordings, sound effects, etc.

Because MusicCaps is only a list of YouTube IDs, the dataset effectively shrinks over time: tracks can be removed from YouTube for various reasons, but the original set of candidate YouTube IDs in MusicCaps is fixed, so the subset of publicly-available YouTube tracks decreases as tracks are inevitably removed. As a result of this shrinkage, it is difficult to compare MusicCaps results directly across works, since different subsets of the data may be available to different authors.

In order to at least partially address this issue, we provide the exact set of YouTube IDs used for evaluation in the code associated with this work. We cannot guarantee direct comparability to other works evaluating on MusicCaps, however, as they have used a different subset of the MusicCaps evaluation dataset.

## H.10    YOUTUBE8M-MUSICTEXTCLIPS

YouTube8M-MusicTextClips[18] (McKee et al., 2023) is a dataset consisting of over $4,000$ high-quality human text descriptions of music found in video clips from the YouTube8M dataset (Abu-El-Haija et al., 2016). It includes 10s audio clips extracted from the videos in YouTube8M, accompanied by human-generated annotations. Since there is no prior work of which we are aware of that uses this dataset for evaluation and captioning data is scarce, we use the *entire* dataset for training (the original split contains 1000 samples for training and 3169 for testing).

# I    EXAMPLE INSTRUCTION-TUNING DATA

For samples of the instruction-tuning data, including question, answer, and the corresponding audio, see the website associated with our paper at `https://bit.ly/3ZyzbGG` .

# J    BASELINE DETAILS

## J.1    ESSENTIA

Essentia[19] is an open-source library and tools for audio and music analysis, description, and synthesis. Essentia packages a variety of different pretrained models. For each task, we select the Essentia model best suited for that task based on the package developers' recommendations alongside our own understanding of the target task. For key estimation, we use the `edma` model, which is derived from the method of Faraldo et al. (2016) and tailored specifically for electronic dance music (which is the genre of the Giant Steps dataset used for tempo evaluation). For tempo estimation, we use their default tempo model.

## J.2    IMAGEBIND-LLM

ImageBind-LLM (Han et al., 2023) is a multimodal language model evolved from LLaMA-Adapter (Gao et al., 2023). It uses an ImageBind (Girdhar et al., 2023) backbone, which allows the model to accept inputs of any of the modalities supported by ImageBind. We note that ImageBind-LLM is not specifically fine-tuned on any audio examples; it instead relies on the ImageBind backbone to ensure good performance across modalities.

## J.3    LISTEN, THINK AND UNDERSTAND (LTU-AS)

LTU-AS Gong et al. (2023b) is "an improved version of LTU" Gong et al. (2023c) and, according to the authors, "stronger in spoken text understanding and music understanding." We use the version

---

[18]`https://zenodo.org/record/8040754`
[19]`https://essentia.upf.edu`

available online in August and September 2023 via the online demo at `https://huggingface.co/spaces/yuangongfdu/ltu-2`.

### J.4 WHISPER AUDIO CAPTIONING (WAC)

We use the fine-tuned Whisper-Large model available in the code and model release associated with Kadlčík et al. (2023)[20]. The model supports different prompt formats, but a format must be selected in order to use the model; we use the recommended Clotho prompt format[21].

### J.5 LP-MUSICCAPS

LP-MusicCaps (Doh et al., 2023) is a Transformer-based captioning model. The model is trained on a large dataset of "pseudo captions", which are generated by providing keyword/tag descriptors to a language model. The model architecture ise a cross-modal encoder-decoder architecture that operates on 10s chunks of log Mel spectrograms, and applies a convolutional audio encoder to the spectrograms in the encoder stack.

## K TRAINING DETAILS

Our model is trained on 4 80GB NVIDIA A40 GPUs. Training takes approximately 54 hours.

The model is trained for 100k steps with a global batch size of 32, cosine learning rate scheduler with 3000 warmup steps and a maximal learning rate of $5e - 5$. We use the AdamW optimizer (Loshchilov & Hutter, 2018) with betas=$(0.9, 0.999)$, $\epsilon = 1e - 6$, and do not apply weight decay. We fine tune both the projection module and the language model throughout, and freeze the audio encoder. The model is trained with BF16 and TF32.

We provide the complete set of software dependencies (Python packages, Conda environment, and Docker image) to reproduce our training environment. We provide additional utilities (scripts + Docker images) to reproduce additional components of our pipeline, such as offline processing of the audio encodings and the extraction of augmented data features. We will publicly release this code on publication of this paper.

## L HUMAN EVALUATION EXPERIMENTS

For all evaluations, we recruit raters via Appen (`https://appen.com`). We restrict the rater pool to only English-speaking raters, and we disable browser-based translation to ensure that raters are not using automated translation tools. Appen includes a test procedure, where raters must accurately complete an assessment of 8 sample questions prior to joining the pool, and must intermittently answer sample questions throughout the rating process to ensure that their rating maintains a standard of quality. Raters are paid for each task they complete. We also apply a control setting in Appen which ensures that no more than 5% of ratings come from a single rater in any task. We use between 382 and 799 workers in each task, depending on rater behavior (raters are free to exit tasks at any point), quality control performance, and the size of the task pool.

### L.1 CAPTIONING

For the captioning task, we provide each model with the prompt "Describe the provided audio in detail," plus an identical audio clip of up to 25 seconds. We ask human raters to assess the quality of these captions.

**Interface:** A screenshot of the interface used in our MusicCaps captioning study is shown in Figure 17. We ask raters to answer the question "Which option is better overall (completely describing the music while also being accurate)?", comparing responses from LLARK and a randomly-selected

---

[20]`https://github.com/prompteus/audio-captioning`

[21]The model was fine-tuned on Clotho and this is the recommended default style; see `https://huggingface.co/MU-NLPC/whisper-large-v2-audio-captioning`

baseline model on a 7-point Likert scale. The ordering of the pairs is randomized so that either model has an equal chance of appearing either first or second.

Only the MusicCaps evaluation included the first question shown in Figure 17. Because MusicCaps contains many examples which do not contain primarily music (sound effects, bodily functions, field recordings, etc.). We use this question to identify samples from MusicCaps where the majority of raters agree that the sample does *not* contain only music, and exclude these samples from our analysis (this affects only $3.04\%$ of the total data resulting from our experiment). The other captioning datasets (MusicNet, MusicCaps) do not require this question, as they are composed entirely of music only.

We randomly sample a total of 1024 pairwise comparisons for each dataset (or as many samples as exist in the dataset, since MusicNet contains only 45 test instances), which equates to approximately 256 pairwise comparisons to LLARK per baseline.

## L.2   REASONING

For reasoning tasks, our human evaluation differs slightly from captioning. We noted in initial pilot studies that (a) baseline models, particularly ImageBind-LLM, tended to give responses that contained either (1) a high degree of specificity with imagined but unverifiable details (such as a track name and artist description, descriptions of an accompanying visual, etc.) or (2) results that were generic and vague enough to apply to nearly any music. We noted that non-expert reviewers had difficulty assessing the quality of these responses. Furthermore, we observed that different models tended to produce structurally consistent responses across all tracks (as shown in Figure 12, with some models tending to produce lengthy responses with others producing much shorter responses). We also adapted our design to control for the model itself (so that reviewers would not simply choose models that they preferred the format of the response, regardless of the content).

Therefore, we designed a study based on audio-text matching. In this study, we present raters with a question + audio pair alongside three randomly-chosen responses from the same model, and then ask the rater to determine which response best answers the question, given the audio. This design encourages model responses that are *specific* to the provided audio, and avoids bias in reviewers that prefer either longer or shorter responses (since these tended to remain consistent for a fixed model, but vary across models, as shown in Figure 12).

We use the MTG-Jamendo dataset for our reasoning study, as it contains a diverse set of genres, including classical, popular, and experimental music.

**Interface:** A screenshot of the interface used in our reasoning audio-to-text study is shown in Figure 18. We ask raters to answer the question "Which option is better overall (completely describing the music while also being accurate)?", comparing responses from LLARK and a randomly-selected baseline model on a 7-point Likert scale. The ordering of the pairs is randomized so that either model has an equal chance of appearing either first or second.

We randomly sample a total of 512 comparisons for each model for this study.

## Listen carefully to the following music example.

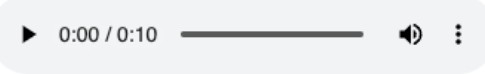

**Option A**
electronic dance music plays continuously

**Option B**
As I listen to the music clip, I can hear a combination of various instruments playing together. The song has a strong and energetic feel, with a fast tempo of approximately 139.5 beats per minute. The rhythm is driven by a prominent drum beat, which provides a steady pulse throughout the clip. The drums are played with intensity, adding to the overall energetic atmosphere of the song.

Apart from the drums, there is also a bass guitar that adds depth and groove to the music. The bass guitar plays a repeating pattern of notes, creating a rhythmic foundation for the other instruments to build upon. The bass guitar's notes are played in sync with the drums, creating a cohesive rhythmic structure.

In addition to the drums and bass guitar, there are also synthesizers that provide melodic and harmonic elements to the song. The synthesizers create a layered and textured sound, adding to the overall complexity and richness of the music. The synthesizers play a combination of sustained chords and melodic lines, contributing to the song's melodic and harmonic structure.

The song is in the key of C# minor, which gives it a slightly dark and moody feel. The chords played in the song are primarily C# major and C# minor, with some variations in between. The chord progression creates a sense of tension and release, adding to the emotional depth of the music.

Overall, the music clip I'm hearing is a dynamic and energetic piece with a strong rhythmic foundation. The combination of drums, bass guitar, and synthesizers creates a vibrant and layered sound. The melodic and harmonic elements add depth and complexity to the music, making it an engaging and captivating listening experience.

## Questions

**Does the audio clip contain only music? (Musical instruments, percussion, and musical vocals with no other sounds or background noise.)** (required)
○ Yes
○ No
○ Not sure/there was a problem displaying the audio

**Which option is better overall (completely describing the music while also being accurate)?** (required)
○ A is much better
○ A is better
○ A is slightly better
○ About the same
○ B is slightly better
○ B is better
○ B is much better

Figure 17: Screenshot of the rating interface used for captioning evaluation on MusicCaps.

## Listen carefully to the following music example.

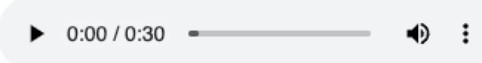

▶   0:00 / 0:30    ━━━━━━━━━    🔊   ⋮

## Question

Describe the styles or genres of this song and explain how the song illustrates each style or genre mentioned.

### Response A

The song is a blend of pop, rock, and folk genres. The pop influence is evident in the catchy melody and the straightforward rhythmic structure. The rock elements are noticeable in the energetic guitar riffs and the powerful drum beats. The folk influence can be heard in the use of traditional chord progressions and the storytelling lyrical style.

### Response B

The song appears to be a blend of several genres including house, electronic, dance, and techno. The tempo of 125 beats per minute is common in these genres, providing a steady rhythm that is conducive to dancing. The song's time signature, indicated by the maximum beat number being 4, is typical of house and techno music. The electronic elements may come from the use of synthesizers or digital sound samples, while the dance elements could be characterized by the repetitive, upbeat nature of the rhythm and the 4/4 time signature.

### Response C

This song can be described as a fusion of electronic, funk, and reggae genres. The electronic genre is reflected in the use of synthesized sounds and electronic beats. The funk influence is evident in the rhythmic grooves and basslines. The reggae elements are noticeable in the off-beat rhythms and the use of minor chords. These genres come together to create a unique and eclectic sound that blends different musical traditions and styles.

### Which response best answers the question, given the provided audio? (required)

○ Response A
○ Response B
○ Response C

Figure 18: Screenshot of the rating interface used for reasoning evaluation on MTG-Jamendo.

## M MODEL CARD

This section presents a Model Card Mitchell et al. (2019) for LLark.

### M.1 MODEL DETAILS

**Person or organization developing model:** LLARK was developed at [ANONYMIZED] by [ANONYMIZED]. **Model date:** LLARK was developed in 2023.

**Model version:** This paper describes version 1.0 of the model. Further releases are not planned.

**Model type:** LLARK is a decoder Transformer model. Its fundamental architecture is that of Llama 2 Touvron et al. (2023), with a Jukebox audio encoder Dhariwal et al. (2020) and a single multimodal adapter layer.

**Information about training algorithms, parameters, fairness constraints or other applied approaches, and features:** Our dataset construction procedure and features are described in Section 4. Our training procedure is described in K.

**Paper or other resource for more information:** This paper is the main resource for LLARK. Additional results are available at `https://bit.ly/3ZyzbGG`. Training and preprocessing code to reproduce our results is available at [ANONYMIZED].

**Citation details:** Please cite this paper.

**License:** LLARK is released under the Apache License, Version 2.0.

**Where to send questions or comments about the model:** Please send correspondence to the corresponding authors indicated on this paper.

### M.2 INTENDED USE

**Primary intended uses:** LLARK is intended to be used for research purposes only.

**Primary intended users:** our primary intended users for this release are researchers in machine learning, audio, music information retrieval (MIR) and related disciplines.

**Out-of-scope use cases:** non-research uses of LLARK should be considered out-of-scope.

### M.3 FACTORS

**Relevant factors:** The most significant factors we identify correspond to "groups" (Mitchell et al. (2019), §4.3.1). The training data for LLARK is based on a variety of sources, which include: user-generated data posted to the Internet; crowdsourced labels; and features estimated using trained machine learning models. The data is processed using Large Language Models (variants of ChatGPT, for text/labels) and Jukebox (for audio), each of which were trained on undisclosed datasets collected from a variety of sources. Each of these elements can introduce bias into the data. These may include: bias toward Western music; bias toward certain gender identities; or biases toward particular forms of language.

**Evaluation factors:** Many of the same factors described above apply to evaluation. In particular, our evaluations reflect certain structural assumptions about music (the use of a consistent tempo; 12-tone scale; and instrumentation). Additionally, we use English-speaking non-expert raters (recruited as described in Section L) for our human evaluations. These raters may introduce their own biases into the evaluation process; in particular, it is possible that these raters do not assess technical and structural musical properties of the model's generated responses.

### M.4 METRICS

**Model performance measures:** Our model is evaluated using a variety of metrics, described in Section 6. The evaluation metrics are described in detail in Section E.

**Decision thresholds:** Our model does not use a decision threshold.

**Variation approaches:** Many of the musical metrics used in our study do not provide precise theoretical estimates of variation (i.e. Acc2, MIREX Score). However, we report the sample sizes of our evaluation sets, which are used to estimate confidence intervals for accuracy-based metrics and binary proportions (i.e., win rates). Wherever such estimates are available and our results do not hold with $p < 0.01$, we do not report or discuss them as practically significant differences.

### M.5 EVALUATION DATA

We use a variety of evaluation datasets; the datasets, motivation, and preprocessing are described in Section 6 and Section H. We also describe evaluation metrics in Section E.

### M.6 TRAINING DATA

We use a mixture of 6 training datasets; the datasets, motivation, and preprocessing are described in Section 4.

### M.7 QUANTITATIVE ANALYSES

Our quantitative analysis is summarized in Section 6.

### M.8 ETHICAL CONSIDERATIONS

There are several ethical considerations, relating both to inputs (i.e., the training data) and outputs.

With respect to inputs: the inputs to our model are public, open-source, Creative Commons-licensed audio and associated annotations. However, each individual audio file can have its own, potentially more restrictive license. Many of the audio files include "no derivatives" licenses. We encourage users of the datasets to familiarize themselves with the restrictions of these licenses; in order to honor such licenses, we do not release any derivatives from the training data in this paper (including query-response pairs or trained model weights).

With respect to outputs: LLARK is a machine learning model trained to generate text conditional on (text, audio) inputs. Its outputs can be factually unreliable, but can also be presented confidently and fluently. As a result, we encourage any users of the model to carefully consider their potential uses of any models based on our training framework (since our model itself is not planned for release). This includes considering the risks of incorrect or misleading text which may be difficult for both experts and non-experts to detect, as well as potential offensive or malicious uses of the model through inputs to its audio and text modalities.

### M.9 CAVEATS AND RECOMMENDATIONS

We recommend further research on all dimensions necessary to improve and understand the performance of models similar to LLARK. This includes improved and publicly accessible training data for music research; better foundation models and architectures; and improved evaluations (both evaluation methodologies, and datasets) specific to music research. In particular, we encourage the development of datasets and evaluation methods that reflect all styles of music, not strictly Western or popular music. Finally, we also encourage the development of bias detection methods that can detect and mitigate biased or harmful outputs in the audio-language modeling domain.

## N FAILURE CASES

This section describes our qualitative, exploratory observations regarding common observed failure modes of LLARK. Our intention in this section is to provide transparency and insight into potential failures of LLARK in order to empower potential future developers and users of such models, and to spur future research on understanding and mitigating failure modes of multimodal audio language models.

Below, we separately identify and discuss a set of failure cases. These failure cases are identified based not only on frequency of occurrence, but also due to their potential impact on downstream users or to their perceived similarity or difference to potential risks of other large models.

## N.1 FAILURE CASE: INCORRECT DETAILS IN LONG-FORM RESPONSES

While we conduct a controlled evaluation of LLARK's music understanding capabilities in Section 6.2, by design, these evaluations isolate *only* music understanding tasks (key classification, tempo prediction, etc.). One failure mode we observed while evaluating LLARK was an apparent decrease in the model's description of these properties when doing so in the context of longer-form text outputs.

As a concrete example of this, we provide an output for LLARK on a reasoning task from the MTG-Jamendo dataset on `https://bit.ly/3ZyzbGG`, for track 223092 from this dataset. Two independently-generated query-response pairs are shown in Table 10. The outputs show signs of incorrect details in the model's outputs: in both responses, LLARK describes the audio as having a tempo of 120BPM, despite the true tempo being roughly 139 BPM. Furthermore, while E minor (Em) is the correct key of the song, LLARK also provides and incorrect key, E major, which would not traditionally be compatible with the key E minor.

These samples reflect a broader trend of LLARK sometimes showing decreased performance and core music understanding tasks during longer-form generation. We hypothesize that this is due to a combination of (1) biases from the pretrained models, and (2) a lack of this form of supervision in the training data. (1) is evident, for example, in the model's bias toward 120BPM – we hypothesize that this is a common tempo observed by the language model during its pretraining (as it is not a prevalent tempo in our training data). (2) is possible due to the fact that, in generating reasoning outputs, we explicitly instruct the language model generating query-response pairs *not* to ask about basic musical details such as tempo and key; as a result, these details may be underrepresented in the reasoning training data and the model may be less likely to learn to generate correct key and tempo data during longer-form outputs to reasoning queries.

**Mitigation strategies:** This failure mode could be addressed through a combination of training data improvements and improved prompting. Our music understanding evaluations (Section 6.2 suggest that LLARK is capable of correct musical detail in isolation. This suggests that our training process is effective (as the "music understanding" task family explicitly generates query-response pairs reflecting these types of details); it is likely that the model simply had limited exposure to "correct" musical detail in reasoning training examples. Changing the instruction-generation prompts to encourage richer musical detail could increase supervisory signals for LLARK at training time. As a second mitigation strategy, improved prompting of LLARK could help mitigate the use of erroneous musical details at inference time, for example, by leaving them out if unnecessary, or via strategies such as chain-of-thought prompting which have been shown to improve quality and consistency of outputs in other domains.

## N.2 FAILURE CASE: OUT-OF-DISTRIBUTION INPUTS AND NON-MUSICAL AUDIO

Another failure case we observed was when the model was provided with non-musical audio. Such audio occurs, for example, in the MusicCaps dataset, which can contain inputs such as the sounds of bodily functions, animal noises, or background noise, all of which are not represented in LLARK's training data mixture.

Example captioning outputs on non-musical audio samples from MusicCaps are shown in Table 11. These examples illustrate how, when provided with non-musical inputs such as birds singing or "a cacophony of burping and farting sounds," LLARK generates text as if the inputs were music from its training data.

This behavior is likely due to a combination of factors. For example, the audio encodings are obtained from JukeBox, which is trained only on music, not sound effects or ambient sounds. Although it is possible that the Jukebox training data could have contained a small fraction of samples of such audio, it is likely that the model's representations are learned to represent musical audio. As a result, the encodings for non-musical inputs may be less informative. An additional cause of this out-of-distribution degradation is likely that the fine-tuning process of LLARK also

| Prompt | LLARK Response |
|---|---|
| How could a music producer recreate the sounds in this track? | A music producer would need to focus on creating an atmospheric soundscape to recreate the sounds in this track. They would likely use a variety of synthesizers, both digital and analog, to create the layered textures that form the basis of this music. The producer would also need to create a steady beat, likely using a drum machine or software, to keep the rhythm. The tempo would be set to around 120 beats per minute to match the pace of the original track. The chord progression would need to be programmed or played to match the sequence of E minor, E major, and D major. |
| What are some characteristics that potentially differentiate the song from other similar songs? | One characteristic that potentially differentiates the song from other similar songs is the use of the synthesizer as the main instrument. The synthesizer is a versatile instrument that can create a wide range of sounds and textures, allowing for unique musical expressions. Additionally, the song's tempo of 120 BPM and the use of the E minor key contribute to its distinctive sound. The combination of these elements, along with the specific chord progression and downbeat pattern, gives the song its own unique identity within the electronic genre. |

Table 10: Selected reasoning outputs illustrating "Incorrect details in long-form responses" failure mode; MTG-Jamendo Track 223092 (see online supplement for audio).

was curated to favor musical audio. While small amounts of ambient and sound-effect data may exist in the datasets described in Section 4, the majority of the training data consists of various forms of amateur and studio-quality music, not "found sounds" or sound effects.

**Mitigation strategies:** One simple strategy to mitigate this behavior would be to add non-musical inputs to the training data of future models. Such training sets (for example, of background and ambient noises, speech, and sound effects) are widely available, and could improve robustness to such forms of audio.

N.3 FAILURE CASE: MISSING OR INCOMPLETE CONTEXT

An additional failure mode of LLARK is failure to provide complete context. Here, we are referring to the limited ability to completely describe certain tasks, world states, or background information necessary to fully answer a query or to present all possible reasonable responses.

One example of this behavior is apparent in the first row of Table 10. Here, LLARK is asked how to recreate a track, and it answers that a producer "would likely use a variety of synthesizers, both digital and analog." While this is indeed one potential strategy, the model's response ignores another important strategy: using a computer.

**Mitigation strategies:** This failure mode reflects the challenges of both getting language models to completely represent knowledge of the world, and also to *communicate* this knowledge in their outputs. While the former is a problem best addressed during pretraining of the language model itself, the latter may be addressed by improved prompting to LLARK (i.e. "describe all potential options") and improvements to the training data pipeline (by adjusting instruction-generation prompts to encourage more thorough, exhaustive, or contextually-rich responses to queries; or by improving our data filtering pipeline to explicitly select for such responses).

N.4 FAILURE CASE: POPULARITY BIAS

We observe a tendency for the model's generations to trend toward "popular" or more strongly-represented musical categories, labels, styles, and descriptors during generation. This is evident, for example, in the GTZAN genre confusion matrix (in Figure 10), where the model systematically prefers "metal" over "rock" and "pop" over "disco" (note that this confusion matrix represents the embeddings of LLARK's outputs, not the exact text of the outputs themselves; see Section 6 for

| MusicCaps Caption (MusicCaps Track ID) | LLARK Response |
|---|---|
| (ID: -pUfYFcsgG4) This recording contains environmental sounds such as birds singing. Then bells start playing a melody with a long release time. This is an amateur recording. This song may be playing outside in a park at a fixed time. | As I listen to the audio clip, I hear a combination of various sounds and elements. The clip starts with a short section of silence, followed by a sustained chord that creates a sense of tension and anticipation. This chord is then followed by a series of quick, staccato notes played on a piano. The piano notes are played in a fast and rhythmic pattern, adding a sense of energy and movement to the music. Throughout the clip, I also hear the sound of a cello playing long, sustained notes. The cello adds a rich and warm tone to the music, providing a contrast to the piano's quick and percussive notes. The cello's notes are played with a smooth and legato style, creating a sense of flow and fluidity. In addition to the piano and cello, I also hear the presence of a synthesizer. The synthesizer adds electronic elements to the music, creating a modern and futuristic sound. The synthesizer's sounds are layered and textured, adding depth and complexity to the overall composition. The tempo of the music is moderate, with a steady beat that provides a sense of stability and structure. I can hear a clear rhythm established by the piano and cello, which is supported by the presence of downbeats. The downbeats occur at regular intervals, marking the beginning of each measure and providing a sense of rhythmic stability. Overall, the audio clip features a combination of piano, cello, and synthesizer, creating a unique and captivating musical texture. The tension and anticipation established by the initial chord, combined with the rhythmic patterns and melodic lines of the piano and cello, make for an intriguing and engaging listening experience. |
| (ID: IOzWDVGWRng) This track features a cacophony of burping and farting sounds which are used in an instrumental manner to make up a Christmas jingle. There's a bell in the background and the sound of sleds and wings flapping. | As I listen to the audio clip, I hear a dynamic and energetic piece of music. The song is characterized by a fast tempo, with a tempo of approximately 171.4 beats per minute. The rhythm is driven by a consistent and strong beat, which gives the song a sense of urgency and excitement. The song features a variety of instruments, including drums, synthesizers, and possibly some percussion elements. The drums provide a solid foundation, with a driving rhythm that adds to the overall energy of the song. The synthesizers create melodic lines and add a modern and electronic feel to the music. The percussion elements, if present, add additional rhythmic layers and texture to the song. Throughout the clip, I hear distinct chord progressions. The chords change at specific times, creating a sense of movement and progression. The song starts with no chord, then transitions to a D major chord, followed by an A major chord. This progression creates a sense of tension and release, adding to the dynamic nature of the song. The key of the song is A minor, which gives it a dark and melancholic quality. The minor key adds a sense of depth and emotional intensity to the music. Overall, the audio clip showcases a high-energy and rhythmic piece of music with a strong beat and melodic elements. The combination of drums, synthesizers, and percussion creates a vibrant and dynamic sound, while the chord progressions and key contribute to the emotional depth of the song. |

Table 11: Selected captions from MusicCaps, with LLARK responses, illustrating the failure mode of out-of-domain audio.

details). In a different form of popularity bias, the tempo predictions of 120BPM for the track in Table 10 (the true tempo is 139BPM) may also reflect popularity bias, as discussed above, from the original language model training data.

While the impact of popularity bias can be obvious in some cases (e.g., when tempo predictions are incorrect), in other cases its impact may be more subtle. While bias toward more commonly-observed categories and inputs may be expected or even preferred behavior for certain contexts, in other contexts, particularly for music, this form of bias can have unintended or harmful effects. We encourage researchers, developers, and users of these models to remain conscious of potential popularity bias in the outputs of musical models.

**Mitigation strategies:** Constructing or assembling more diverse training and evaluation datasets for large-scale music and language models would likely help to mitigate this form of bias, as would specific algorithmic interventions to encourage effective training on underrepresented or rare inputs or outputs. Furthermore, popularity bias may also reflect the greedy autoregressive strategy used to generate the models' output text; increasing the stochasticity inherent to the generation process or even guiding this process toward novel and diverse outputs would also likely reduce this bias.

### N.5 FAILURE CASE: CHATBOT BIAS

We identify a failure mode which we term "chatbot bias:" this reflects the tendency of the model to generate generally positive, vague, verbose descriptors which sound authoritative but can be semantically vacuous in context. Examples of these phrases from LLARK's outputs include:

- "The combination of these elements, along with the specific chord progression and downbeat pattern, gives the song its own unique identity within the electronic genre"
- "Throughout the clip, I notice the presence of chords that support the melodic lines. These chords add a sense of depth and complexity to the music, enhancing the emotional impact of the piece. The chords change subtly over time, creating a sense of tension and release that draws the listener in."
- "The combination of these styles and genres creates a unique and vibrant sound that is both energetic and catchy"

While it can be difficult to definitively state that these types of vague, flowery descriptions are always unhelpful or incorrect, this is part of the point of this failure mode: the model generates plausible, seemingly descriptive or helpful text, which is so vague or generic as to be meaningless outside of a small set of contexts.

We hypothesize that multiple factors drive this behavior. One likely cause is the RLHF process used to encourage the pretrained language model to be "helpful" to humans Touvron et al. (2023). It is likely that this process endows the model with a positive tone that pervades such samples ("gives the song its own unique identity", "creating a sense of tension and release that draws the listener in", "a unique and vibrant sound that is both energetic and catchy"). In particular, this reflects the emphasis on being "helpful" often used in RLHF training.

A second likely cause is the instruction-following training data. In particular, we see that the ChatGPT variants used to generate the instruction-following pairs displays a tendency toward positive, overly verbose, generic descriptors. As a result, the language model (already predisposed to such language via its RLHF tuning) is further fine-tuned toward "chatbot biased" language during LLARK training.

**Mitigation strategies:** Eliminating this form of bias is challenging. One reason is that such bias may not be undesirable in all cases – some users and applications may prefer this type of language, while others might require more precise, technical, and verifiable musical descriptions. However, we note that multiple steps could reduce this bias. These include: (1) not using RLHF-tuned models, when desired and when the instruction-following capabilities of such models are considered less advantageous than eliminating chatbot bias; (2) reducing the occurrence of such language in the training data, perhaps via better prompting at instruction-generation time; (3) reducing the occurrence at LLARK's inference time via prompting.

