# OpenReview forum: "LLark: A Multimodal Foundation Model for Music"
_ICLR.cc/2024/Conference — Submitted to ICLR 2024_

### Official Review · Reviewer_p6wE · 2023-10-31

**Soundness:** 2 fair
**Presentation:** 2 fair
**Contribution:** 2 fair
**Rating:** 5
**Confidence:** 5

**Summary:**

This paper introduces a language-based music understanding model designed to excel in various music comprehension and retrieval tasks. The model's architecture is constructed by harnessing the power of the jukebox audio encoder and the Llama 2 text encoder. To create the training data, ChatGPT (or GPT-4) is employed to generate instructional texts based on music track metadata, encompassing information such as BPM, genre, chord progression, and key signature. This comprehensive model offers thorough analysis and understanding of music inputs. The authors showcase the impressive capabilities of their model, named 'Llark,' by demonstrating its high performance across a wide range of music-related tasks, including music genre classification, key signature classification, tempo estimation, and music captioning. Furthermore, the authors have dedicated a substantial portion of the appendix to provide in-depth insights into their experimental results and the data generation process.

**Strengths:**

The proposed LLark model stands out as a tool for tackling a wide array of music understanding and retrieval tasks. Notably, the realm of cross-modality models, particularly those bridging language and music, has remained relatively under-explored. The majority of existing models have primarily concentrated on text-image and general audio-text associations, leaving a gap in the field. This paper marks an initialization in the direction of language-based music understanding, utilizing LLM. The selection of music-related tasks employed to evaluate the capabilities of the LLark model is logical and well-founded.

**Weaknesses:**

However, there are several notable shortcomings in this paper concerning its overall motivation, technical novelty, and experimental design, which diminish its quality and hinder its acceptance at the ICLR conference.

First, the assertion that LLark serves as a foundational music model is overstated. Upon closer examination, LLark emerges as a language-based audio understanding model, enriched with music instruction-based data. It lacks significant technical novelty for the following reasons:

(1) The audio encoder (Jukebox) and text encoder (Llama 2) employed in LLark were introduced in previous works, and their training primarily relied on large-scale data pre-training. Consequently, the paper doesn't place substantial emphasis on advancing the early-stage representation learning of both audio and language models.

(2) The model architecture itself, present in both encoders, exhibits no novel adaptations to better suit the domains of music understanding or language comprehension.

(3) The instruction-based data generation process, while employed effectively, has been previously explored in various machine learning domains, including audio, as demonstrated by works such as LAION-CLAP [1] and WavCaps [2]. The method used for prompt creation in this paper does not introduce particularly novel design elements, potentially falling short of the requisite technical novelty standards for ICLR.

Consequently, apart from LLark's exploration of LLM's potential applications in the context of music as a sub-category of the audio modality, both the audio and text encoders, as well as the data creation approach, lack substantial innovation. It appears to be more of a fusion of pre-existing works rather than a meticulously designed contribution to the field of music understanding.

Additionally, LLark's design, which is more of a combination of existing components than a novel creation, results in a limitation. It primarily excels at simple classification-based and language-generation tasks, thus falling short of being a comprehensive music foundation model capable of addressing diverse aspects of music content, such as music source separation and music generation. Consequently, labeling LLark as a music foundation model appears somewhat overambitious, as a language-output-only model does not meet the criteria for such a designation.

This limitation exposes a second weakness. If LLark were indeed a potent model capable of unifying various music understanding tasks, it would represent a significant breakthrough, offering a one-stop solution for music classification, understanding, and captioning. However, LLark does not meet these expectations. Based on my extensive knowledge of Music Information Retrieval (MIR) development, LLark lags behind state-of-the-art models in several MIR subtasks, and even falls short of other music understanding models, such as MERT [3]. Here are some pertinent statistics:

(1) In genre classification on GTZAN, the current state-of-the-art [4] achieves an accuracy of 83.5%, while MERT, a music understanding model without a language component, achieves around 79.3%. In contrast, LLark only achieves 56%, which falls significantly short of practical usability standards.

(2) For key-signature classification on GiantStep, the current state-of-the-art model [3] achieves 74.3% accuracy, whereas LLark attains 70%.

(3) In tempo estimation on GiantStep, the current state-of-the-art model [5] reaches an accuracy of 88.6%, while LLark achieves 86%.

Remarkably, LLark does not exhibit superior performance despite leveraging two large models, Jukebox-5b and Llama2-7B. It's worth noting that these state-of-the-art models often have parameter sizes of less than 100 million or even 1-5 million. MERT, a Hubert-based music understanding model, effectively achieves equal or superior performance with a parameter size ranging from 95 million to 330 million, employing straightforward linear modules. Furthermore, models like MERT, CLAP, and others can extend their capabilities to encompass broader applications, such as music separation and text-to-music generation tasks. Thus, introducing a language model with billions of parameters that delivers fewer tasks and less performance seems counterintuitive. Moreover, comparing LLark with language-audio (LU-AST) or language-image (IB-LLM) models is not comprehensive, as they are trained on different datasets and do not primarily focus on music data. Additionally, these models, in my view, are also far from achieving state-of-the-art performance to demonstrate practical usability

It is important to highlight that LLark's inference setting may not be as rigorously defined as proposed. The instruction-based data generation method provides the model with pre-structured slots for various music elements, such as genre, tempo, instrument, and chord. In this setup, it is expected that LLark can generate outputs corresponding to these slots when prompted with questions. Consequently, it may not truly qualify as a zero-shot setting.

Regarding music captioning, while LLark's performance in this aspect demonstrates the power of language models, it is arguable whether this achievement is solely attributed to LLark or whether it mainly leverages the adaptation learned within Llama 2. Furthermore, when it comes to evaluation metrics, such as BLEU scores, LLark does not consistently outperform previous state-of-the-art approaches.

Third, the motivation behind incorporating a language model within a music foundation model remains somewhat unclear. While LLM represents a remarkable advance in AI, the necessity of using a language model to answer questions about music attributes like tempo, key signature, chord estimation, or genre is a matter of debate. Such an interface may be more welcome when provided with labels or temporal markers, rather than a language-based query. Although music captioning demonstrates the potential, it is not fully explored in this paper. To harness the full potential of LLM in music understanding, it is crucial to either prove that language can serve as an effective tool or instruction for new music tasks (e.g., text-to-music generation, text-guided separation) or show that LLark can learn more from limited instruction-based data. However, this aspect remains unaddressed in the paper, as it is evident that LLark's performance relies heavily on specific instruction texts, and it may struggle to comprehend music without those explicit cues.

In summary, LLark appears to lack the necessary level of technical novelty in designing new architectures for a music foundation model, does not provide comprehensive evidence to establish itself as a complete music foundation model, and fails to demonstrate superior performance or conduct exhaustive comparisons against established Music Information Retrieval (MIR) state-of-the-art models or other music understanding approaches. As such, it may not meet the standards expected for acceptance at the ICLR conference, and more substantial evidence is required to demonstrate its functionality and efficiency

[1] Large-scale Contrastive Language-Audio Pretraining with Feature Fusion and Keyword-to-Caption Augmentation, ICASSP 2023

[2] WavCaps: A ChatGPT-Assisted Weakly-Labelled Audio Captioning Dataset for Audio-Language Multimodal Research, Arxiv

[3] MERT: Acoustic Music Understanding Model with Large-Scale Self-supervised Training, Arxiv

[4] Masked Modeling Duo: Learning Representations by Encouraging Both Networks to Model the Input, ICASSP 2023

[5] MAP-Music2Vec: A Simple and Effective Baseline for Self-Supervised Music Audio Representation Learning, ISMIR 2022

**Questions:**

1. Regarding CLAP, you mentioned in the appendix that it lacks sufficient music data and temporal-related embeddings to serve as the audio encoder. However, it's worth noting that their official GitHub repository provides a checkpoint trained on more than 630K data, including music, speech, and general audio. Have you explored this checkpoint? Additionally, there seems to be a reference error related to LAION-CLAP; it should refer to [1] instead of [2].

2. Do you have any experimental results for LLark on tasks related to temporal aspects, such as chord estimation or beat tracking? Including such results could enhance the paper's overall demonstration of LLark's performance and its ability to adapt to a wider range of Music Information Retrieval (MIR) tasks.

3. On the demo page, there appears to be an issue with the music captioning demo. Specifically, I observed that Beethoven's captioning has three identical captions. Is this a posting error that needs to be addressed?

[1] Large-scale Contrastive Language-Audio Pretraining with Feature Fusion and Keyword-to-Caption Augmentation, ICASSP 2023

[2] CLAP: Learning Audio Concepts From Natural Language Supervision, ICASSP 2023

**Details Of Ethics Concerns:**

Not Applied

---

> ### Author Response · Authors · 2023-11-16
> **Reviewer p6wE response (1/4)**
>
> Thank you to the reviewer for their detailed, thorough, and expert response. We discuss the main themes below. Then, we answer the reviewers’ questions individually. We would like to acknowledge upfront that the reviewers’ highly constructive responses have been incorporated to the paper in many places, and that they have overall improved the presentation of results and contextualized our work better in comparison to existing models, including the representation-learning methods the reviewer highlighted. The paper has benefited significantly from the reviewers’ expertise in MIR/audio modeling and benchmarking - thank you!
>
> * **Lack of novelty (numbered according to reviewers’ original numbering):** (1, 2) The reviewer is correct that LLark is built upon existing modality-specific models, and uses instruction-following techniques that have worked well for vision and language. As we discuss in our response to reviewer gHco above, this is both a benefit to our approach (not requiring expensive retraining of language or audio models) and in line with existing multimodal contributions in other domains [1-6]. The use of a straightforward architecture with “no novel adaptations to better suit the domains'' is also in line with recent multimodal models such as [1, 6, 7] which use simple projection layers as multimodal adapters. (3) Instruction tuning is neither used in LAION-CLAP (which uses keyword-to-caption augmentation to train a contrastive model) nor in WavCaps (which uses an LLM for pseudolabeling captioning data). However, we do not claim to be the first to use instruction-tuning, even for music. Instead, our contribution is in line with data-centric machine learning, where our results are distinguished by their scale (1.2M musical query-response pairs), diversity (over 160k tracks), and quality (using high-quality GPT models) to achieve results not demonstrated by any existing multimodal model for music. We believe that this is in line with similar works in the audio space appearing in ICLR; for example, MT3 [8, ICLR’22], also used preexisting, unmodified architecture and datasets.
>
> * **Status as foundation model:** The term “foundation model” is nebulous; we apologize to the reviewer for any confusion and think they raise a fair objection. We are using the term in the sense of the Stanford Center for Research on Foundation Models (CRFM), which states on its website (https://crfm.stanford.edu/) that “In recent years, a new successful paradigm for building AI systems has emerged: Train one model on a huge amount of data and adapt it to many applications. We call such a model a foundation model.” While we believe LLark meets this definition, we have removed the term “foundation model” from the title and have updated the paper text to avoid characterizing LLark as a “foundation model” (we are not currently able to change the OpenReview submission title, but have changed the title in the PDF).
> * **Performance vs. state-of-the-art (SOTA) on MIR tasks:**
>   * **Discussion of task-specific vs. generalist models:** The reviewer correctly notes that LLark does not achieve SOTA on the MIR tasks in the paper. We have added discussion to the paper highlighting the gap between LLark and the SOTA models mentioned (Section E) and have included a more explicit discussion of representation learning models in the related work (Section D). We specifically added the SOTA metrics the reviewer highlights for tempo, key, and genre. However, not beating SOTA relative to task-specific models is in line with other generalist models for music. This includes MERT [9], which only achieves SOTA performance on 3 of the 23 tasks evaluated in the paper (Tables 1 and 2 of [9]; these tables show that task-specific models still hold the SOTA across 16 of 23 tasks), and even then MERT’s performance is only *after* linear probing directly on the target task (see below). Like the reviewer, we agree that MERT is still a useful scientific contribution due to its ability to achieve good performance on many challenging tasks with a unified model -- and we believe that LLark makes similar gains (for example, LLark excels on captioning tasks not addressed by any representation learning model). Similar to other generalist models in both the audio- and vision-language domains, our objective was not to achieve SOTA performance on any single task, but instead to build a model which (i) achieves strong performance on many tasks, including unseen tasks (as opposed to the task-specific models the reviewer cites, which will almost always outperform any generalist model) and (ii) outperforms existing audio-language models. We believe our evaluations, including in Section 6, demonstrate that these criteria are met. In fact, the numbers the reviewer provides also highlight how close LLark is to SOTA performance in both the key and tempo tasks, despite being capable of performing *both* tasks at this level (along with many other tasks).

---

> ### Author Response · Authors · 2023-11-16
> **Reviewer p6wE response (2/4)**
>
> * Performance vs. state-of-the-art (SOTA) on MIR tasks [cont]:
>   * **Clarification regarding linear probing:** We believe it is important to note that some of the models the reviewer highlights as SOTA, including MERT and the model of [23], achieve the reported performance *only after fine-tuning a probe directly on each target task* (this procedure is described in [9], section B.2, and [23], Fig. 1). Thus, while these are useful to provide an upper bound on the best possible performance for the tasks, these represent the performance by a model explicitly trained on those datasets, not generalist models never exposed to the target task or data. Thus, we believe the best comparison for LLark is other audio-text models which can generalize directly to new tasks, without fine-tuning on task-specific data.
>   * **Discussion of genre on GTZAN:** The reviewer highlights the GTZAN dataset. Since the reviewer is undoubtedly familiar with the literature, we expect the reviewer also understands that, while it is used widely, GTZAN is also a problematic dataset that can be unreliable, in isolation, for evaluating deep music understanding. The issues with this dataset have been widely discussed in the literature, as we are sure the reviewer is aware (for example, it assigns only a single genre to each song; genres are assigned by artist, not by track; the 10 genres selected are arbitrary and in fact often different levels of the same branch of a genre hierarchy, such as “metal” and “rock” or “disco” and “pop”). We believe that the detailed results on GTZAN present in the supplement (e.g. Figure 10) show that LLark’s responses on this dataset are in fact reasonable, and in line with responses humans would make (for example, LLark most often labels “metal” songs as “rock” -- an entirely reasonable response for a model that is maximizing the probability of its outputs). We note that in the second genre dataset (MedleyDB), the gap between LLark and the top-performing model (IB-LLM) is less than 1pp, which reflects the uniqueness of GTZAN and the need to conduct a holistic evaluation without overly emphasizing a single data point, particularly for ill-defined tasks such as genre classification.
>   * **Question about tempo estimation SOTA:** The reviewer suggests that [20] is SOTA on Giant Steps tempo estimation. However, we could not find tempo reported in this paper (https://archives.ismir.net/ismir2022/latebreaking/000049.pdf). Could the reviewer please confirm the source of this number? Otherwise, it appears to us that the current best SOTA on Giant Steps tempo is the model of [21], reported in [22] as Acc2 of 92.5.
> * **Comparison to representation learning models (MERT and CLAP):** The reviewer states that “Furthermore, models like MERT, CLAP, and others can extend their capabilities to encompass broader applications, such as music separation and text-to-music generation tasks.” Both MERT and CLAP are music representation-learning models. These models do not directly make predictions for any downstream task; they are either used in a zero-shot fashion by comparing embedding distances between an audio input and a text label (as in CLAP [10], see Fig. 1, and LAION CLAP [11], Fig. 1 middle) or via linear probing (as in MERT [9], sec. B.2, and LIAON CLAP [11], Fig. 1 bottom). LLark is capable of directly outputting text responses for any query or task, and this is because it addresses a fundamentally different task: generating language from music + instructions. However, just as the reviewer suggests that the representations learned by MERT and CLAP could be used for source separation or music generation, the same task could be performed with LLark. For example, LLark’s hidden states could be used as general-purpose embeddings, and LLark could be extended to output discrete acoustic tokens to generate music. In this sense, the model we propose is capable of the same hypothetical extensions as those described for MERT and CLAP, but unlike MERT and CLAP, it is also capable of performing classification, regression, and text generation tasks out of the box without zero-shot adaptation or probing.

---

> > ### Author Response · Authors · 2023-11-16
> > **Reviewer p6wE response (3/4)**
> >
> > * **Comparison to other (smaller) models:** We acknowledge that LLark is larger than models such as MERT. However, we are not aware of evidence that supports the reviewer’s claim that MERT “effectively achieves equal or superior performance”. As mentioned above, MERT’s performance is only evaluated in [9] after probing, which directly fine-tunes a new output MLP layer for MERT on each target task; MERT alone is not capable of directly outputting class labels for any task without this task-specific probing step. We did not explore fine-tuning LLark on individual datasets or using linear probing on its output hidden states, but believe that only a comparison with probing would be fair to LLark. The reviewer raises an interesting question: how could we decrease LLark’s parameter count? Our ablation study provides some (preliminary) evidence to address this. In particular, it suggests (Figure 6) that the language model could be decreased in size while preserving performance on most MIR tasks (except tempo). We have added a more explicit discussion of model size considerations in this section. Furthermore, since we only use the output of the 36th layer of Jukebox, our model does not use approximately half of the decoder -- which means that the effective parameter count of Jukebox in our application is around 50% lower.
> > * **Inference setting:** We agree with the reviewer that our description of “zero-shot datasets” is confusing; we use the term “zero-shot dataset” to refer to datasets unseen at training time. We have updated the text for clarity.
> > * **Music captioning:** The reviewer states “it is arguable whether this achievement is solely attributed to LLark or whether it mainly leverages the adaptation learned within Llama 2” -- it is a combination of both. Certainly, Llama 2 already contains very strong text generation capabilities. However, Llama has no multimodal capabilities on its own. The power of a multimodal model is in aligning the LM’s generation to the musical inputs so that the generated text corresponds to the musical inputs, not just the instruction text; we believe our captioning results demonstrate that this has largely been achieved. While we provide the linguistic metrics (BLEU,  ROUGE, etc.) in the supplement, that these are unreliable indicators of quality as they only measure adherence to a reference caption, while many captions are possible for a given piece of audio. As a result, we believe that human evaluation of captioning is the best indicator of model quality; as we note, this is known as the “gold standard” of chatbot evaluation in part for this reason [14].
> > * **Motivation for use of a language model for music:** We share the reviewer’s thoughtful consideration of whether a language model (LM) is the “right” approach for some music tasks. We believe the reviewer would agree that a LM is certainly appropriate for some tasks, such as captioning and certain reasoning tasks. Their main concern seems to be for certain MIR and audio tasks. Here, we feel that the main value of a LM is the useful fact that many tasks can be formulated in natural language (as we demonstrate with e.g. key, tempo). This flexibility is what makes LMs so useful as general models, and is our motivation for using an instruction-following framework. This flexibility relieves us of having to adapt the model to individual tasks (as is required for a probing step for CLAP or MERT) unless further fine-tuning is desired.
> >
> >   The reviewer states that “To harness the full potential of LLM in music understanding,  it is crucial to either prove that language can serve as an effective tool or instruction for new music tasks (e.g., text-to-music generation, text-guided separation) or show that LLark can learn more from limited instruction-based data”. We believe that the audio tasks mentioned, especially text-to-music generation, have already been demonstrated widely with LLMs [15-18]. In the second case (limited-data learning), our results in Figure 8 may provide some relevant data: in particular, we show that, indeed, LLark can be trained on smaller datasets, with only limited performance changes on the MIR tasks even when using 50% of the original data. This provides at least some evidence that, indeed, LLark can learn more from limited instruction-based data, and is in line with existing works [19] which show that quality and diversity are far more important than scale for instruction-tuning.

---

> > > ### Author Response · Authors · 2023-11-16
> > > **Reviewer p6wE response (4/4)**
> > >
> > > **Responses to questions:**
> > > 1. The reviewer is correct, we use LAION CLAP; the Microsoft CLAP was incorrectly referenced in a few places in the paper. (That is now corrected.) We use the checkpoint recommended by the authors specifically for music available at https://github.com/LAION-AI/CLAP  .
> > > 2. We do not evaluate chord or beat tracking, but agree that this would be a useful future direction for this work.
> > > 3. We will address this issue in the final (non-anonymized) version of the website. Thank you for flagging it.
> > >
> > > Once again, we are grateful for the reviewer’s thoughtful and expert responses and hope that we can continue to address any lingering concerns through constructive dialogue during the author-reviewer response window.
> > >
> > > **References**
> > > * [1] Liu, Haotian, et al. "Visual instruction tuning." arXiv preprint arXiv:2304.08485 (2023).
> > > * [2] Li, Bo, et al. "Otter: A multi-modal model with in-context instruction tuning." arXiv preprint arXiv:2305.03726 (2023).
> > > * [3] Awadalla, Anas, et al. "Openflamingo: An open-source framework for training large autoregressive vision-language models." arXiv preprint arXiv:2308.01390 (2023).
> > > * [4] Gao, Peng, et al. "Llama-adapter v2: Parameter-efficient visual instruction model." arXiv preprint arXiv:2304.15010 (2023).
> > > * [5] Han, Jiaming, et al. "Imagebind-llm: Multi-modality instruction tuning." arXiv preprint arXiv:2309.03905 (2023).
> > > * [6] Zhu, Deyao, et al. "Minigpt-4: Enhancing vision-language understanding with advanced large language models." arXiv preprint arXiv:2304.10592 (2023).
> > > * [7] Rohan Bavishi et al. Fuyu-8B: A Multimodal Architecture for AI Agents. https://www.adept.ai/blog/fuyu-8b
> > > * [8] Gardner, Josh, et al. "MT3: Multi-task multitrack music transcription." ICLR 2022.
> > > * [9] Li, Yizhi, et al. "MERT: Acoustic Music Understanding Model with Large-Scale Self-supervised Training." arXiv preprint arXiv:2306.00107 (2023).
> > > * [10] Elizalde, Benjamin, et al. "Clap learning audio concepts from natural language supervision." ICASSP 2023-2023 IEEE International Conference on Acoustics, Speech and Signal Processing (ICASSP). IEEE, 2023.
> > > * [11] MAP-Music2Vec: A Simple and Effective Baseline for Self-Supervised Music Audio Representation Learning, ISMIR 2022.
> > > * [12] Korzeniowski, Filip, and Gerhard Widmer. "End-to-end musical key estimation using a convolutional neural network." 2017 25th European Signal Processing Conference (EUSIPCO). IEEE, 2017.
> > > * [13] Wu, Yusong, et al. "Large-scale contrastive language-audio pretraining with feature fusion and keyword-to-caption augmentation." ICASSP 2023-2023 IEEE International Conference on Acoustics, Speech and Signal Processing (ICASSP). IEEE, 2023.
> > > * [14] Touvron, Hugo, et al. "Llama 2: Open foundation and fine-tuned chat models." arXiv preprint arXiv:2307.09288 (2023).
> > > * [15] Agostinelli, Andrea, et al. "Musiclm: Generating music from text." arXiv preprint arXiv:2301.11325 (2023).
> > > * [16] Copet, Jade, et al. "Simple and Controllable Music Generation." arXiv preprint arXiv:2306.05284 (2023).
> > > * [17] Schneider, Flavio, Zhijing Jin, and Bernhard Schölkopf. "Mo\^ usai: Text-to-Music Generation with Long-Context Latent Diffusion." arXiv preprint arXiv:2301.11757 (2023).
> > > * [18] Li, Peike, et al. "Jen-1: Text-guided universal music generation with omnidirectional diffusion models." arXiv preprint arXiv:2308.04729 (2023).
> > > * [19] Zhou, Chunting, et al. "Lima: Less is more for alignment." arXiv preprint arXiv:2305.11206 (2023).
> > > * [20] MAP-Music2Vec: A Simple and Effective Baseline for Self-Supervised Music Audio Representation Learning, ISMIR 2022
> > > * [21] Schreiber, Hendrik, and Meinard Müller. "Musical tempo and key estimation using convolutional neural networks with directional filters." arXiv preprint arXiv:1903.10839 (2019).
> > > * [22] de Souza, Mila Soares de Oliveira, Pedro Nuno de Souza Moura, and Jean-Pierre Briot. "Music Tempo Estimation via Neural Networks--A Comparative Analysis." arXiv preprint arXiv:2107.09208 (2021).
> > > * [23] McCallum, Matthew C., et al. "Supervised and unsupervised learning of audio representations for music understanding." arXiv preprint arXiv:2210.03799 (2022).

---

> > > > ### Author Response · Authors · 2023-11-18
> > > > **Request to review author response**
> > > >
> > > > Dear reviewer p6wE,
> > > >
> > > > Thank you for your time and effort spent reviewing our paper. We have submitted our response to your concerns, including a revised version of our paper (updates are in red text). Please let us know your comments, whether our revisions and clarifications have addressed your concerns, or whether we can provide further clarification. Thank you again!

---

> ### Author Response · Authors · 2023-11-21
>
> Once again, we thank the reviewer for your time reviewing the paper. As the author-reviewer phase is ending soon, we request the reviewer to please review the author response and let us know whether the comments and revisions to the paper (indicated in red text) have addressed your concerns, or whether we can provide any further clarification before the discussion window closes. Thank you!

---

> > ### Comment · Reviewer_p6wE · 2023-11-22
> > **Response to the Authors' Rebuttal**
> >
> > Thank you for providing around 4 pages of responses that addressing a lot of my questions and comments in the review.
> >
> > First of all, I want to greatly express my appreciation on the authors because of their throughout and correct response, showing that my review is correctly and carefully being read and discussed. After carefully reading the response, I am glad that we come into agreements on some questions I proposed and the authors' revision meets my expectation. In all, I would like to raise my score to 5 "marginally below the acceptance". I gave my reasons below.
> >
> > The changes in the title is very imperative in my view, from "foundation model" to the "multi-modal model" shapes the whole paper's focus and target, aligning it more similarly with MERT and CLAP. I appreciate the authors' reference for the foundation model definition by Stanford but it seems like we both agree that LLark meets the "foundation model" definition but does not meet the true or ideal "foundation model" target as we both hope a foundation model can not only address the classification and understanding tasks but the generation and content-extraction tasks (e.g., music generation and separation).  If LLark is reshaped as the multi-modal model, the extendability can be reconsidered because you can add more modules at it end to support generation, separation and more tasks and I believe LLark has the capability to do that. Here, I welcome the experiments on text-to-music generation and source separation by using LLark instead of CLAP and MERT as presented in many current works. But this is practical at this stage and definitely I'm not requiring the authors to present these results.
> >
> > However, if the LLark is reshaped as the multi-modal model, I think the novelty of the paper should be reconsidered and this is the reason for why I cannot give the 6 or above scores. I think the current architecture of LLark contains an audio encoder and a text encoder, with musical instruction-tunning training paradigm. The difference I could identify to CLAP is that it uses different encoders because of the performance and design reasons. This might not be meet the expectation of ICLR, because LLark is not the first paper trying to align audio and text (CLAP) or music and text (Mulan).
> >
> > For the benchmark verification, I apologize that I cited a wrong refer for GS-tempo estimation. This is the correct one (with 88.7): https://arxiv.org/pdf/1903.10839.pdf. However, I think the authors correct one of my opinions that I may be to strict to request the alignment of the performance in different tasks on a multi-modal model, since its benefit its the flexibility and generalization to all tasks. So, I agree with the authors' response and I completely credit the LLark's performance in achieving not top-1 but good results in many music downstream tasks, and also leaving good space for finetuning.
> >
> > In conclusion, I am very satisfied with the authors' response. But the only concern is can such a revision on the paper still be regarded as the initial submission as it might reshape the novelty of this paper. Here I would like to discuss more on it with meta-reviewer and other reviewers and I am willing to raise my score again.
> >
> > Thank you very much. And I admit that, after the revision, this paper meets my expectation.

---

> > > ### Author Response · Authors · 2023-11-22
> > > **Follow up discussion with Reviewer p6wE (1/2)**
> > >
> > > Dear Reviewer p6wE,
> > >
> > > Thank you for your thoughtful consideration of our response. We are glad to hear that our revisions and responses have improved the reviewers' consideration of our work, and that they are "very satisfied" with our response and that it now "meets [their] expectations". Again, we feel that the reviewers' expert and thorough initial review helped improve the work considerably, and are grateful for the reviewers' efforts.
> > >
> > > The SOTA GS-tempo estimation model mentioned by the reviewer is cited in our revision as the current SOTA in Section E.1.2 (under "Task-Specific SOTA Baseline"). The reviewers' expertise in MIR tasks has improved the paper's comprehensiveness substantially.
> > >
> > > **Evaluating revisions vs. initial submission:** We acknowledge the reviewers' desire to discuss with other reviewers and meta-reviewers whether the revisions to the paper can "still be regarded as the initial submission" as the reviewer mentions. We would like to point out that this is the stated aim of the author-reviewer response window. We note that Step 8 of the ICLR Reviewer Guide (https://iclr.cc/Conferences/2024/ReviewerGuide) is:
> > > > "Provide final recommendation: Update your review, taking into account the new information collected during the discussion phase, and any revisions to the submission. State your reasoning and what did/didn’t change your recommendation throughout the discussion phase."
> > >
> > > The reviewer is empowered (and, indeed, instructed) to assess the quality and signficance of our final, revised submission in their final rating, and we greatly appreciate their efforts to do so!
> > >
> > > **Novelty of model:** After increasing their score, the reviewer states that novelty of the paper is the main additional reason why they do not rate the paper a 6 or higher. We understand this concern (which is also discussed in our response to Reviewer zBmL above). We would like to point out two potential clarifications here as the reviewer continues to consider the work:
> > >
> > >   * **Comparison to contrastive audio-text embedding models (CLAP and MULAN)**: The reviewer states that "This might not meet the expectation of ICLR, because LLark is not the first paper trying to align audio and text (CLAP) or music and text (Mulan)." However, both CLAP and MULAN are audio-text embedding models (with MULAN trained specifically for music). Neither model is capable of *generating* text, or of adapting its representations conditional on a set of user-provided instructions. This is the main difference between our work and these contrastive models. LLark can *generate* text conditional on audio+text, and can do so in a way that performs a *wide variety of downstream tasks with zero fine-tuning, probing, or zero-shot adaptation strategies required* (we discuss why probing/zero-shot adaptation are necessary for CLAP in our previous response; this analysis also applies to MULAN). We are glad to see the reviewer also recognizes this, stating regarding LLark: "its benefit [is] the flexibility and generalization to all tasks" (in contrast to embedding methods, which cannot generate text, and which require probing or zero-shot adaptation strategies to perform downstream tasks). We believe this constitutes a major difference between LLark and CLAP, MULAN, or other models such as MERT, which are still useful for many applications but differ greatly in their functionality.

---

> ### Author Response · Authors · 2023-11-22
> **Follow up discussion with Reviewer p6wE (2/2)**
>
> [continued from previous point]
>   * **Additional considerations beyond architectural novelty:** We believe that our contribution extends beyond purely proposing a novel model (since other audio-language models do exist, as the reviewer and our paper both note, and as discussed in our point "Lack of novelty" in the original Reviewer p6wE response (1/4)). This is why we have emphasized our contribution as one of data-centric machine learning and encompassing data, models, and evaluation. In addition to a novel model (which is *not* what we consider our standalone contribution), we believe that the training data generation strategy, strong empirical results, and the quality and rigor of evaluation are important to our work. To our knowledge, LLark contains the most thorough set of musical evaluations of any audio-text model, where we evaluate MIR tasks (tempo estimation, key estimation, genre classification, instrument identification), conduct high-quality human evaluations for both music captioning and reasoning tasks, and also provide a set of objective (non-human evaluated) metrics across these tasks. Our ablation studies and extensive supplementary results provide further insights related to scaling and model components, as we discuss above and in the paper. We believe these evaluations and empirical insights are also critical contributions of our work, as rigorous evaluation is also critical to reliable progress in the field of machine learning [1]. As we noted in our previous response, other music-related works have also appeared in the ICLR conference without proposing novel model architectures, such as MT3 (ICLR 2021), which uses "an off-the-shelf T5 architecture" (see p.2 of [2]).
>
> Once again, we thank the reviewer for their thoughtful engagement. Please let us know if we are able to provide any additional clarifications or further improve the paper to reflect the reviewers' concerns during the author-reviewer discussion window.
>
> References:
>
> [1] Liao, Thomas, et al. "Are we learning yet? A meta review of evaluation failures across machine learning." NeurIPS 2021.
>
> [2] Gardner, Josh, Ian Simon, Ethan Manilow, Curtis Hawthorne, and Jesse Engel. "MT3: Multi-task multitrack music transcription." ICLR 2022.

---

### Official Review · Reviewer_JrBS · 2023-11-01

**Soundness:** 3 good
**Presentation:** 4 excellent
**Contribution:** 3 good
**Rating:** 6
**Confidence:** 4

**Summary:**

This paper collects several common MIR tasks (captioning, tagging, tempo estimation, etc) under a common text-based formulation that allows them to be trained and evaluated via instruction-tuning a single LLM with audio conditioning. It presents a new model called LLark along with a number of experimental evaluations.

**Strengths:**

- modelling improvements on several MIR tasks
- presents a clear and compelling path for future MIR work in this direction
- comprehensive details on the data processing choices and implementation provided in the supplements
- technical details are clearly communicated with an emphasis on making things easy to reproduce

**Weaknesses:**

- Some of the most impressive results in the paper are regarding the long-form text outputs - but this is also where the non-expert evaluations are limited, and where the negative side effects of LLM’s come into play. I think that lumping these together under the term “model hallucinations” loses important nuance about how this will play out when LLark gets deployed into products, etc. When I look at the outputs in the online supplement, I see really impressive text with lots of musical terms thrown in that are related to the track - but I think more focused evaluations with experts would immediately reveal that much of this captioning output is nonsense - and isn’t it creating a problem to have such nicely packaged up nonsense that to non-experts it looks right? That sense of authority and confusion is less of a problem with the pre-LLM baseline models. I understand that this kind of evaluation may be out of scope for this paper, and there’s room for critical follow-up work once the model is made open-source - But I can examples of this happening immediately in the online supplement, and I think it’s worth at least highlighting these kinds of things in the paper.  For example:

 -   under “How could a music producer recreate the sounds in this track? —> they would likely use a variety of synthesizers, both digital and analog” - that’s typical LLM word-salad stuff that in some cases seems harmless, but it’s super misleading! You definitely don’t need to go out and buy a bunch of analog synthesizers to produce that kind of track.

- “What are some characteristics that potentially differentiate the song from other similar songs —> the use of the synthesizer as the main instrument…Additionally, the song's tempo of 120 BPM and the use of the E minor key… gives the song its own unique identity within the electronic genre” This is all nonsense - pretty much every song in the electronic genre features the synthesiser, 120 bpm is the least distinctive tempo you can choose, and E minor is a very typical key!

**Questions:**

Given the time to do it, how would you design an evaluation to address the kinds of issues raised under "weaknesses" above? How much of that can be communicated within this paper rather than left to future work?

---

> ### Author Response · Authors · 2023-11-16
> **Reviewer JrBS response (1/2)**
>
> Thank you to the reviewer for their thorough analysis of the paper and even the supplementary material. We are pleased that the reviewer acknowledged the modeling improvements in the work, found the data and technical details “comprehensive” and “clearly communicated,” and felt the paper “presents a clear and compelling path for future MIR work in this direction”. We discuss the reviewers’ main concerns below, but would like to highlight up front that we have made several changes and additions to the paper in order to reflect the reviewers’ concerns, and we believe they have improved the paper considerably. These include: a new supplementary section on “Failure Cases” (Section N), a Model Card (Section M) to highlight limitations and concerns about potential downstream use, and updates to the Ethics statement (Section A). We are adding content to the online supplement to highlight the behaviors noted by the reviewer and discussed in these sections as well.
>
> We interpret one of the reviewers’ main concerns to be a need for more clearly highlighting and understanding the ways in which our model’s outputs can be misleading or incorrect. As the reviewer notes, LLMs have a tendency to generate “impressive text” with a “sense of authority” and this can be confusing or downright detrimental in some use cases when the model is incorrect. The persuasiveness of this kind of text only grows as the models’ language modeling capabilities improve. And while we made careful efforts to evaluate the music understanding capabilities of the model (e.g. Section 6.1), even long-form text responses are only helpful insofar as they are also correct, whether users know it or not. We share the reviewers’ concern, and both assessing and handling the (un)reliability of outputs is widely acknowledged to be an open research question [1] and a  major challenge for any language mode. This is made more challenging by the fact that there is no current methodology for language models to convey their uncertainty in the way that a human expert would; instead, autoregressive text generation models simply output the sequence of tokens which maximize the estimated probability conditional on the inputs -- regardless of whether the models’ estimates of these probabilities is reliable, or even whether the predictions themselves are confident.
>
> We share this concern, and have made several revisions to both clearly highlight potential issues with LLark’s outputs, make clear potential (mis)uses of the model, and clearly highlight risks and limitations.
>
> In particular, we have made the following changes:
> * The reviewer identifies a subset of problematic examples from the online supplement. In order to better highlight and categorize such behaviors across all tasks and datasets, we have introduced a new “Failure Cases” section to the paper (Section N), motivated by similar efforts in recent multimodal model releases (e.g. [2]). We have explicitly analyze some of the examples the reviewer mentions, among many others, in this new section, and we identify and discuss five important failure modes of the LLark model.
> * We have added a Model Card (Section M; we discuss this in detail below).
> * We have added a clear statement regarding the risks and limitations of using language models in this context to the ethics statement of the paper in (Section A).
>
> **Response to question:** The reviewer asks *“Given the time to do it, how would you design an evaluation to address the kinds of issues raised under "weaknesses" above? How much of that can be communicated within this paper rather than left to future work?”*
>
> The reviewer raises an important question. We offer several thoughts:
>
> * (1) One approach we believe is clearly motivated here: to clearly identify and highlight failure cases of the model. This has been used in other multimodal works, for example [2], and is in line with a broader movement toward investigating and publicizing the undesirable outputs of models (“red teaming” [3] is a version of this for safety-critical applications, where the model is explicitly prompted to perform dangerous tasks). We have added this to the paper in a new “Failure Cases” section (Section N), and we discuss five prominent failure cases (N.1-N.5), including examples, potential causes, and offer potential solutions to address each.
> * (2) For each failure mode, we propose specific mitigation strategies in Section N. We believe that each of these strategies, if implemented, could significantly reduce these failure modes, and each could be verified with a small set of targeted evaluations specific to the failure mode.

---

> > ### Author Response · Authors · 2023-11-16
> > **Reviewer JrBS response (2/2)**
> >
> > * (3) The community has existing tools to highlight the relevant attributes of models and their intended uses in order to mitigate the potential for adverse impact on users. One key tool is Model Cards [4]. We have added a Model Card to the paper (Section M), which we believe will also help to avoid the kinds of issues raised under the reviewers’ “weaknesses” concerns, and provide as a single guide to where critical information can be found throughout the paper.
> > * (4) “Fact checking” of the free-text responses to captioning and reasoning tasks could be a useful evaluation approach. This would entail identifying musical “facts” stated by the model (i.e. references to a specific key, tempo, or instrument) in its response, and determining whether these are correct, given the audio. This strategy could provide quantifiable evidence to either confirm or disconfirm a hypothesis of whether these errors are prevalent in free-text outputs. However, properly conducting this procedure is a subtle and complex undertaking and beyond the scope of the reviewer response window.
> > * (5) An extreme, but plausible, option could be to rely solely on expert reviewers. However, this is expensive, not scalable, and -- we believe -- not necessary to address some of the most significant concerns the reviewer raises. (This is also an option that would be left to future work, due to the time and cost of designing, recruiting for, and conducting a study with expert musicians.)
> >
> > Again, we are grateful to the reviewer for their thorough assessment and valuable feedback which have improved the paper. Please notify us if there are further concerns the reviewer has that we may address during the author response window, and we are grateful for the reviewers’ thorough efforts!
> >
> > References
> > * [1] Touvron, Hugo, et al. "Llama 2: Open foundation and fine-tuned chat models." arXiv preprint arXiv:2307.09288 (2023).
> > * [2] Han, Jiaming, et al. "Imagebind-llm: Multi-modality instruction tuning." arXiv preprint arXiv:2309.03905 (2023).
> > * [3] Ganguli, Deep, et al. "Red teaming language models to reduce harms: Methods, scaling behaviors, and lessons learned." arXiv preprint arXiv:2209.07858 (2022).
> > * [4] Mitchell, Margaret, et al. "Model cards for model reporting." Proceedings of the conference on fairness, accountability, and transparency. 2019.

---

> > > ### Author Response · Authors · 2023-11-18
> > > **Request to review author response**
> > >
> > > Dear reviewer JrBS,
> > >
> > > Thank you for your time and effort spent reviewing our paper. We have submitted our response to your concerns, including a revised version of our paper (updates are in red text). Please let us know your comments, whether our revisions and clarifications have addressed your concerns, or whether we can provide further clarification. Thank you again!

---

> > > ### Comment · Reviewer_JrBS · 2023-12-01
> > > **Official Comment from Reviewer JrBS**
> > >
> > > Thanks for your effort to update the appendix in response to my review. I've read the updated paper. I think Appendix N is helpful in improving the transparency around the limitations of the model, though I'm not sure how many readers will read all the way to Appendix N!

---

> ### Author Response · Authors · 2023-11-21
>
> Once again, we thank the reviewer for your time reviewing the paper. As the author-reviewer phase is ending soon, we request the reviewer to please review the author response and let us know whether the comments and revisions to the paper (indicated in red text) have addressed your concerns, or whether we can provide any further clarification before the discussion window closes. Thank you!

---

### Official Review · Reviewer_zBmL · 2023-11-02

**Soundness:** 3 good
**Presentation:** 3 good
**Contribution:** 4 excellent
**Rating:** 6
**Confidence:** 4

**Summary:**

The paper presents LLark, a multimodal foundation model for music understanding. It's trained using instruction-based tuning and metadata augmentation on open-source music datasets. LLark combines a pretrained audio encoder with a large language model, effectively merging language capabilities with traditional music information retrieval models for improved music understanding. LLark excels in various music tasks, showing impressive generalization capabilities. Human evaluators also show strong agreement with LLark's performance in captioning and reasoning tasks.

**Strengths:**

1.	This work blends the knowledge from audio understanding models with a touch of ChatGPT for LLaMA, addressing high annotation costs in the music domain by leveraging language models and pretrained MIR models.

2.	The model attains impressive results, particularly in reasoning tasks, showcasing its potential for practical applications.

3.	The incorporation of instruction tuning and metadata augmentation provides reusable insights within the realm of music information retrieval.

4.	The paper leverages publicly accessible, open-source, and permissively licensed music datasets for training, which ensures data accessibility and reproducibility.

**Weaknesses:**

1.	Section 1 mentions LLark's architecture as a contribution, but the integration of large language models (LLM) and domain-specific models has been previously explored in the literature [1-4], as properly cited in the paper. To strengthen the paper, consider providing a more detailed explanation of what sets LLark's architectural design apart from existing implementations.

2.	Limiting the audio encoder to 25-second clips (mentioned in Sections 4.1 and 7) is a notable drawback, as it lacks the understanding of important contextual information in real-world scenarios (most music audio is longer than 25 seconds), leading to potential misinterpretations and missed insights in audio content analysis.

3.	Section 4.2.1 should investigate how LLark's performance is affected by the features extracted from traditional MIR models [5], especially regarding temporal-related music properties like timestamped chords and beat grids. Additional ablation studies should be included to confirm the performance gain of metadata augmentation.

4.	In Sections 6.3 and 6.4, it’s ok to rely on human evaluation as we don't have holistic objective metrics to assess captioning and reasoning tasks. But leaning entirely on non-experts for this might not be the best call. It might be better to balance this out with some expert raters rather than going all-in on non-experts.

5.	In Figure 4, the matching rates are quite low across all models (around the random guessing level), leading to questions about whether LLark's performance on certain tasks is only marginally better than random guessing. Section 6.4 attributes this result to “limitations in the musical expertise of the (non-expert) raters in our study," further emphasizing my previous concerns about relying solely on non-experts for human evaluation.

6.	With no clear indication of randomness in option order (or maybe I missed?), such as whether LLark is consistently presented as the first option, this raises concerns about potential biases [6] in the evaluation results of GPT-4 in Sections 6.3 and 6.4.

7.	Wrong citation in Appendix F: the ablation study used the CLAP developed by LAION [7], but wrongly cited the Microsoft one [8]. This made me confused for a while. Please correct the citation.

8.	In Appendix F, ablation studies may not offer a fair comparison due to the significant model size differences between CLAP [7] and Jukebox [9]. To assess temporal information's significance, please consider an additional ablation study where Jukebox's weights are still frozen but average pooling is applied to its output embeddings without preserving temporal information.
If the authors can provide clear and well-reasoned responses to these concerns with a revised manuscript, I would be open to considering an increase in the rating score.


[1] Wenliang Dai, Junnan Li, Dongxu Li, Anthony Meng Huat Tiong, Junqi Zhao, Weisheng Wang, Boyang Li, Pascale Fung, and Steven Hoi. Instructblip: Towards general-purpose vision-language models with instruction tuning. arXiv preprint arXiv:2305.06500, 2023.

[2] Haotian Liu, Chunyuan Li, Qingyang Wu, and Yong Jae Lee. Visual instruction tuning. arXiv preprint arXiv:2304.08485, 2023a

[3] Peng Gao, Jiaming Han, Renrui Zhang, Ziyi Lin, Shijie Geng, Aojun Zhou, Wei Zhang, Pan Lu, Conghui He, Xiangyu Yue, Hongsheng Li, and Yu Qiao. Llama-adapter v2: Parameter-efficient visual instruction model. arXiv preprint arXiv:2304.15010, 2023.

[4] Deyao Zhu, Jun Chen, Xiaoqian Shen, Xiang Li, and Mohamed Elhoseiny. Minigpt-4: Enhancing vision-language understanding with advanced large language models. arXiv preprint arXiv:2304.10592, 2023.

[5] Sebastian Böck, Filip Korzeniowski, Jan Schlüter, Florian Krebs, and Gerhard Widmer. madmom: a new Python Audio and Music Signal Processing Library. In Proceedings of the 24th ACM International Conference on Multimedia, 2016. doi: 10.1145/2964284.2973795.

[6] Peiyi Wang, Lei Li, Liang Chen, Dawei Zhu, Binghuai Lin, Yunbo Cao, Qi Liu, Tianyu Liu, Zhifang Sui: Large Language Models are not Fair Evaluators. CoRR abs/2305.17926 (2023)

[7] Yusong Wu, Ke Chen, Tianyu Zhang, Yuchen Hui, Taylor Berg-Kirkpatrick, and Shlomo Dubnov. Large-scale contrastive language-audio pretraining with feature fusion and keyword-to-caption augmentation. In ICASSP 2023-2023 IEEE International Conference on Acoustics, Speech and Signal Processing (ICASSP), pp. 1–5. IEEE, 2023.

[8] Benjamin Elizalde, Soham Deshmukh, Mahmoud Al Ismail, and Huaming Wang. Clap learning audio concepts from natural language supervision. In ICASSP 2023-2023 IEEE International Conference on Acoustics, Speech and Signal Processing (ICASSP), pp. 1–5. IEEE, 2023.

[9] Prafulla Dhariwal, Heewoo Jun, Christine Payne, Jong Wook Kim, Alec Radford, and Ilya Sutskever. Jukebox: A generative model for music. arXiv preprint arXiv:2005.00341, 2020.

**Questions:**

1.	Why does the dataset contain 1.2M text samples and only 164K audio clips, despite there being only three types of tasks?

2.	Is LLark limited by the upper bounds of the MIR model's performance used for metadata augmentation, and if so, how does this limitation impact its performance?

3.	In Figure 4, all models seem to have low performance. Is it possible that the performance of baselines is lower than random guessing?

4.	Section 6.2 suggests that the improved genre performance of ImageBind-LLM [10] is attributed to genre tags used in pre-training. However, according to my knowledge, ImageBind [11] is trained on image-related pairs only, and there are no audio-text pairs used during pre-training. Why are genre tags relevant in this context?

5.	How many raters were involved in the human evaluations for captioning and reasoning tasks?

[10] Jiaming Han, Renrui Zhang, Wenqi Shao, Peng Gao, Peng Xu, Han Xiao, Kaipeng Zhang, Chris Liu, Song Wen, Ziyu Guo, et al. Imagebind-llm: Multi-modality instruction tuning. arXiv preprint arXiv:2309.03905, 2023.

[11] Rohit Girdhar, Alaaeldin El-Nouby, Zhuang Liu, Mannat Singh, Kalyan Vasudev Alwala, Armand Joulin, and Ishan Misra. Imagebind: One embedding space to bind them all. In Proceedings of the IEEE/CVF Conference on Computer Vision and Pattern Recognition, pp. 15180–15190, 2023.

---

> ### Author Response · Authors · 2023-11-16
> **Reviewer zBmL response (1/5)**
>
> We are grateful to the reviewer for their thoughtful and detailed response! We are particularly glad to see that the reviewer observed the “impressive results,” “ potential for practical applications”, the potential to “provides reusable insights within the realm of music information retrieval,” and our efforts to “ensure data accessibility and reproducibility” by utilizing publicly accessible, permissively-licensed datasets.
>
> The reviewer provides a useful list of weaknesses, and questions. We address these individually inline below, but would like to acknowledge up front that we made several additions, clarifications, and improvements to the paper to incorporate the reviewers’ detailed feedback and address the reviewers’ concerns about the paper, framing, and results.
>
> **Weaknesses:**
>
> 1. *Section 1 mentions LLark's architecture as a contribution, but the integration of large language models (LLM) and domain-specific models has been previously explored in the literature [1-4], as properly cited in the paper. To strengthen the paper, consider providing a more detailed explanation of what sets LLark's architectural design apart from existing implementations.*
>
>     Thank you - we have updated the language in Section 1. To summarize here, the main novelty is in using a *generative* model as the encoder, but our intended primary contribution is *not* the novelty of the architectural components, but in demonstrating that this architecture is effective when paired with our data pipeline. As we discuss in depth in our response to Reviewer gHco (points (a), (b) in that response), we believe this contribution -- focused on data-centric machine learning for multimodal modeling -- is in line with the contributions of other studies in the vision-language space [5-10], but our work is the most comprehensive such effort in the music space.
>
> 2. *Limiting the audio encoder to 25-second clips (mentioned in Sections 4.1 and 7) is a notable drawback, as it lacks the understanding of important contextual information in real-world scenarios (most music audio is longer than 25 seconds), leading to potential misinterpretations and missed insights in audio content analysis.*
>
>     We acknowledge that this is a limitation of the work as presented (as discussed in Section 7). However, this is not a fundamental limitation of the approach. The audio encoding size could be extended indefinitely (up to the language model’s context window size; 4096 tokens in the case of Llama 2 7B). This could be achieved by passing chunked audio into Jukebox, and could be further compressed to accommodate longer audio by decreasing the audio frame rate (currently 10fps, or 250 tokens per 25-second clip). We adopt the 25-second limitation due to convenience in this work, and to limit the computational expense of using Jukebox (since audio clips in these datasets can be an hour or more). We leave analysis of longer audio encodings to future work - but note that our code can easily support such experiments, as it can run on arbitrarily-sized audio encodings out of the box.
>
> 3. *Section 4.2.1 should investigate how LLark's performance is affected by the features extracted from traditional MIR models [5], especially regarding temporal-related music properties like timestamped chords and beat grids. Additional ablation studies should be included to confirm the performance gain of metadata augmentation.*
>
>     We agree that this would be a useful analysis, and would have liked to conduct this absent computational or budgetary constraints. However, an ablation study which removed the metadata augmentation would require completely rerunning our entire dataset creation pipeline, withholding the augmented metadata features when the query-response pairs are created (the dataset creation, not the training, was the most expensive component of our work, so this was not tractable for us). Additionally, we felt the results of this ablation would be clear: we would expect a performance degradation on the “music understanding” tasks in particular, due to the lack of any supervision on these tasks during training, and a corresponding decrease in musical detail in free-text tasks. Our open-source code release would enable others to conduct this analysis in the future -- or, perhaps more likely, to *extend* the metadata augmentation to further improve the model by incorporating more and better pseudolabeling models, as MIR research progresses and new models become available.

---

> ### Author Response · Authors · 2023-11-16
> **Reviewer zBmL response (2/5)**
>
> 4. *In Sections 6.3 and 6.4, it’s ok to rely on human evaluation as we don't have holistic objective metrics to assess captioning and reasoning tasks. But leaning entirely on non-experts for this might not be the best call. It might be better to balance this out with some expert raters rather than going all-in on non-experts.*
>
>     The reviewer is highlighting a crucial and unresolved challenge in the evaluation of text generation models in general (discussed in e.g. [1]), and for music in particular. We agree that non-experts are limited in what they can assess. Overall, our approach to evaluation was to let existing works in the MIR and language modeling space be our guides, to maximize our ability to compare results to previous works. To our knowledge, prior work conducting human evaluation of audio-language models has exclusively used non-experts for both musical and non-musical applications [e.g. 2, 3] -- but human evaluations tend to be rare due to the high cost of conducting these studies. We believe there are three separate considerations raised by the reviewers’ point (and others in the review):
>     * **Objective vs. subjective metrics:** Any human evaluation is necessarily subjective. Objective metrics can help to offset the human element from evaluations, helping us to avoid “leaning entirely on non-experts” while measuring different behaviors. While the reviewer is correct that “we don’t have holistic objective metrics to assess captioning and reasoning tasks”, we provide several objective metrics assessing our models’ outputs on the tasks discussed in 6.3 and 6.4, in the supplementary material. For example: (i) Figures 11 and 12, which show that LLark provides captions that are competitive with existing models in terms of both diversity and length, for a fixed prompt (keeping in mind that the higher diversity of tokens in LLaMA-Adapter are largely due to detailed hallucinations, as our user study shows; we can provide examples if the reviewer would find this useful); (ii) Figure 13, which shows that LLark can follow instructions; (iii) Tables 3 and 5, which show (using GPT-4 as judge) that LLark consistently contains more musical detail in its responses; and (iv) Table 6, which shows that, with minimal prompt tuning (i.e. asking for a “short” caption vs. the standard captioning prompt), LLark is competitive with or outperforms captioning models even in its ability to match reference captions in various language metrics (BLEU, BLEU-4, METEOR, ROUGE, CIDER). We caution against overreliance on the language metrics, however, as these only measure adherence to a reference caption, which can reflect the biases of the original captions [4] and penalizes even high-quality captions which deviate from the original caption.
>     * **Experts vs. non-experts:** We acknowledge the reviewer’s concerns about non-experts being used in the subjective evaluations, and attempted to highlight this limitation in Section 7. Our goal is to construct a model which is useful to both experts and non-experts, which motivated our use of non-experts for selected tasks. Our study design (including the use of non-experts) was matched to recent language modeling evaluations, for example [1, 2, 3] which use non-experts for language, music, and audio models. As the reviewer is undoubtedly aware, the cost of assembling musical experts to evaluate our model was also prohibitively high. However, since our intention was to construct a model that is useful for a broad user base, we believe non-expert evaluations are still useful for this. Furthermore, there is not only a binary choice between expert vs. non-expert evaluators -- objective metrics (discussed above) are also a useful tool in assessing model quality without relying exclusively on non-experts.
>     * **Correctness of musical aspects of text responses:** One possible concern with overreliance on non-expert human evaluation is that non-experts cannot easily assess the correctness of musical information (tempo, key, etc.). We share this concern. We point to our formal evaluations on music understanding tasks (Section 6.1), which evaluate the model’s outputs of musical properties that may be otherwise assessed by experts (key, tempo, and instrument estimates) and show that the model’s musical responses regarding these attributes are of high correctness. At the same time, we acknowledge that incorrect, but convincing-sounding outputs may affect the model. As a result, we have (1) updated the ethics section of the paper to reflect these issues with truthfulness of outputs; (2) added a Model Card [11] to the paper (Section M); (3) added a “Failure Modes” section to the paper (Section N) to highlight and categorize cases where these behaviors may emerge with LLark to ensure that users and researchers alike are appraised of the risks unique to music-language models. We also propose strategies to address each failure mode.

---

> ### Author Response · Authors · 2023-11-16
> **Reviewer zBmL response (3/5)**
>
> 5. *In Figure 4, the matching rates are quite low across all models (around the random guessing level), leading to questions about whether LLark's performance on certain tasks is only marginally better than random guessing. Section 6.4 attributes this result to “limitations in the musical expertise of the (non-expert) raters in our study," further emphasizing my previous concerns about relying solely on non-experts for human evaluation.*
>
>     We acknowledge the reviewers’ concern here. Open-ended reasoning tasks are, by far, the most challenging task in our study, as they require combining music understanding, language modeling, and external “world knowledge” to answer correctly. While our model outperforms existing instruction-following audio-text models, there is room for future improvement. We believe that the rigorous formal reasoning evaluation in our study, which has not been attempted for similar models in either music-, audio-, or vision-language models to our knowledge, provides a useful benchmark for future work in this direction. We provide two additional points of context:
>     * Our above discussion of objective vs. subjective metrics is relevant here. In particular, the subjective metrics can only tell “part of the story” with respect to reasoning evaluation. We hesitate to rely only on audio-text matching (see next point), and believe that the objective metrics for this task also provide useful context. Specifically, this includes the results in Table 3 and Figure 12.
>     * In practice, “random guessing” is not the lower bound for audio-text matching performance for models in our study, and “perfect matching” (100%) may be an impractically high bar. Recall that our design presents raters with one piece of audio + a text query, and three responses to that query (from the same model, in random order).
>
>       **Lower bound:** we observe that many models produce “convincing hallucinations” that attracted reviewers at a high rate whenever they appeared, regardless of whether they were correct or not. This was a major contributor to the poor performance of ImageBind-LLM (see Figure 12, which shows that ImageBind-LLM’s responses were both extremely long, and had a high number of unique tokens; these often described fictional artists and elaborate visual scenes not associated with the input but were selected by raters at high rates). In contrast, for LTU-AS, the descriptions tended to be short and generic (as shown in Figure 12), which led reviewers to select the more detailed responses at a high rate. Both behaviors can lead to lower-than-chance performance. (Consider the extreme case, not possible with this specific design, where one answer is always chosen regardless of the correct response; in that design the matching rate would plummet to 1 / size of response pool, effectively zero.)
>
>       **Upper bound:** note that our audio-text matching setup is a very challenging task even under the best of circumstances: it requires that a response addresses a question correctly, and that the response matches the audio in sufficient detail such that a rater can link the textual response to the audio in the presence of other potentially-similar responses from other tracks. We hypothesize that even in the presence of expert-human-provided responses, matching rates would likely not achieve 100%. This is particularly true for non-expert raters, as the reviewer notes, since they may not be able to match attributes such as key or tempo to the audio, and again motivates a more holistic approach to our evaluations (i.e. considering the level of musical detail in Table 3, and the music understanding results in Table 2). As a result, while we acknowledge that performance even higher than LLark’s is achievable, we do not expect that 100% matching rate would be possible.
> We have also added further analysis summarizing these points to Section G.3.
>
> 6. *With no clear indication of randomness in option order (or maybe I missed?), such as whether LLark is consistently presented as the first option, this raises concerns about potential biases [6] in the evaluation results of GPT-4 in Sections 6.3 and 6.4.*
>
>     The ordering of all models and answer choices in all of our human and GPT-4 studies are always randomized. This is mentioned in the “interface” sections in L.1 (captioning) and L.2 (reasoning); we have also added it to the main text in Sections 6.3 and 6.4.
>
> 7. *Wrong citation in Appendix F: the ablation study used the CLAP developed by LAION [7], but wrongly cited the Microsoft one [8]. This made me confused for a while. Please correct the citation.*
>     We have corrected this mistake; thank you for noting it.

---

> > ### Author Response · Authors · 2023-11-16
> > **Reviewer zBmL response (4/5)**
> >
> > 8. *In Appendix F, ablation studies may not offer a fair comparison due to the significant model size differences between CLAP [7] and Jukebox [9]. To assess temporal information's significance, please consider an additional ablation study where Jukebox's weights are still frozen but average pooling is applied to its output embeddings without preserving temporal information.*
> >
> >     We acknowledge the reviewers’ concern that this ablation does not control for the temporal dimension of the embedded audio, and have added more detail on this point in the ablation study (Section F.1). We explored this approach (average pooling over the entire time dimension in Jukebox embeddings) in the early stages of the project, and found that adding the temporal information improved the model substantially, but did not conduct a complete ablation study. We commit to preparing such a study for the camera-ready deadline and believe it will make an interesting comparison -- but also note that it is an explanatory ablation study and ultimately will not improve the model’s results, only our understanding of them.
> >
> > **Questions**
> >
> > 1. *Why does the dataset contain 1.2M text samples and only 164K audio clips, despite there being only three types of tasks?*
> >
> >     Multiple query-response pairs could be generated for each audio example (this allowed us to generate the large sets of annotations for each element of each dataset; i.e. for MagnaTagATune which contains nearly 200 binary annotations, many questions could be written for a single piece of audio). This indicates that, overall, there were on average 7.31 query-response pairs for each audio clip across the three task families. We have added a clarification to the end of Section 4.2.
> >
> > 2. *Is LLark limited by the upper bounds of the MIR model's performance used for metadata augmentation, and if so, how does this limitation impact its performance?*
> >
> >     The reviewer is correct, Llark cannot outperform the model used to label the data. However,  Llark is not tied to the specific feature extractors used in our work: as these models improve over time, better labelers can be used. So in theory, our pipeline can be adapted to utilize any of the task-specific SOTA models for these and other tasks. We added a discussion of this point to the future work section. We also provide the performance of the key and tempo “teacher models” in the paper in Section E1.1 (key) and E1.2 (tempo), which we report here for convenience: the key model achieved a MIREX score of 0.729 and the tempo model achieved an Acc2 of 0.947, indicating that LLark was closer to saturating the key performance than the tempo performance. (We hypothesize that this is due to the challenges of learning numeric labels -- i.e., performing regression -- with language models, discussed in Section F and illustrated in Figure 7).
> >
> > 3. *In Figure 4, all models seem to have low performance. Is it possible that the performance of baselines is lower than random guessing?*
> >
> >     Yes. This is part of the motivation for our study design: we found that raters were susceptible to choose “persuasive hallucinations” -- highly detailed responses that were not linked to the specific audio in a question -- if we did not control for the model providing the responses as we did in our audio-text matching design. If the “true” output for a model on a song is too generic, users will often choose a more detailed, if hallucinated, response. Please also see our discussion of this in “weaknesses”, point 5, above. We have also added further analysis along the lines of our response to Section G.3.
> >
> > 4. *Section 6.2 suggests that the improved genre performance of ImageBind-LLM [10] is attributed to genre tags used in pre-training. However, according to my knowledge, ImageBind [11] is trained on image-related pairs only, and there are no audio-text pairs used during pre-training. Why are genre tags relevant in this context?*
> >
> >     We have updated the paper text to clarify (Section 6.2). Summarizing briefly, our goal here was to acknowledge the possibility that ImageBind was explicitly trained for genre recognition (due to the likelihood of genre tags (as these are widely used in tagging “web video” data like that used in training ImageBind, such as in hashtags like #jazz) and, in particular, that the specific songs in GTZAN were in this training dataset (because GTZAN is composed primarily of common popular songs and artists). If this is the case (which we cannot verify due to inaccessibility of ImageBind training data), then ImageBind’s strong performance may be the result of seeing this data during training (as image-text and audio-image pairs, which then allow it to align the audio with the genre text tags). Even if exact genre information is not present, having GTZAN tracks in the training data would provide direct supervision that would make its performance on GTZAN not zero-shot (and not even a valid “test” set for ImageBind).

---

> > > ### Author Response · Authors · 2023-11-16
> > > **Reviewer zBmL response (5/5)**
> > >
> > > 5. *How many raters were involved in the human evaluations for captioning and reasoning tasks?*
> > >
> > >     There were between 382 and 799 workers in each task. As we describe in the supplementary details on our human evaluation procedure, we also require raters to pass quality control assessments, which leads to a small fraction of raters around 10%) being removed from the final pool. We also apply a control setting in Appen which ensures that no more than 5% of ratings come from a single rater in any task. We have added this information to the evaluation experiment details in Section L.
> > >
> > > **We understand that this response is lengthy -- but the reviewer gave detailed feedback and we wanted to be sure to discuss each point. If we have not sufficiently addressed any of the reviewers’ concerns, please let us know so that we may further address during the author response period.**
> > >
> > > References
> > >   * [1] Touvron, Hugo, et al. "Llama 2: Open foundation and fine-tuned chat models." arXiv preprint arXiv:2307.09288 (2023).
> > >   * [2] Doh, SeungHeon, et al. "Lp-musiccaps: Llm-based pseudo music captioning." arXiv preprint arXiv:2307.16372 (2023).
> > >   * [3] Gong, Yuan, et al. "Listen, Think, and Understand." arXiv preprint arXiv:2305.10790 (2023).
> > >   * [4] Lee, Minhee, SeungHeon Doh, and Dasaem Jeong. "Annotator Subjectivity in the MusicCaps Dataset." (2023).
> > >   * [5] Liu, Haotian, et al. "Visual instruction tuning." arXiv preprint arXiv:2304.08485 (2023).
> > >   * [6] Li, Bo, et al. "Otter: A multi-modal model with in-context instruction tuning." arXiv preprint arXiv:2305.03726 (2023).
> > >   * [7] Awadalla, Anas, et al. "Openflamingo: An open-source framework for training large autoregressive vision-language models." arXiv preprint arXiv:2308.01390 (2023).
> > >   * [8] Gao, Peng, et al. "Llama-adapter v2: Parameter-efficient visual instruction model." arXiv preprint arXiv:2304.15010 (2023).
> > >   * [9] Han, Jiaming, et al. "Imagebind-llm: Multi-modality instruction tuning." arXiv preprint arXiv:2309.03905 (2023).
> > >   * [10] Zhu, Deyao, et al. "Minigpt-4: Enhancing vision-language understanding with advanced large language models." arXiv preprint arXiv:2304.10592 (2023).
> > >   * [11] Mitchell, Margaret, et al. "Model cards for model reporting." Proceedings of the conference on fairness, accountability, and transparency. 2019.

---

> > > > ### Author Response · Authors · 2023-11-18
> > > > **Request to review author response**
> > > >
> > > > Dear reviewer zBmL,
> > > >
> > > > Thank you for your time and effort spent reviewing our paper. We have submitted our response to your concerns, including a revised version of our paper (updates are in red text). Please let us know your comments, whether our revisions and clarifications have addressed your concerns, or whether we can provide further clarification. Thank you again!

---

> ### Author Response · Authors · 2023-11-21
>
> Once again, we thank the reviewer for your time reviewing the paper. As the author-reviewer phase is ending soon, we request the reviewer to please review the author response and let us know whether the comments and revisions to the paper (indicated in red text) have addressed your concerns, or whether we can provide any further clarification before the discussion window closes. Thank you!

---

### Official Review · Reviewer_gHco · 2023-11-03

**Soundness:** 3 good
**Presentation:** 3 good
**Contribution:** 2 fair
**Rating:** 6
**Confidence:** 4

**Summary:**

This project uses a pretrained audio encoder, and a pretrained language encoder. A learned mapping of the audio encoder output to the embedding space of the language encoder is introduced. A learned output language decoder is also learned. All of this is trained by optimising a loss on (audio in, text in, text out) triples. These training triples are constructed with an LLM + some ad hoc filtering steps, and are basically q & a "what sort of song is this" etc.

The pieces are well known and this is largely an empirical study. The results are promising and the paper is easy to read.

**Strengths:**

The paper is easy to read, and the results look pretty good. Although it merely rehashes existing datasets and models, it seems like a nice combination and well thought out.

**Weaknesses:**

It's largely an empirical study / engineering type of project. The authors seem to say they will give code and data but no checkpoint.

**Questions:**

Have you included the latest results on the benchmarking tasks? I am not a specialist in this area and I am curious if other reviewers find missing comparisons.

---

> ### Author Response · Authors · 2023-11-16
> **Reviewer gHco response (1/3)**
>
> We are grateful to the reviewer for the thoughtful response, and glad that the reviewer assessed that “[t]he results are promising,” that the work is “well thought out” and the paper is easy to read. As we discuss below, the reviewers’ questions also helped us to improve the clarity of the paper - thank you!
>
> The reviewer’s generally positive feedback seems somewhat at odds with their rating. It appears that the reviewers’ main reservation about our study is the nature of the contribution: in particular, the reviewer appears to feel that the empirical nature of the work, and a belief that it “merely rehashes existing datasets and models,” mean the work is not significant. We would respectfully like to reframe our contribution to the reviewer, and put it into context with existing works. Our paper falls in line with existing highly impactful contributions in the multimodal modeling space, and with work on data-centric machine learning, where the emphasis is on empirically understanding how to build high-performing models and on solving important applied problems, regardless of the architecture used.
>
> We provide a few specific responses below on how our work relates to others in the data-centric multimodal learning space:
>
>   * (a) *Using existing models is a feature, not a bug:* We believe the claim that we use “existing datasets and models” is a benefit, not a drawback, of our approach. Our ablation studies (Figures 6, 7) show the impact of using these specific models, and demonstrates that starting from the highest-quality pretrained (language, audio) models is critical to the quality of the resulting output. Utilizing existing resource-intensive foundation models is also an effective use of training resources. Furthermore, neither our study nor the work of others suggests that new (audio, language) models are necessary for multimodal music understanding (see next point).
>
>   * (b) *Many impactful multimodal models utilize pretrained modality-specific encoders:* It is standard practice, and in line with the contributions in other multimodal models, to use “existing datasets and models” as the building blocks of a multimodal model. This includes recent SOTA multimodal models in the vision-language space, such as LLaVA [1] (Llama, CLIP), Otter [2] and OpenFlamingo [3] (Llama, CLIP), LLaMA-Adapter [4] (Llama, CLIP), ImageBind-LLM [5] (Llama, ImageBind) Mini-GPT4 [6] (Vicuna, ViT).
>
>   * (c) *Open-source data is a benefit in the music domain:* The reviewer also voices a concern about using existing datasets. To this, we offer  two considerations:
>     * (1) All of the works discussed in the previous paragraph [1-6] start by building a multimodal training set from existing open-source datasets. We follow this widely-used approach, albeit in the music domain. Our contribution is in line with these widely-cited and used works. Using open-source data is a benefit for music researchers in particular, as it allows others to reliably reproduce and extend our results (due to the strict regulations surrounding recorded music, this can be challenging for closed-source datasets!). Reviewer zBmL also highlighted this as a **strength** of the paper: “The paper leverages publicly accessible, open-source, and permissively licensed music datasets for training, which ensures data accessibility and reproducibility.”
>     * (2) We do not simply use unmodified off-the-shelf data to train the model. Indeed, our data preprocessing (via instruction-tuning across 3 task families) and metadata augmentation (which uses music-specific feature extractors to add critical musical information) is a key contribution of our work. Again, this is in line with existing instruction-tuning works in the language and vision domains [1-6]. LLark is the most comprehensive such effort in the music domain.
>
> * (d) *Simple multimodal projection architecture is in line with existing work:* The simplicity of the multimodal adapter is both a benefit of our approach, and in line with existing work. Recent state-of-the-art models across the multimodal space have shown that simple adapter layers are highly effective: e.g. LLaVA [1], Mini-GPT4 [6], FUYU [7], among others. Given no clear reason to add further complexity to the architecture (and the emerging likelihood that this is the most effective technique for multimodal alignment), we felt this approach was appropriate and is validated by our results.

---

> > ### Author Response · Authors · 2023-11-16
> > **Reviewer gHco response (2/3)**
> >
> > Additionally, none of the above-cited works are in the music space -- and so it was not obvious or trivial to consider that this approach would work for music. Our main contribution is in building a high-quality multimodal model for music by leveraging instruction-tuning for music, which includes our novel data annotation and augmentation strategy. This is particularly important in the music domain, where data annotation is expensive, labor-intensive, and lacks alignment across datasets (different datasets have different features). Demonstrating that we could overcome these limitations with an effective dataset construction strategy in LLark is a differentiating factor in our work.
> >
> > **Answers to specific questions:**
> >
> > * *“Have you included the latest results on the benchmarking tasks?”* Our goal in this study is to compare to state-of-the-art *audio-language models* on the target tasks, which allows for an apples-to-apples comparison to LLark. Our evaluations compare against the open-source SOTA for audio-language modeling at the time of our submission. If the reviewer is asking about task-specific models (i.e., a tempo estimation model, or a key estimation model), we do not explicitly compare to such models, because (a) they are fundamentally far more limited than an audio-text model, and (b) they are often explicitly trained only on the target task and dataset (i.e. only on the training split of the target task). However, we provide some numbers from related literature (provided by reviewer p6wE) here (these references and results have also been added to the paper):
> >     * **Tempo estimation on Giant Steps:** 88.6% Acc2 score (the model of [8], as benchmarked in [11]). Note that the model of [8] was trained on MTG-tempo, from the same source (Beatport) and genre distribution as the evaluation dataset, whereas Llark was not.
> >     * **Key estimation on Giant Steps:** 74.3% accuracy (the model of [9], as cited in [10], Table 1). Note that the model of [9] was trained on MTG-Key, from the same source (Beatport) and genre distribution as the evaluation dataset, whereas Llark was not.
> >     * **Genre classification on GTZAN:** 83.5% (model of [11], as cited in [10], Table 2). This result is achieved by fine-tuning (via probing) directly on the GTZAN dataset. We discuss issues with the GTZAN dataset in our response to p6wE below.
> >     * We are not aware of prior published results performing genre recognition or instrument recognition on MedleyDB, but believe that our evaluations provide strong baselines via other audio-text models.
> >     * Please note that these are task-specific, not generalist models. We expect that the “best” model for a given task will often be a model architected and trained specifically for that task, which is why we focus on evaluating against other audio language models, not task-specific models, in Section 6.1.
> >
> > * *“The authors seem to say they will give code and data but no checkpoint.”*
> >   * We have provided the complete end-to-end code to recreate our datasets and train the model in the supplementary material with our initial submission, including Python scripts and Dockerfiles for executing the code. If the reviewer has any issues accessing this, please let us know!
> >   * All of the raw data used in our work is entirely open-source, and we provide links to the sources in the supplement (Section H). Due to the non-derivatives license of many of the audio files used in these datasets, we are legally prohibited from providing model checkpoints, but our training code enables retraining or extended the model.
> >
> > As we noted above, if there are other concerns the reviewer has that led to the lower rating in light of their generally positive feedback, we would appreciate further clarification so that we may address them during the response period.

---

> ### Author Response · Authors · 2023-11-16
> **Reviewer gHco response (3/3)**
>
> References:
>
> [1] Liu, Haotian, et al. "Visual instruction tuning." arXiv preprint arXiv:2304.08485 (2023).
>
> [2] Li, Bo, et al. "Otter: A multi-modal model with in-context instruction tuning." arXiv preprint arXiv:2305.03726 (2023).
>
> [3] Awadalla, Anas, et al. "Openflamingo: An open-source framework for training large autoregressive vision-language models." arXiv preprint arXiv:2308.01390 (2023).
>
> [4] Gao, Peng, et al. "Llama-adapter v2: Parameter-efficient visual instruction model." arXiv preprint arXiv:2304.15010 (2023).
>
> [5] Han, Jiaming, et al. "Imagebind-llm: Multi-modality instruction tuning." arXiv preprint arXiv:2309.03905 (2023).
>
> [6] Zhu, Deyao, et al. "Minigpt-4: Enhancing vision-language understanding with advanced large language models." arXiv preprint arXiv:2304.10592 (2023).
>
> [7] Rohan Bavishi et al. Fuyu-8B: A Multimodal Architecture for AI Agents. https://www.adept.ai/blog/fuyu-8b
>
> [8] Schreiber, Hendrik, and Meinard Müller. "Musical tempo and key estimation using convolutional neural networks with directional filters." arXiv preprint arXiv:1903.10839 (2019).
>
> [9] Korzeniowski, Filip, and Gerhard Widmer. "End-to-end musical key estimation using a convolutional neural network." 2017 25th European Signal Processing Conference (EUSIPCO). IEEE, 2017.
>
> [10] Li, Yizhi, et al. "MERT: Acoustic Music Understanding Model with Large-Scale Self-supervised Training." arXiv preprint arXiv:2306.00107 (2023).
>
> [11] McCallum, Matthew C., et al. "Supervised and unsupervised learning of audio representations for music understanding." arXiv preprint arXiv:2210.03799 (2022).
>
> [12] de Souza, Mila Soares de Oliveira, Pedro Nuno de Souza Moura, and Jean-Pierre Briot. "Music Tempo Estimation via Neural Networks--A Comparative Analysis." arXiv preprint arXiv:2107.09208 (2021).

---

> > ### Author Response · Authors · 2023-11-18
> > **Request to review author response**
> >
> > Dear reviewer gHco,
> >
> > Thank you for your time and effort spent reviewing our paper. We have submitted our response to your concerns, including a revised version of our paper (updates are in red text). Please let us know your comments, whether our revisions and clarifications have addressed your concerns, or whether we can provide further clarification. Thank you again!

---

> ### Author Response · Authors · 2023-11-21
>
> Once again, we thank the reviewer for your time reviewing the paper. As the author-reviewer phase is ending soon, we request the reviewer to please review the author response and let us know whether the comments and revisions to the paper (indicated in red text) have addressed your concerns, or whether we can provide any further clarification before the discussion window closes. Thank you!

---

### Author Response · Authors · 2023-11-22
**Please review the author responses**

Dear reviewers,

We wanted to state, one final time, our appreciation for your participation in the review process thus far.

We have provided extensive responses and revisions in order to respond to each reviewers' comments and concerns about the paper. Since it is the final day in the author-reviewer dialogue period, we would like to request again that reviewers please notify us if there are any further clarifications we can make to address their concerns about our paper, or any outstanding clarifications or concerns that have not been addressed by our responses and revisions. We hope that addressing reviewers' concerns has been able to impact our score positively.

---

### Meta-Review · Area_Chair_X9Py · 2023-12-06

**Metareview:**

This project uses a pretrained audio encoder, and a pretrained language encoder. A learned mapping of the audio encoder output to the embedding space of the language encoder is introduced. A learned output language decoder is also learned. All of this is trained by optimising a loss on (audio in, text in, text out) triples. These training triples are constructed with an LLM + some ad hoc filtering steps, and are basically q & a "what sort of song is this" etc.

Strengths:
- The paper is easy to read, and the results look pretty good. Although it mostly rehashes existing datasets and models, it seems like a nice combination and well thought out.
- Authors promise to make code available, and the model is using publicly available data only, so it could be seeding future work (no checkpoint, though)

Weaknesses:
- Reviewers and authors agree that the title of the paper should be changed to better reflect the actual contents of the paper as a multi-modal rather than a foundational model
- The paper has limited architectural and algorithmic novelty (LLark is not the first paper trying to align audio and text (CLAP) or music and text (Mulan)), making it less of a match to ICLR
- Llark performs well, but does not truly push SOTA on benchmark tasks (despite leveraging two large models, Jukebox-5b and Llama2-7B), so it can also not be considered a "performance reference"

**Justification For Why Not Higher Score:**

At present, the title of the PDF is different from the title of the paper in the review system - which would be a desk reject.

Authors changed the title after discussion with reviewer p6wE, would need to decide if this is a "foundation" model, or something else.

**Justification For Why Not Lower Score:**

n/a

---

### Decision · Program_Chairs · 2024-01-16

Reject